# Just One Layer Norm
# Guarantees Stable Extrapolation

**Juliusz Ziomek**[*,†]**, George Whittle**[*]**, Michael A. Osborne**
Machine Learning Research Group, University of Oxford
[*] Equal Contribution
[†] Corresponding Author
{juliusz, george, mosb} @ robots.ox.ac.uk

## Abstract

In spite of their prevalence, the behaviour of Neural Networks when extrapolating far from the training distribution remains poorly understood, with existing results limited to specific cases. In this work, we prove general results—the first of their kind—by applying Neural Tangent Kernel (NTK) theory to analyse infinitely-wide neural networks trained until convergence and prove that the inclusion of just one Layer Norm (LN) fundamentally alters the induced NTK, transforming it into a bounded-variance kernel. As a result, the output of an infinitely wide network with at least one LN remains bounded, even on inputs far from the training data. In contrast, we show that a broad class of networks without LN can produce pathologically large outputs for certain inputs. We support these theoretical findings with empirical experiments on finite-width networks, demonstrating that while standard NNs often exhibit uncontrolled growth outside the training domain, a single LN layer effectively mitigates this instability. Finally, we explore real-world implications of this extrapolatory stability, including applications to predicting residue sizes in proteins larger than those seen during training and estimating age from facial images of underrepresented ethnicities absent from the training set.

## 1 Introduction

Neural Networks (NN) have dominated the Machine Learning landscape for more than a decade. Various deep architectures became the state-of-the-art approaches across countless applications. However, despite this empirical success, we still have a limited understanding of how these models learn and how they behave when making predictions.

What poses particular difficulty is analysing the network's prediction as inputs get further away from the training data. Clearly, a network achieving a low training set error should output values close to the ground truth labels on the training points. Yet, it is difficult to say how the network will behave outside of the training domain, as the training process does not necessarily constrains the network's output there. As such, the extrapolatory behaviour of these networks is very much undefined.

Lack of understanding of this extrapolatory behaviour can be dangerous. For example, a deep learning-based control system that applies voltages to a robot arm based on camera inputs can encounter pixel values with brightness exceeding those found in the training set. If this causes the network to output extreme values, this could easily lead to a serious safety hazard in the system. As such, understanding how networks behave outside of the training domain and whether their predictions are stable out of distribution is critical.

One of the most complete theoretical approaches to studying the behaviour of Deep Neural Networks is the Neural Tangent Kernel (NTK) theory [19, 23]. Previous work in that stream [23] managed to

39th Conference on Neural Information Processing Systems (NeurIPS 2025).

show that as the network width approaches infinity, the trained network's output is equivalent to the posterior mean of a Gaussian Process (GP) [29] with a specific kernel function that depends only on the network's architecture. While any real neural network has, of course, a finite width, in many cases, the predictions made by the NTK theory matched the empirical observations [19, 23, 1, 13].

In this work, we use NTK theory to analyse the effect on extrapolatory, or out-of-distribution (OOD), predictions of including Layer Normalisation (LN) [2] operations in a network. We show that for an infinitely-wide, fully-connected standard network with positive $n$-homogeneous nonlinearities, there always exists a fixed training dataset such that their predictions get pathologically large, i.e. "explode". On the other hand, we show that the NTK of infinitely-wide, fully-connected architectures containing at least one LN operation admits an upper bound for any input, no matter how far from the training domain. Using this, we show that such networks trained until convergence have a bounded output even infinitely far from the training domain, for any training set. We then show how these properties are useful across different practical applications, including extrapolating residue size prediction to proteins larger than those seen during training and age prediction for unrepresented ethnicities absent from training data. We detail our contributions below:

- We show with Theorem 3.1 that for infinitely-wide fully-connected positive $n$-homogeneous nonlinearities, but without any LN, the absolute value of the expectation (with expectation taken over initialisation) can be arbitrarily high outside of the training domain.

- We show with Theorem 3.3 that the inclusion of a single LN anywhere in the network imposes an upper bound on the absolute value of expected output anywhere in the domain. That is, they enjoy stable extrapolation.

- We verify these theoretical results on a number of toy problems and show that standard neural networks "explode" outside of the training domain, whereas the presence of at least one LN prevents this explosion from happening.

- We show how these properties are useful when making out-of-distribution predictions, such as extrapolating residue size prediction to proteins larger than those seen during training and age prediction for unrepresented ethnicities absent from training data.

Furthermore, we extend our extrapolation stability results with Theorem C.3 to a more general class of nonlinearities and with Theorem C.1 to, under certain circumstances, any architecture within the Tensor Program framework [35, 36, 37].

## 2   Preliminaries and Problem Setting

We assume we are solving a supervised learning problem given a dataset $\mathcal{D}_{\text{train}} = \{(\boldsymbol{x}_i, y_i)\}_{i=1}^{\theta}$, where for all $i = 1, \ldots, n$, by $\boldsymbol{x}_i \in \mathbb{R}^{n_0}$ we denote an $n_0$-dimensional input datapoint and by $y_i \in \mathbb{R}$ corresponding to the target variable. We assume there are no repeated datapoints, that is, $\boldsymbol{x}_i = \boldsymbol{x}_j \implies i = j$. We will denote by $\mathcal{X}_{\text{train}} = \{\boldsymbol{x}_i\}_{i=1}^n$ and $\mathcal{Y}_{\text{train}} = \{y_i\}_{i=1}^n$. We use this data to train a NN $f_\theta : \mathbb{R}^{n_0} \to \mathbb{R}$, parametrised by some $\theta \in \Omega \subset \mathbb{R}^d$. We measure the success on this learning problem by the MSE loss function $L(\theta; \mathcal{D}_{\text{train}}) = \frac{1}{2} \| f_\theta(\boldsymbol{x}_i) - y_i \|_2^2$. We assume the parameters of NN are initialised as described in Appendix A. They then evolve via gradient descent update as $\theta^t = \theta^{t-1} - \eta \nabla_\theta L(\theta; \mathcal{D}_{\text{train}})$ with learning rate $\eta > 0$ at each step $t > 0$. The work of [19] observed that (even finite) NNs during training follow a kernel gradient descent with a kernel:

$$\Theta(\boldsymbol{x}, \boldsymbol{x}'; \theta) = \langle \nabla_\theta f_\theta(\boldsymbol{x}), \nabla_\theta f_\theta(\boldsymbol{x}') \rangle,$$

also known as the *empirical NTK*. They then show that for an MLP, as the width of the smallest layer approaches infinity, the NTK converges to a limit $\Theta(\boldsymbol{x}, \boldsymbol{x}')$ that stays constant throughout the training process. Furthermore, the work of [23] showed that if the network is trained with learning rate small enough then in the limit $t \to \infty$, the network converges to:

$$f_{\theta^\infty}(\boldsymbol{x}^\star) = \Theta(\boldsymbol{x}^\star, \mathcal{X}_{\text{train}}) \Theta(\mathcal{X}_{\text{train}}, \mathcal{X}_{\text{train}})^{-1} (\mathcal{Y}_{\text{train}} - f_{\theta^0}(\boldsymbol{x}^\star)) + f_{\theta^0}(\boldsymbol{x}^\star),$$

where $f_{\theta^0}(\boldsymbol{x}^\star)$ is the output of freshly initalised network with parameters $\theta^0$ for input $\boldsymbol{x}^\star$. As for the standard parameterisation $\mathbb{E}[f_{\theta^0}(\boldsymbol{x}^\star)] = 0$, we thus also get that:

$$\mathbb{E}[f_{\theta^\infty}(\boldsymbol{x}^\star)] = \Theta(\boldsymbol{x}^\star, \mathcal{X}_{\text{train}}) \Theta(\mathcal{X}_{\text{train}}, \mathcal{X}_{\text{train}})^{-1} \mathcal{Y}_{\text{train}},$$

which is precisely the posterior mean of a GP with kernel function $\Theta(\boldsymbol{x}, \boldsymbol{x}')$ conditioned on data $\mathcal{D}_{\text{train}}$ evaluated at point $\boldsymbol{x}^\star$. While the original result of [23] assumed $\|\boldsymbol{x}^\star\| \leq 1$, this assumption can be lifted without affecting the limit, as we explain in Appendix B. This result is valid, as long as the infinite width limit $\Theta(\boldsymbol{x}, \boldsymbol{x}')$ exists. The work of [35, 36, 37] showed this limit exists for any combinations of basic operations (called Tensor Programs), which include Convolutions, Attention, and Layer Normalisations, to name a few. For completeness, we define the $n$-dimensional Layer Normalisation (LayerNorm) operation LN : $\mathbb{R}^n \to \mathbb{R}^n$ below:

$$\text{LN}(\boldsymbol{z}) = \frac{\boldsymbol{z} - \mu}{\sigma},$$

where $\mu = \frac{1}{n}\sum_{i=1}^{n}\boldsymbol{z}_i$ and $\sigma = \sqrt{\frac{1}{n}\sum_{i=1}^{n}(\boldsymbol{z}_i - \mu)^2}$ and the division and subtraction of vector $\boldsymbol{z}$ and scalars $\mu$ and $\sigma$ is executed for each element of $\boldsymbol{z}$.

## 3 Theoretical Results

We now proceed to analyse how the inclusion of LNs in the network's architecture affect the NTK, and therefore the behaviour of the trained networks. We conduct our analysis under the assumption that the network's activation function $\phi(\cdot)$ satisfies the following assumptions:

**Assumption 3.1.** Activation functions $\phi : \mathbb{R} \to \mathbb{R}$ are almost-everywhere differentiable and act element-wise.

**Assumption 3.2.** Activation functions $\phi : \mathbb{R} \to \mathbb{R}$ are positive $n$-homogeneous with $n > 0.5$, i.e. $\phi(\lambda x) = \lambda^n \phi(x) \ \forall \lambda > 0$.

**Assumption 3.3.** The minimum eigenvalue $\lambda_{\min}$ of the Gram matrix of the NTK over the training dataset, $\Theta_{\text{train}}$, is strictly positive, i.e. $\lambda_{\min} > 0$.

Note that the former of the the activation function assumptions is a requirement of all activation functions, and the latter is true for the popular ReLU [12] activation function and its Leaky [25] and Parametric [15] variants, which are specifically positive 1-homogeneous. This can easily be seen by noting that, for $\lambda > 0$, $\phi_{\text{ReLU}}(\lambda x) = \max(0, \lambda x) = \lambda \max(0, x) = \lambda^1 \phi_{\text{ReLU}}(x)$. The significance of $n > 0.5$ becomes clear when considering an important consequence of Assumption 3.2; the activation derivative $\dot{\phi}(\cdot)$, which is positive $n-1$-homogeneous, is square integrable with respect to the standard normal measure, i.e. $\dot{\phi}(\cdot) \in L^2(\mathbb{R}, \gamma)$, only if the monomial degree of the derivative close to 0 is greater than $-0.5$, which is essential for NTK analysis.

While other popular activation functions, such as GELU and Swish [18], do not satisfy the latter criteria due to their behaviour for small inputs, one can see that for large inputs the behaviour of these nonlinearities approaches positive $n$-homogeneity. In Appendix C, we extend our analysis to rigorously consider a broader class of nonlinearities containing almost all popular activation functions [10], observing that our most important result holds under these relaxed assumptions too.

The assumption on the eigenvalues of the Gram matrix is standard in all NTK literature, and essentially ensures neither the dataset nor the kernel is degenerate.

### 3.1 Standard Neural Networks are unbounded

We first proceed by analysing the extrapolatory behaviour of standard networks without LN operations:

**Theorem 3.1.** *Consider an infinitely-wide network $f_\theta(\boldsymbol{x})$ with nonlinearities satisfying Assumption 3.1 and Assumption 3.2, and fully-connected layers. Then, there exists a finite dataset $\mathcal{D}_{train} = (\mathcal{X}_{train}, \mathcal{Y}_{train})$ such that any such network trained upon $\mathcal{D}$ until convergence has* [1]

$$\sup_{\boldsymbol{x} \in \mathbb{R}^{n_0}} |\mathbb{E}\left[f_\theta(\boldsymbol{x})\right]| = \infty,$$

*where the expectation is taken over initialisation.*

---

[1]When proving statements about infinite networks $f_\theta(\boldsymbol{x})$, we refer to the limiting network. Eg. in this case the statement is equivalent to $\sup_{\boldsymbol{x} \in \mathbb{R}^{n_0}} |\mathbb{E}\left[\lim_{n \to \infty} f_\theta(\boldsymbol{x})\right]|$, where $n$ is the width of smallest layer.

*Proof sketch.* We show that such a network has an NTK which, for a large input to even one of its arguments, is approximately positive $n^L$-homogeneous, where $L$ is the number of layers in the network, and thus grows without bound when nontrivial. Furthermore, we show that this behaviour is strictly positive, and thus indeed nontrivial, for any pair of inputs with positive cosine similarity. Using the Representer theorem to express the prediction of the trained network as a weighted sum of NTK evaluations of the input with all training inputs, and by constructing the training dataset with nontrivial targets such that all pairs of training inputs have positive cosine similarity, we show that there exists a direction within the training set along which the magnitude of the trained network prediction must be nontrivial and therefore grow without bound with input magnitude.

$\square$

In the language of kernel methods, this is equivalent to showing that the reproducing kernel Hilbert space (RKHS) of the NTK, the span of which forms the set of all functions learnable by the network, contains only functions which grow without bound. For the training dataset described above, the function learned by the network does not enjoy perfect cancellation of these growths across training inputs, thus itself grows without bound. This result generalises existing theoretical results [34] (also built on NTK theory), and presents a clear problem: previously unseen inputs can generate pathologically large outputs, which could easily lead to a serious safety hazard in critical systems.

## 3.2 Inclusion of even a single Layer Norm bounds network predictions

By analysing the changes induced in the NTK, and therefore the predictions of the trained network, by the inclusion of an LN operation, we proceed to derive our main result:

**Theorem 3.2.** *The NTK of a network $f_\theta(\boldsymbol{x})$ with nonlinearities satisfying Assumption 3.1 and Assumption 3.2, fully-connected layers, and at least one Layer Norm anywhere in the network, enjoys the property that:*

$$\forall_{\boldsymbol{x} \in \mathbb{R}^{n_0}} \; |\Theta\left(\boldsymbol{x}, \boldsymbol{x}\right)| \leq C$$

*for some constant $C > 0$.*

*Proof sketch.* We prove that inclusion of a Layer Norm at layer $h_{LN}$ of the network causes the variance NTK, i.e. $\Theta(\boldsymbol{x}, \boldsymbol{x})$, to take the form of a ratio of two (strictly positive) functions of input norm, each at leading order a monomial of order $\|\boldsymbol{x}\|^{n^{h_{LN}}}$ independent of the direction of $\boldsymbol{x}$, thus for large input magnitudes approaches a finite limit, also independent of input direction. Furthermore, we show that by the assumptions on the nonlinearities, both functions remain bounded for finite input magnitude. Thus, the variance NTK itself is bounded. $\square$

Equipped with this result, we now proceed to study the extrapolatory behaviours of infinitely-wide networks with LN. We have previously proven that for standard infinitely-wide networks, one can always find a dataset causing arbitrarly large outputs. We now prove the opposite for network with LN—that is, for any dataset and any test point the output of networks with LN remains bounded with the bound being finite for any finite dataset.

**Theorem 3.3.** *Given an infinitely-wide network $f_\theta(\boldsymbol{x})$ satisfying Assumption 3.1 and Assumption 3.2, fully-connected layers, and at least one layer-norm anywhere in the network, trained until convergence on any training data $\mathcal{D}_{train} = (\mathcal{X}_{train}, \mathcal{Y}_{train})$, we have that for any $\boldsymbol{x} \in \mathbb{R}^{n_0}$:*

$$|\mathbb{E}[f_\theta(\boldsymbol{x})]| \leq B(\mathcal{D}_{train}) = \sqrt{\frac{C \max_{y \in \mathcal{Y}_{train}} |y|}{\lambda_{min}}} |D_{train}|,$$

*where the expectation is taken over initialisation and $\lambda_{min}$ is the smallest eigenvalue of $\Theta_{train}$ and $C$ is the kernel-dependent constant from Theorem C.2 or Theorem 3.2.*

*Proof sketch.* Note that the GP posterior mean at a single test point $\boldsymbol{x}$ is a product of two vectors $\Theta(\boldsymbol{x}, \mathcal{X}_{\text{train}})\Theta^{-1}(\mathcal{X}_{\text{train}}, \mathcal{X}_{\text{train}})$ and $y$. By Cauchy-Schwarz, the absolute value of a dot product of two vectors cannot be greater that the product of their norms. Clearly $|y| \leq \sqrt{|D_{\text{train}}| \max_{y \in \mathcal{Y}_{\text{train}}} |y|}$ and due to bounded variance property of the kernel we have $|\Theta(\boldsymbol{x}, \mathcal{X}_{\text{train}})| \leq \sqrt{|D_{\text{train}}|C}$, as explained in Proposition E.1. The proof is finished by observing that multiplication with $\Theta^{-1}(\mathcal{X}_{\text{train}}, \mathcal{X}_{\text{train}})$ cannot increase the norm by more than $\sqrt{\frac{1}{\lambda_{\min}}}$, which by Assumption 3.3 is finite. $\square$

Again in the language of kernel methods, the RKHS of the NTK, and therefore the set of functions learnable by such a network, contains only bounded functions. As the training targets are finite, by the self-regularising (with respect to the RKHS norm) property of kernel regressions, the trained network learns a finite combination of these functions, and therefore must itself be bounded. Note the bound is derived for the worst-case dataset and as such might not be tight in the average case. However, we would like to emphasise again that it is remarkable such a bound exists at all, in contrast to Theorem 3.1. This is a significant result and the first of its kind; by including even a single Layer Norm, one guarantees that the trained network's predictions remain bounded, providing a certificate of safety for critical tasks. We add here that, while the above only considers the case of fully connected networks with nonlinearities satisfying Assumption 3.2, in Appendix C we extend our analysis to a broader class of architectures and nonlinearities, providing this same guarantee for almost all popular activation functions and arbitrary architectures in the Tensor Program framework starting with a linear layer followed by a Layer Norm operation.

### 3.3 Is considering learning dynamics necessary?

Within the analysis above, we analysed the behaviours of randomly initialised neural networks trained with gradient descent until convergence via the NTK theory. Given the promising properties of LN networks, one might reasonably ask if it possible to arrive at such a bound without considering learning dynamics. However, we now show that in this relaxed setting, one may choose network parameters optimal with respect to the training loss such that even with a Layer Norm, no such guarantee is possible:

**Proposition 3.4.** *There exists a trained network $\tilde{f}_{\tilde{\theta}}(\boldsymbol{x})$ including Layer Norm layers such that*

$$\sup_{\boldsymbol{x} \in \mathbb{R}^{n_0}, \tilde{\theta} \in \tilde{\Omega}^\star} \left| \tilde{f}_{\tilde{\theta}}(\boldsymbol{x}) \right| = \infty,$$

*where $\tilde{\Omega}^\star = \arg\min_{\tilde{\theta} \in \tilde{\Theta}} L\left(\tilde{\theta}, \mathcal{D}_{train}\right)$ is the set of minimisers of the training loss.*

*Proof sketch.* There exists a network $f_\theta(\boldsymbol{x})$ with fully-connected layers, operating in an over-parametrised regime such that additional linear layer parameters may not further decrease the training loss over some finite training set $\mathcal{D}_{\text{train}}$, and $\sup_{\boldsymbol{x} \in \mathbb{R}^{n_0}, \theta \in \Omega^\star} |f_\theta(\boldsymbol{x})| = \infty$, where $\Omega^\star$ is the set of minimisers of the training loss (for example, a single linear layer network with non-trivial noiseless linear $\mathcal{D}_{\text{train}}$). One can construct a network $\tilde{f}_{\tilde{\theta}}(\boldsymbol{x})$ from $f_\theta(\boldsymbol{x})$ containing a standard Layer Norm operation, such that $\forall \boldsymbol{x} \in \mathbb{R}^{n_0}, \exists \tilde{\theta} \in \tilde{\Omega}$ s.t. $\tilde{f}_{\tilde{\theta}}(\boldsymbol{x}) = f_\theta(\boldsymbol{x}) \ \forall \theta \in \Omega$. As $f_\theta(\boldsymbol{x})$ already achieves the minimum training loss, it follows that the set of $\tilde{\theta}$ exactly recovering $f_\theta(\boldsymbol{x})$ is a subset of $\tilde{\Omega}^\star \ \forall \theta \in \Omega^\star$, so any properties of trained $f_\theta(\boldsymbol{x})$ are also present in this subset of trained $\tilde{f}_{\tilde{\theta}}(\boldsymbol{x})$. Thus, $\sup_{\boldsymbol{x} \in \mathbb{R}^{n_0}, \tilde{\theta} \in \tilde{\Omega}^\star} \left| \tilde{f}_{\tilde{\theta}}(\boldsymbol{x}) \right| = \infty$. $\qquad\square$

The above proves that, without explicitly considering training dynamics and initialisation, one cannot arrive at such guarantees. This lends credence to the popular theory that the training process itself is of particular importance to network generalisation [5, 26, 4, 32], and demonstrates the power of NTK theory in understanding the behaviour of Neural Networks.

## 4 Experiments

We now proceed to verify our theoretical results with empirical experiments, as well as show how the insights we derived can be utilised in practical problem settings. For details about compute and the exact hyperparameter settings, see Appendix J. We implemented all networks using PyTorch [27]. We open-source our codebase [2].

---

[2] `https://github.com/JuliuszZiomek/LN-NTK`

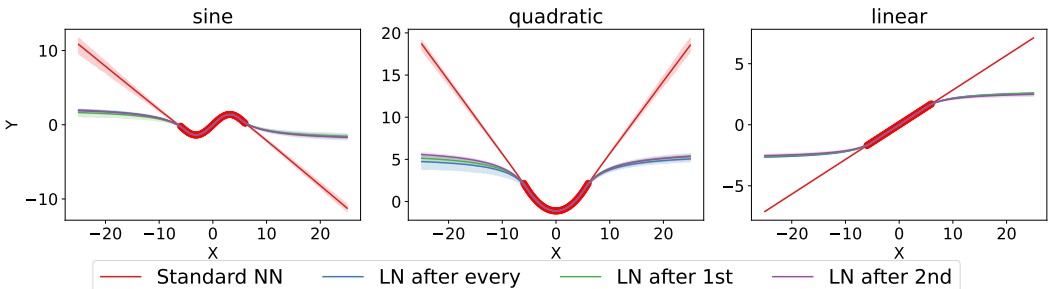

Figure 1: Predictions made by networks with various architectures when trained on synthetic datasets. Red dot show the train set datapoints. The solid lines indicate average values over 5 seeds and shaded areas are 95% confidence intervals of the mean estimator.

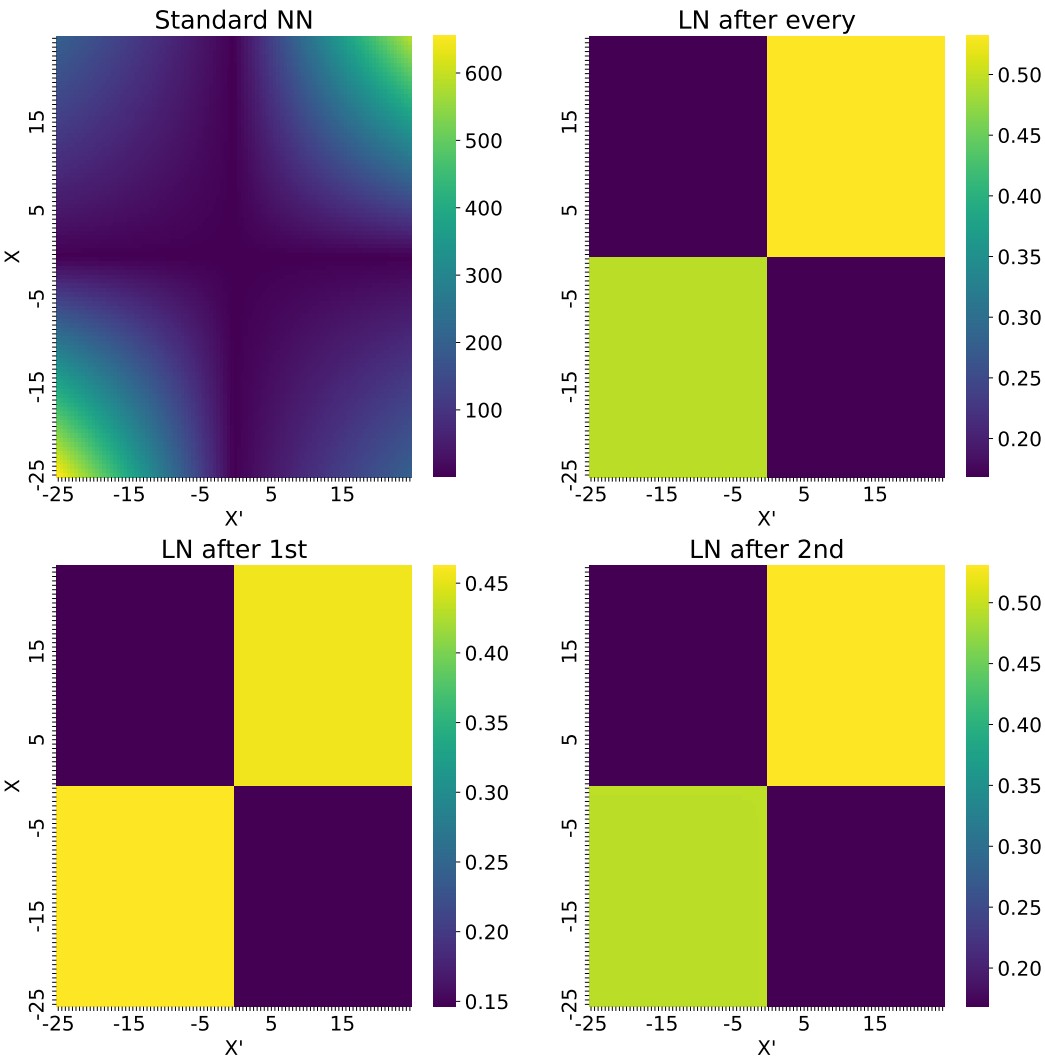

Figure 2: Heatmaps showing the values of empirical NTK values $\Theta(\boldsymbol{x}, \boldsymbol{x}')$ plotted on domain $\boldsymbol{x}, \boldsymbol{x}' \in [-25, 25]$ with brighter colours indicating higher values. Note that the scales are different for each heatplot, with the values range for the NTK of Standard NN being orders of magnitude higher than others. The displayed values are averages over 5 seeds.

## 4.1 Verifying Theoretical Results on Toy Problems

We first verify our theory on a number of toy problems. We specifically choose three one-dimensional datasets, where the relationship between the target and input variables are a sinusoid, a linear, and a quadratic function respectively For each dataset, we fit neural networks with different architectures. We first adopt a baseline standard MLP with ReLU nonlinearities, consisting of two layers, each of size 128. We then compare three different variants including Layer Normalisation: Preceding all, the first, and the last hidden layers. We train each architecture to convergence on each dataset and plot the results in Figure 1. To account for randomness in initialisation, we repeat this experiment over five seeds and plot the mean predictions together with 95% confidence interval of the mean estimator.

The results clearly show that regardless of dataset or initialisation, the network without an LN extrapolates linearly and thus "explodes", verifying Theorem 3.1, while the presence of even a single LN operation anywhere within the network prevents that behaviour from occurring, verifying Theorem 3.3. In fact, what we typically see is that a standard NN simply continues the local trend that appears near the boundaries of the training distribution, whereas the outputs of networks with at least one LN quickly saturate. Interestingly, varying the position of our number of LNs in the network has an insignificant effect on prediction. We show additional experiments in Figures 5 and 6 in Appendix. These results show that the same effect is observed when changing the activation function to GELU or SiLU, and that BatchNorm does not prevent the explosion as LayerNorm does, which is expected under our theory as BatchNorm does not alter the NTK. Additionally, in Figure 7, we show that LayerNorm also prevents explosion in a transformer architecture, thus verifying experimentally Theorem C.3.k

Our theory indicates the properties of the induced NTK should drastically change after the inclusion of even a single LN operation throughout the network. To verify this claim, we show the empirical NTK at initialisation, that is $\nabla f_\theta(\boldsymbol{x})^\top \nabla f_\theta(\boldsymbol{x}')$, as a heatplot. We note that as the network width approaches infinity, the empirical NTK should approach the limiting NTK, which we studied above. We show the heatplot in Figure 2. The displayed values are averages across all seeds.

By inspecting the plots in Figure 2, one immediately sees that there is a completely different structure to the empirical NTK between Standard NN and the architectures with LN, where the latter exhibit a "chessboard"-like pattern, clearly closely related to the cosine similarity of the inputs, which when low leads to minimal covariance. At the same time, this kernel value quickly saturates and reaches an upper bound that does not seem to vary with the magnitude of inputs, again as expected. On the other hand, the pattern displayed by the empirical NTK of a standard network is vastly different, with values steadily increasing with input's magnitude without saturation. This behaviour seems present even when the inputs have opposite signs, as indicated by the lighter corners of $(25, -25)$ and $(-25, 25)$. As such, we again see a form of explosion prevented by the presence of LN.

## 4.2 Extrapolating to bigger proteins

As the first real-world problem we study the UCI Physicochemical Properties of Protein Tertiary Structure dataset [28], which consists of 45,730 examples with nine physicochemical features extracted from amino acid sequences. The prediction target is the root mean square deviation (RMSD) of atomic distances in protein tertiary structures, reflecting the degree of structural similarity. As training set, we use randomly selected 90% of all proteins with surface area less than 20 thousand square angstroms. We then construct two validation sets, one in-domain with the remaining 10% of proteins with smaller surface area, and one out-of-domain with all proteins whose surface area is larger than 20 thousand square angstroms. We fit the models on the training set and evaluate them on both validation sets. We utilise the same network architectures as in the toy problem, that is, two hidden layers of size 128, with the following variants: no LN, LN preceding every hidden layer and LN preceding 1st and 2nd hidden layers only. Additionally, we compare with the well-known XGBoost [6] method. We show the numerical results in Table 1.

We observe that all methods perform similarly when evaluated in-distribution, but this is no longer the case under out-of-distribution (OOD) evaluation. Both the standard neural network and XGBoost perform substantially worse OOD and are outperformed by neural networks that include at least one layer normalisation. When inspecting the OOD coefficient of determination ($R^2$) across networks with different LN placements, we find that the position of the normalization layer does not appear to affect performance significantly—all LN-equipped networks yield mean OOD $R^2$ values within each

other's confidence intervals. To complement the $R^2$ analysis, Figure 3 presents the distribution of model predictions on OOD data for each architecture, as well as the relationship between predicted values and the total surface area of the proteins. The standard neural network without LN tends to predict unrealistically large residue sizes for large proteins, while XGBoost effectively extrapolates to a constant value. Biophysically, larger proteins often exhibit higher RMSD values because longer chains and more complex folds are harder to model accurately. However, this relationship is not strictly linear, as it depends on how the protein folds and the degree of structural compactness. Consequently, while one may expect residue size to increase with protein size, a simple linear extrapolation without bounds is unlikely to be accurate. Conversely, the constant extrapolation seen with XGBoost likely underestimates residue sizes, which should continue to grow beyond those observed in the training set. Neural networks that include at least one LN layer appear to strike a more appropriate balance—continuing the upward trend without producing overly extreme predictions. Thus, incorporating LN improves the network's extrapolatory behaviour, leading to more stable and realistic OOD predictions for this problem.

Table 1: In-distribution (ID) and out-of-distribution (OOD) coefficients of determinstation $R^2$ (higher the better) on the UCI Protein dataset. Displayed value is average over 10 seeds with 95% confidence intervals (rounded to 2 decimal places). The best values in each row and overlapping methods are highlighted in bold.

| Method | LN every | LN first | LN second | Standard NN | XGBoost |
|---|---|---|---|---|---|
| OOD $R^2$ | $\mathbf{0.50 \pm 0.02}$ | $\mathbf{0.50 \pm 0.02}$ | $\mathbf{0.50 \pm 0.01}$ | $0.31 \pm 0.02$ | $0.27 \pm 0.04$ |
| ID $R^2$ | $\mathbf{0.60 \pm 0.01}$ | $0.59 \pm 0.01$ | $\mathbf{0.60 \pm 0.01}$ | $0.58 \pm 0.01$ | $\mathbf{0.61 \pm 0.01}$ |

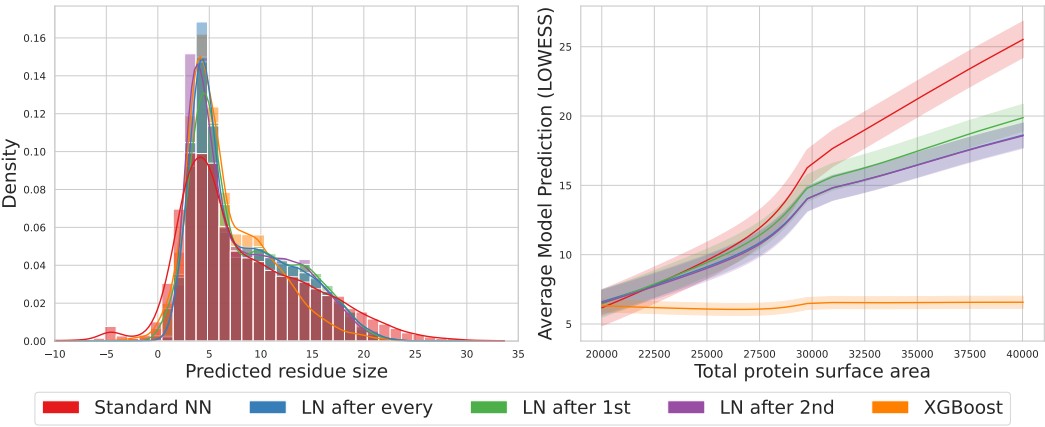

Figure 3: Histogram of predictions made by each model on the out-of-distribution data for protein experiment (left) and LOWESS [7] trendlines fitted to the relationship between protein surface area and average prediction for each method (right). Both plots are produced from data aggregated from 10 seeds. Shaded areas in the right plot are 95%-confidence intervals.

### 4.3 Predicting Age from Facial Images of Underrepresented Ethnicities

As a second real-world experiment, we target a computer vision application, where the target is to predict a person's age from their facial image. We utilise the UTKFace dataset[3], a large-scale facial image dataset annotated with age, gender, and ethnicity labels. The dataset contains over 20,000 facial images spanning a wide range of ages from 0 to 116 years, captured in unconstrained conditions. Each image is cropped and aligned to 200×200 pixels, and age labels are provided as discrete integers. The ethnicity label has five unique values: White, Black, Asian, Indian and Others. To evaluate extrapolation capabilities, we construct the training set using only samples with ethnicity labels White, Black, Asian, or Indian. During evaluation, the in-distribution set consists of validation images from these same four ethnicities, while the out-of-distribution set includes remaining validation set images.

---

[3]https://susanqq.github.io/UTKFace/

We train a number of different networks, each one starting with a frozen ResNet-18 [17] as a feature extractor. After that, each architecture includes two hidden, fully-connected layers of size 128 with ReLU non-linearities. Same as in the previous two experiments, we experiment with presence and position of LNs. We train one network without LN, one network with LN after every fully-connected layer and with only one LN after 1st and 2nd fully-connected layers respectively. We train each network until convergence on training set and then evaluate on in and out-of-distribution evaluation sets. We show the results in Table 2.

We observe that while all architectures perform equally well on in-distribution data, their performances vary significantly out-of-distribution. The architecture performing best OOD contains an LN preceding only the first hidden layer, but its confidence intervals overlap with that of architecture with an LN preceding every hidden layer. Having an LN preceding only the second hidden layer performs worse, but still clearly outperforms having no LN in the network. In Figure 4, we show average model error for a given OOD test input as a function of average cosine similarity of ResNet features of that input to training samples. We see that as similiarity goes to 0.0, meaning the testing input becomes dissimilar to the training data, the average error of all models grows, but this grow is fastest for the model with no LN, indicating that models with LN make more accurate extrapolation, implying the stable extrapolation property can also be useful in high-dimensional feature spaces.

Table 2: In-distribution (ID) and out-of-distribution (OOD) coefficients of determinstation $R^2$ (higher the better) on the UTKFace age prediction problem. Displayed value is average over 250 seeds with 95% confidence intervals (rounded to 2 decimal places) . The best values in each row and overlapping methods are highlighted in bold.

| Method | LN every | LN first | LN second | Standard NN |
|---|---|---|---|---|
| OOD $R^2$ | $\mathbf{0.42 \pm 0.05}$ | $\mathbf{0.47 \pm 0.03}$ | $0.36 \pm 0.04$ | $0.27 \pm 0.03$ |
| ID $R^2$ | $\mathbf{0.66 \pm 0.01}$ | $\mathbf{0.68 \pm 0.01}$ | $\mathbf{0.66 \pm 0.01}$ | $\mathbf{0.66 \pm 0.01}$ |

## 5    Related Work

Theoretical research in this area is limited, as is typically the case for neural network theory. Xu et al. [34] also utilise NTK theory to analyse extrapolatory behaviour for a range of infinitely-wide architectures, but for our problem setting of fully connected networks they show only that networks with ReLU activations extrapolate linearly; a special case of Corollary E.1 used in the proof of Theorem 3.1. Hay et al. [14] provide extrapolation bounds when learning functions on manifolds, but this requires a specific training method and prior knowledge of the function, rather than a general property of networks as we prove. Courtois et al. [9] discuss the general implications of the existence of NTK theory on network extrapolation, but contribute no bounds or quantitative results.

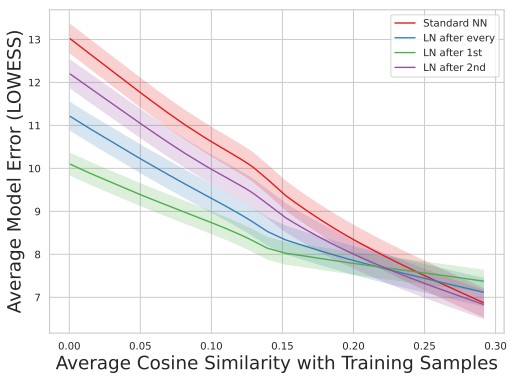

Figure 4: LOWESS [7] trendlines fitted to average OOD model prediction on the UTK-Face task as a function of average cosine similarity of ResNet-18 features with training samples. Shaded areas are 95% confidence intervals produced over 250 seeds.

Empirical research is more abundant; Kang et al. [20] empirically observe that in very high-dimensional input spaces trained networks tend to extrapolate at the optimal constant solution, that is, the constant solution which minimises error on the training dataset, including with Layer Norm. This relates to our Proposition E.2 and the proof of Corollary E.2, which suggest that for a network with ReLU activation function and a very high number of dimensions, the NTK is approximately constant for relatively low input magnitudes, regions of which at these dimensions may still have the majority of their volume described as out-of-distribution, and therefore extrapolatory. Additionally, in very high dimensions the criterion on

the dataset to guarantee prediction instability occurs with very low probability, with most training input pairs having cosine similarity close to 0. Vita et al. [31] show that extrapolatory behaviour is affected by the choice of optimiser for training. Fahlberg et al. [11] investigate network extrapolation in protein engineering, finding inductive biases of networks drive extrapolatory behaviour, which is complimentary to our results.

Regarding work understanding Layer Norm in general, Xu et al. [33] found the effect of the Layer Norm operation on gradients during backpropagation to be more important than its effect on the forward pass, which matches our observations as these gradients are the mechanism by which Theorem 3.2 is possible. Zhu et al. [38] empirically found replacing Layer Norm operations with a scaled tanh function to be effective in large transformers; we speculate that this success comes about from enforcing the sigmoidal behaviour of the infinite width Layer Norm (obvious from our Equation 10) even at lower network widths.

# 6   Conclusions

Within this work, we have shown how the inclusion of even a single LN operation throughout the network fundamentally changes its extrapolatory behaviour. We explained this phenomenon from the NTK point of view and verified our claims with a number of empirical studies, showing how this property may be useful in real-world scenarios. We believe the conclusions drawn from our work can be useful to practitioners. While LNs are already very popular, there are still many architectures that do not include it. However, if one wishes for the output to be bounded even outside of the training domain, one should consider utilising (at least one) LN somewhere within the network's architecture.

We note that extrapolation is, in general, an incredibly difficult problem. As we go outside of data distribution, what exactly consistutes a "good" extrapolation is unknown to us unless further assumptions can be made. A stable (i.e. bounded) extrapolation, while desirable in many cases outlined in this paper, does not need to be always correct. For example, there might be some datasets when we want to continue a linear trend even as we go far away from training data. This pertains to the broader problem of selecting a kernel based on limited observations, such that the function of interest lies in the RKHS, which has been extensively studied in GP literature [24, 3, 39, 40]. Within this work, we do not claim that LN universally "solves" the problem of extrapolation. Instead, we aimed to highlight the differences in extrapolatory behaviour of networks with and without LN, so that practitioners can make more informed architectural choices.

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

## A  Parametrisation and Initialisation

Throughout this paper, we use the following parametrisations and, where applicable, initialisations for network components:

### A.1  Linear Layers

Denoting a general layer input as $\boldsymbol{\xi}^{(i-1)} \in \mathbb{R}^{n_{i-1}}$ and output as $\boldsymbol{z}^{(i)} \in \mathbb{R}^{n_i}$, we parametrise a linear layer as

$$\boldsymbol{z}^{(i)} = \frac{1}{\sqrt{n_i}} \boldsymbol{W}^{(i)} \boldsymbol{\xi}^{(i-1)} + \sigma_b \boldsymbol{b}^{(i)}, \tag{1}$$

with parameters

$$\boldsymbol{W}^{(i)} \in \mathbb{R}^{n_i \times n_{i-1}},$$
$$\boldsymbol{b}^{(i)} \in \mathbb{R}^{n_i},$$

and hyperparameter

$$\sigma_b > 0.$$

Note that $\sigma_b \neq 0$ is necessary to ensure that the kernel is non-degenerate, which is a requirement of Assumption 3.3. Furthermore, we initialise these parameters as

$$W_j^{(i)} \in \mathbb{R}^{n_i} \sim \mathcal{N}\left(\mathbf{0}, \mathcal{I}\right) \qquad\qquad \forall j = 1, \dots, n_{i-1}, \qquad (2)$$

$$\boldsymbol{b}^{(i)} \sim \mathcal{N}\left(\mathbf{0}, \mathcal{I}\right), \qquad (3)$$

where the row vector $W_j^{(i)}$ is the $j$th row of $\boldsymbol{W}^{(i)}$, such that

$$\boldsymbol{W}^{(i)} = \left[ W_1^{(i)\top} \dots W^{(n_i)\top} \right]^\top \qquad (4)$$

## A.2 Layer Norm

We consider Layer Norm to operate without additional scaling parameters. Thus, denoting a general layer input as $\boldsymbol{z}^{(i)} \in \mathbb{R}^{n_i}$ and output as $\tilde{\boldsymbol{z}}^{(i)} \in \mathbb{R}^{n_i}$,

$$\tilde{\boldsymbol{z}}_j^{(i)} = \frac{\boldsymbol{z}_j^{(i)} - \mu}{\sigma} \qquad\qquad \forall j = 1, \dots, n_i, \qquad (5)$$

$$\mu = \frac{1}{n_i} \sum_{n=j}^{n_i} \boldsymbol{z}_j^{(i)}, \qquad (6)$$

$$\sigma = \sqrt{\frac{1}{n_i} \sum_{n=j}^{n_i} \left( \boldsymbol{z}_j^{(i)} - \mu \right)^2}. \qquad (7)$$

## A.3 Activation Functions

We consider activation functions, or nonlinearities, $\phi : \mathbb{R} \to \mathbb{R}$, to act elementwise. Thus, denoting a general layer input as $\boldsymbol{z}^{(i)} \in \mathbb{R}^{n_i}$ and output as $\boldsymbol{\xi}^{(i)} \in \mathbb{R}^{n_i}$,

$$\boldsymbol{\xi}_j^{(i)} = \sqrt{c_\phi}\, \phi\left( \boldsymbol{z}_j^{(i)} \right) \qquad\qquad \forall j = 1, \dots, n_i \qquad (8)$$

where $c_\phi$ is the activation function-dependant constant defined in Equation 19.

# B  Limit for $\|\boldsymbol{x}^\star\|_2 > 1$

While Theorem 2.1 of [23] assumes $\|\boldsymbol{x}\|_2 < 1$, we note this is only neccessary for establishing the inequality labelled as S85, which states that $n^{-1/2} \|\boldsymbol{x}^l\|_2 \le K_1$, where $x^l$ is the post-activation of $l$th layer with $\boldsymbol{x}^0$ being network input, $n$ is the size of smallest layer and $K_1$ is some constant. For any finite $n$ the set of inputs for which $n^{-1/2} \|\boldsymbol{x}^l\|_2 \le K_1$ is some subset of the entire input space $\mathbb{R}^n$. However, for any finite input $x$, all post-activation are also finite and thus taking $n > \max_{l=0,\dots,L} \frac{\|\boldsymbol{x}^l\|_2^2}{K_1^2}$ sufficies. As such, for an arbitrary input $x \in R^{n_0}$ we can always find sufficiently large network and even for points $\|\boldsymbol{x}^\star\|_2 > 1$ the limiting result holds.

# C  Extended Results

## C.1 The soft-cosine similarity

**Definition C.1.** The soft-cosine similarity between two vectors $\boldsymbol{x}, \boldsymbol{x}' \in \mathbb{R}^n$, parametrised by $\sigma^2 \ge 0$, is given by

$$\mathcal{S}\left(\boldsymbol{x}, \boldsymbol{x}'; \sigma^2\right) = \frac{\frac{1}{n}\boldsymbol{x}^\top \boldsymbol{x}' + \sigma^2}{\sqrt{(\frac{1}{n}\boldsymbol{x}^\top \boldsymbol{x} + \sigma^2)(\frac{1}{n}\boldsymbol{x}'^\top \boldsymbol{x}' + \sigma^2)}}.$$

*Remark* C.1. For $\sigma^2 = 0$, the soft-cosine similarity between two vectors $\boldsymbol{x}, \boldsymbol{x}' \in \mathbb{R}^n$, $\mathcal{S}\left(\boldsymbol{x}, \boldsymbol{x}'; \sigma^2\right)$, reduces to the standard cosine similarity between $\boldsymbol{x}$ and $\boldsymbol{x}'$, $\cos\left(\angle\left(\boldsymbol{x}, \boldsymbol{x}'\right)\right)$.

*Proof.* $\mathcal{S}\left(\boldsymbol{x}, \boldsymbol{x}'; 0\right) = \frac{\frac{1}{n}\boldsymbol{x}^\top \boldsymbol{x}'}{\sqrt{\frac{1}{n}\boldsymbol{x}^\top \boldsymbol{x}}\sqrt{\frac{1}{n}\boldsymbol{x}'^\top \boldsymbol{x}'}} = \frac{\boldsymbol{x}^\top \boldsymbol{x}'}{\|\boldsymbol{x}\|\|\boldsymbol{x}'\|} = \cos\left(\angle\left(\boldsymbol{x}, \boldsymbol{x}'\right)\right)$ by definition. $\qquad\square$

*Remark* C.2. The soft-cosine similarity between two vectors $\boldsymbol{x}, \boldsymbol{x}' \in \mathbb{R}^n$, $\mathcal{S}\left(\boldsymbol{x}, \boldsymbol{x}'; \sigma^2\right)$, is bounded to the set $[-1, 1]$, and $\forall \sigma^2 > 0$, achieves the maximum value of 1 only for $\boldsymbol{x} = \boldsymbol{x}'$.

*Proof.* $\mathcal{S}\left(\boldsymbol{x}, \boldsymbol{x}'; \sigma^2\right)$ can be rewritten as the cosine similarity between the two augmented vectors $\tilde{\boldsymbol{x}} = \left(\frac{1}{\sqrt{n}}\boldsymbol{x}, \sigma_b\right)$ and $\tilde{\boldsymbol{x}}' = \left(\frac{1}{\sqrt{n}}\boldsymbol{x}', \sigma_b\right) \in \mathbb{R}^{n+1}$, so is trivially bounded to the set $[-1, 1]$. The cosine similarity achieves the maximum value of 1 only for $\tilde{\boldsymbol{x}} = \lambda\tilde{\boldsymbol{x}}'$ for some $\lambda > 0$. When $\sigma^2 > 0$, the matching last element $\sigma^2$ constrains $\lambda = 1$, and thus $\boldsymbol{x} = \boldsymbol{x}'$. Note that for the same reason, when $\sigma^2 > 0$ the minimum value of $-1$ is unattainable. $\qquad\square$

## C.2 Asymptotically Positive $n$-homogeneous Functions

**Definition C.2.** A function $\phi(\cdot) : \mathbb{R} \to \mathbb{R}$ is asymptotically positive $n$-homogeneous if

$$\lim_{\lambda \to \infty} \frac{\phi(\lambda x)}{\lambda^n} = \hat{\phi}(x) \quad \forall x \in \mathbb{R}, \lambda > 0,$$

where $\hat{\phi}(\cdot) : \mathbb{R} \to \mathbb{R}$ is positive $n$-homogeneous, that is, $\hat{\phi}(\lambda x) = \lambda^n \hat{\phi}(x) \quad \forall x \in \mathbb{R}, \lambda > 0$. Furthermore, we assume throughout that $\hat{\phi}(\cdot)$ is nontrivial.

**Corollary C.1.** *All positive $n$-homogeneous functions $\phi : \mathbb{R}^d \to \mathbb{R}$ are also asymptotically positive $n$-homogeneous.*

*Proof.*

$$\lim_{\lambda \to \infty} \frac{\phi(\lambda \boldsymbol{x})}{\lambda^n} = \lim_{\lambda \to \infty} \frac{\lambda^n \phi(\boldsymbol{x})}{\lambda^n}$$
$$= \phi(\boldsymbol{x}).$$

By assumption of the theorem, $\phi(\boldsymbol{x})$ is positive $n$-homogeneous, thus $\phi(\boldsymbol{x})$ is asymptotically positive $n$-homogeneous by definition. $\qquad\square$

Note that many popular activation functions [10] are asymptotically positive $n$-homogeneous, which we explicitly show for a small selection:

*Remark* C.3. The ReLU activation function [12] $\phi_{\text{ReLU}}(x) = \max(0, x)$, and its Leaky [25] $\phi_{\text{LReLU}}(x) = \max(0.01x, x)$ and parametric [15] $\phi_{\text{PReLU}}(x) = \max(\alpha x, x), \alpha \in [0, 1)$ variants are asymptotically positive 1-homogeneous (and indeed positive 1-homogeneous).

*Proof.* To begin, note that both ReLU and Leaky ReLU are special cases of Parametric ReLU, with $\phi_{\text{ReLU}}(x) = \phi_{\text{PReLU}}(x, 0)$ and $\phi_{\text{LReLU}}(x) = \phi_{\text{PReLU}}(x, 0.01)$, thus it suffices to show that these properties hold for Parametric ReLU. We proceed by showing that Parametric ReLU is positive $n$-homogeneous:

$$\begin{aligned}
\phi_{\text{PReLU}}(\lambda x; \alpha) &= \phi_{\text{PReLU}}(\lambda x; \alpha) \\
&= \lambda \max(\alpha x, x) \\
&= \lambda^1 \phi_{\text{PReLU}}(x; \alpha) \qquad\qquad \forall x \in \mathbb{R}, \lambda > 0,
\end{aligned}$$

which shows positive 1-homogeneity by definition. Thus due to Corollary C.1, ReLU, Leaky ReLU, and Parametric ReLU are all asymptotically positive 1-homogeneous. $\qquad\square$

*Remark* C.4. The GELU and Swish (or SiLU) activation functions [18], given by $\phi_{\text{GELU}}(x) = x\Phi(x)$, where $\Phi(x) = \int_{\infty}^{x} \frac{1}{\sqrt{2\pi}} \exp\left(-\frac{t^2}{2}\right) dt$ is the standard normal cumulative distribution function and $\phi_{\text{Swish}}(x; \beta) = x\sigma(\beta x), \beta \geq 0$, where $\sigma(x) = \frac{1}{1-\exp(-x)}$ is the sigmoid function respectively, are asymptotically positive 1-homogeneous.

*Proof.* To begin, note that both GELU and Swish/SiLU take the form $\phi(x) = x\omega(x)$ with

$$\lim_{\lambda \to \infty} \omega(\lambda x) = \begin{cases} 1 & x > 0 \\ 0 & x < 0 \end{cases}, \quad \text{and}$$

$$\lim_{x \to -\infty} x\omega(x) = 0$$

where $\omega(x) = \Phi(x)$ for GELU and $\omega(x) = \sigma(\beta x) = \frac{1}{1+\exp(-\beta x)}$ for Swish/SiLU. It then follows that

$$\begin{aligned}
\lim_{\lambda \to \infty} \frac{\phi(\lambda x)}{\lambda^1} &= \lim_{\lambda \to \infty} \frac{\lambda x \omega(\lambda x)}{\lambda} \\
&= \lim_{\lambda \to \infty} x\omega(\lambda x) \\
&= x \lim_{\lambda \to \infty} \omega(\lambda x) \\
&= \begin{cases} x & x \geq 0 \\ 0 & x < 0 \end{cases} \\
&= \max(0, x) \\
&= \phi_{\text{ReLU}}(x) \quad \forall x \in \mathbb{R},
\end{aligned}$$

where the second transition follows from the dominance of $\omega(x)$ over $x$ in the negative limit and the boundedness of $\omega(x)$.

From Remark C.3, we see that $\phi_{\text{ReLU}}(x)$ is positive 1-homogeneous. Hence by definition, both GELU and Swish/SiLU are asymptotically positive 1-homogeneous.

$\square$

*Remark* C.5. The tanh $\phi_{\text{tanh}}(x) = \tanh(x)$ and sigmoid $\phi_{\text{sigmoid}}(x) = \sigma(x) = \frac{1}{1-\exp(-x)}$ activation functions [30] are asymptotically positive 0-homogeneous.

*Proof.* Noting that $\phi_{\text{sigmoid}}(x) = \frac{1}{2}\left(\phi_{\text{tanh}}(x)\left(\frac{x}{2}\right) + 1\right)$, it suffices to show this for $\phi_{\text{tanh}}(x)$ alone:

$$\begin{aligned}
\lim_{\lambda \to \infty} \frac{\phi_{\text{tanh}}(\lambda x)}{\lambda^0} &= \lim_{\lambda \to \infty} \tanh(\lambda x) \\
&= \begin{cases} 1 & x > 0 \\ 0 & x = 0 \\ -1 & x < 1 \end{cases} \\
&= \hat{\phi}_{\text{tanh}}(x). \\
\hat{\phi}_{\text{tanh}}(\lambda x) &= \begin{cases} 1 & \lambda x > 0 \\ 0 & \lambda x = 0 \\ -1 & \lambda x < 1 \end{cases} \\
&= \begin{cases} 1 \cdot \lambda^0 & x > 0 \\ 0 \cdot \lambda^0 & x = 0 \\ -1 \cdot \lambda^0 & x < 1 \end{cases} \\
&= \lambda^0 \hat{\phi}_{\text{tanh}}(x) \quad \forall x \in \mathbb{R}, \lambda > 0,
\end{aligned}$$

thus $\hat{\phi}_{\text{tanh}}(\cdot)$ is positive 0-homogeneous and therefore both $\phi_{\text{tanh}}(\cdot)$ and $\phi_{\text{sigmoid}}(\cdot)$ are asymptotically positive 0-homogeneous by definition.

$\square$

Note that beyond what we explicitly show here, many other popular activation functions are also asymptotically positive $n$-homogeneous.

**Assumption C.1.** Activation functions $\phi : \mathbb{R} \to \mathbb{R}$ are asymptotically positive $n$-homogeneous with $n > 0.5$, such that by Lemma E.1 the derivatives $\dot{\phi}(\cdot)$ are asymptotically positive $n-1$-homogeneous with, and both $\phi(\cdot), \dot{\phi}(\cdot) \in L^2(\mathbb{R}, \gamma)$, i.e. they are square-integrable with respect to the standard normal measure $\gamma$.

Note that Assumption C.1 directly implies Assumption 3.2, but the same cannot be said about the inverse of this.

### C.3 Extended worst-case bound on Layer Norm Networks

**Theorem C.1.** *The NTK of an infinitely-wide network $f_\theta(\boldsymbol{x})$ starting with a linear layer follows by a Layer Norm operation, depends only on the soft-cosine similarity between the two inputs $\boldsymbol{x}$ and $\boldsymbol{x}'$ parametrised by $\sigma_b^2$, i.e.*

$$\forall_{\boldsymbol{x}, \boldsymbol{x}' \in \mathbb{R}^{n_0} \times \mathbb{R}^{n_0}} \Theta\left(\boldsymbol{x}, \boldsymbol{x}'\right) = \tilde{\Theta}\left(\mathcal{S}\left(\boldsymbol{x}, \boldsymbol{x}'; \sigma_b^2\right)\right).$$

**Corollary C.2.** *The NTK of an infinitely-wide network starting with linear layer followed by layer-norm enjoys the property that:*

$$\forall_{\boldsymbol{x} \in \mathbb{R}^{n_0}} \Theta(\boldsymbol{x}, \boldsymbol{x}) = C$$

*for some constant $C > 0$.*

*Proof.* Due to Theorem C.1, $\Theta(\boldsymbol{x}, \boldsymbol{x}) = \tilde{\Theta}\left(\mathcal{S}\left(\boldsymbol{x}, \boldsymbol{x}; \sigma_b^2\right)\right)$. However, due to Remarks C.1 and C.2, $\mathcal{S}(\boldsymbol{x}, \boldsymbol{x}; \sigma_b^2) = 1 \ \forall \boldsymbol{x} \in \mathbb{R}^{n_0}, \sigma^2 \geq 0$. Hence, $\Theta(\boldsymbol{x}, \boldsymbol{x}) = \tilde{\Theta}(1) = C \ \forall \boldsymbol{x} \in \mathbb{R}^{n_0}$, where $C$ is a positive constant depending only on the architecture, $\sigma_b$, and the initialisation of the rest of the network. $\square$

**Theorem C.2.** *The NTK of a network $f_\theta(\boldsymbol{x})$ with nonlinearities satisfying Assumption 3.1 and Assumption C.1, fully-connected layers, and at least one layer-norm anywhere in the network, enjoys the property that:*

$$\forall_{\boldsymbol{x} \in \mathbb{R}^{n_0}} \left|\Theta\left(\boldsymbol{x}, \boldsymbol{x}\right)\right| \leq C$$

*for some constant $C > 0$.*

**Theorem C.3.** *Given an infinitely-wide network $f_\theta(\boldsymbol{x})$ starting with a linear layer followed by a Layer Norm operation, or with asymptotically positive $n$-homogeneous, nonlinearities satisfying Assumption 3.1 bar the condition of positive $n$-homogeneity, fully-connected layers, and at least one layer-norm anywhere in the network, trained until convergence on any training data $\mathcal{D}_{train}$, we have that for any $\boldsymbol{x} \in \mathbb{R}^{n_0}$:*

$$\mathbb{E}[|f_\theta(\boldsymbol{x})|] \leq B(\mathcal{D}_{train}) = \sqrt{\frac{C \max_{y \in \mathcal{Y}_{train}} |y|}{\lambda_{min}}} |D_{train}|,$$

*where the expectation is taken over initialisation and $\lambda_{min}$ is the smallest eigenvalue of $\Theta_{train}$ and $C$ is the kernel-dependent constant from Theorem C.2.*

## D Proof of Theorem C.1

**Theorem C.1.** *The NTK of an infinitely-wide network $f_\theta(\boldsymbol{x})$ starting with a linear layer follows by a Layer Norm operation, depends only on the soft-cosine similarity between the two inputs $\boldsymbol{x}$ and $\boldsymbol{x}'$ parametrised by $\sigma_b^2$, i.e.*

$$\forall_{\boldsymbol{x}, \boldsymbol{x}' \in \mathbb{R}^{n_0} \times \mathbb{R}^{n_0}} \Theta\left(\boldsymbol{x}, \boldsymbol{x}'\right) = \tilde{\Theta}\left(\mathcal{S}\left(\boldsymbol{x}, \boldsymbol{x}'; \sigma_b^2\right)\right).$$

*Proof.* Let us denote the neural network as $f_\theta = (o_1, o_2, \ldots, o_L)$ to be a collection of operations $o_1, o_2, \ldots, o_L$ parametrised by some $\theta \in \Omega = \{\theta_1, \cdots \theta_L\}$ such that:

$$f_\theta(\boldsymbol{x}) = o_L(\ldots(o_1(\boldsymbol{x}))).$$

Let then, $o_1$ be a linear layer and $o_2$ be the Layer Norm operator, as described by Parametrisation 1 and Parametrisation 5 respectively. As such $\theta_1 = (\boldsymbol{W}^{(1)} \in \mathbb{R}^{n_1 \times n_0}, \boldsymbol{b}^{(1)} \in \mathbb{R}^{n_1})$ and $\theta_2 = \emptyset$. Notice that we can always represent the output of this neural network as $\tilde{f}_{\tilde{\theta}}(\tilde{\boldsymbol{x}})$, where $\tilde{f} = (o_3, \ldots, o_L)$ and $\tilde{\boldsymbol{x}} = o_2(o_1(\boldsymbol{x}))$ Let us denote by $\Theta(\boldsymbol{x}, \boldsymbol{x})$ the NTK of neural network $f_\theta$ evaluated at points $\boldsymbol{x}$ and $\boldsymbol{x}'$. Using the decomposition in Equation 10 of [36], we get that:

$$\Theta(\boldsymbol{x}, \boldsymbol{x}') = \sum_{l=1}^{L} \langle \nabla_{\theta_l} f_\theta(\boldsymbol{x}), \nabla_{\theta_l} f_\theta(\boldsymbol{x}') \rangle$$

$$= \langle \nabla_{\theta_1} f_\theta(\boldsymbol{x}), \nabla_{\theta_1} f_\theta(\boldsymbol{x}') \rangle + \sum_{l=3}^{L} \langle \nabla_{\theta_l} f_\theta(\boldsymbol{x}), \nabla_{\theta_l} f_\theta(\boldsymbol{x}') \rangle$$

$$= \langle \nabla_{\theta_1} f_\theta(\boldsymbol{x}), \nabla_{\theta_1} f_\theta(\boldsymbol{x}') \rangle + \sum_{l=3}^{L} \langle \nabla_{\theta_l} \tilde{f}_{\tilde{\theta}}(\boldsymbol{x}), \nabla_{\theta_l} \tilde{f}_{\tilde{\theta}}(\boldsymbol{x}') \rangle$$

$$= \langle \nabla_{\theta_1} f_\theta(\boldsymbol{x}), \nabla_{\theta_1} f_\theta(\boldsymbol{x}') \rangle + \tilde{\Theta}(\tilde{\boldsymbol{x}}, \tilde{\boldsymbol{x}}'),$$

where the first-to-second line transition it true because the layer-norm operation has no parameters and second-to-third is true because by construction $f_\theta(\boldsymbol{x}) = \tilde{f}_{\tilde{\theta}}(\boldsymbol{x})$. Let us denote $\boldsymbol{z}^{(1)} = o_1(\boldsymbol{x}) = \frac{1}{\sqrt{n_0}} \boldsymbol{W}^{(1)} \boldsymbol{x} + \sigma_b \boldsymbol{b}^{(1)}$. Due to Initialisation 2 and Initialisation 3, each $\boldsymbol{z}_j^{(1)}$ (for $j = 1, \ldots, n_1$) is an iid Gaussian variable with moments:

$$\mathbb{E}\left[\boldsymbol{z}_j^{(1)}\right] = \mathbb{E}\left[\frac{1}{\sqrt{n_0}} \boldsymbol{W}_j^{(1)} \boldsymbol{x} + \sigma_b \boldsymbol{b}_j^{(1)}\right]$$

$$= \frac{1}{\sqrt{n_0}} \mathbb{E}\left[\boldsymbol{W}_j^{(1)}\right] \boldsymbol{x} + \mathbb{E}\left[\boldsymbol{b}_j^{(1)}\right]$$

$$= 0 \qquad\qquad \forall j = 1, \ldots, n_1, \qquad (9)$$

$$\mathrm{Var}\left(\boldsymbol{z}_j^{(1)}\right) = \mathrm{Var}\left(\frac{1}{\sqrt{n_0}} \boldsymbol{W}_j^{(1)} \boldsymbol{x} + \sigma_b \boldsymbol{b}_j^{(1)}\right)$$

$$= \frac{1}{n_0} \boldsymbol{x}^\top \mathrm{Var}\left(\boldsymbol{W}_j^{(1)}\right) \boldsymbol{x} + \sigma_b^2 \mathrm{Var}\left(\boldsymbol{b}_j^{(1)}\right)$$

$$= \frac{1}{n_0} \boldsymbol{x}^\top \boldsymbol{x} + \sigma_b^2 \qquad\qquad \forall j = 1, \ldots, n_1. \qquad (10)$$

We can similarly denote $\boldsymbol{z}^{(1)'} = o_1(\boldsymbol{x}')$, which equivalently has elements iid with moments given as above. Each pair $\boldsymbol{z}_i^{(1)}$ and $\boldsymbol{z}_j^{(1)'}$ (for $i, j = 1, \ldots, n_1$) are Gaussian distributed and are linearly dependent through the random variables $\boldsymbol{W}^{(1)}$ and $\boldsymbol{b}^{(1)}$, so are joint-Gaussian distributed with covariance given by:

$$\mathrm{Cov}\left(\boldsymbol{z}_i^{(1)}, \boldsymbol{z}_j^{(1)'}\right) = \mathbb{E}\left[\boldsymbol{z}_i^{(1)} \boldsymbol{z}_j^{(1)'}\right]$$

$$= \mathbb{E}\left[\left(\frac{1}{\sqrt{n_0}} \boldsymbol{W}_i^{(1)} \boldsymbol{x} + \sigma_b^{(1)} \boldsymbol{b}_i\right)\left(\frac{1}{\sqrt{n_0}} \boldsymbol{W}_j^{(1)} \boldsymbol{x}' + \sigma_b \boldsymbol{b}_j^{(1)}\right)\right]$$

$$= \frac{1}{n_0} \boldsymbol{x}^\top \mathbb{E}\left[\boldsymbol{W}_i^{(1)\top} \boldsymbol{W}_j^{(1)}\right] \boldsymbol{x}' + \sigma_b^2 \mathbb{E}\left[\boldsymbol{b}_i^{(1)} \boldsymbol{b}_j^{(1)}\right]$$

$$= \frac{1}{n_0} \boldsymbol{x}^\top \mathcal{I} \boldsymbol{x}' \delta_{ij} + \sigma_b^2 \delta_{ij}$$

$$= \left(\frac{1}{n_0} \boldsymbol{x}^\top \boldsymbol{x}' + \sigma_b^2\right) \delta_{ij},$$

where the first line follows from the fact that both $z_i^{(1)}$, and $z_j^{(1)'}$ are zero-mean scalars, the third line follows from the independence of $b^{(1)}$ and $W^{(1)}$ and the fact that $W_i^{(1)} x = x^\top W_i^{(1)^\top}$, and $\delta_{ij}$ denotes the Kronecker delta.

We now consider the Layer Norm operation as described in Parametrisation 5 providing

$$\tilde{x}_j = o_2(z^{(1)})_j = \frac{z_j^{(1)} - \mu}{\sigma} \tag{11}$$

where $\mu$ and $\sigma$ as defined in Equation 6 and Equation 7 respectively are the maximum likelihood estimators for $\mathbb{E}\left[z_j^{(1)}\right]$ and $\sqrt{\text{Var}\left(z_j^{(1)}\right)}$ respectively.

Due to asymptotic consistency of MLE, we have that with probability 1 $\mu \to \mathbb{E}\left[z_j^{(1)}\right]$ and $\sigma \to \sqrt{\text{Var}\left(z_j^{(1)}\right)}$ as $n_1 \to \infty$. As such, in the limit $n_1 \to \infty$, we have that for both $x$ and $x'$ with probability 1:

$$\tilde{x}_j \to= \frac{z_j^{(1)} - \mathbb{E}\left[z_j^{(1)}\right]}{\sqrt{\text{Var}\left(z_j^{(1)}\right)}} \sim \mathcal{N}(0, \mathcal{I}), \tag{12}$$

$$\tilde{x}_j' \to= \frac{z_j^{(1)'} - \mathbb{E}\left[z_j^{(1)'}\right]}{\sqrt{\text{Var}\left(z_j^{(1)'}\right)}} \sim \mathcal{N}(0, \mathcal{I}). \tag{13}$$

The joint distribution over $\tilde{x}$ and $\tilde{x}'$ is then fully specified by the covariance between $\tilde{x}_i$ and $\tilde{x}_j'$,

$$\text{Cov}\left(\tilde{x}_i, \tilde{x}_j'\right) = \frac{\text{Cov}\left(z_i^{(1)}, z_j^{(1)'}\right)}{\sqrt{\text{Var}\left(z_i^{(1)}\right) \text{Var}\left(z_j^{(1)'}\right)}}$$

$$= \frac{\left(\frac{1}{n_0} x^\top x' + \sigma_b^2\right) \delta_{ij}}{\sqrt{\left(\frac{1}{n_0} x^\top x + \sigma_b^2\right)\left(\frac{1}{n_0} x'^\top x' + \sigma_b^2\right)}}$$

$$= \mathcal{S}\left(x, x'; \sigma_b^2\right) \delta_{ij}$$

where $\mathcal{S}\left(x, x'; \sigma_b^2\right)$ is the soft-cosine similarity from Definition C.1. Note that due to normalisation, this is now also exactly the correlation between $\tilde{x}$ and $\tilde{x}'$.

Due to result of [36], we have that $\tilde{\Theta}\left(\tilde{x}, \tilde{x}'\right)$ converges to a limit that is a function of inputs $\tilde{x}, \tilde{x}'$. However, note that $\tilde{x}$ and $\tilde{x}'$ are now only related to $x$ and $x'$ jointly through $\mathcal{S}\left(x, x'; \sigma_b^2\right)$. As such, $\tilde{\Theta}\left(\tilde{x}, \tilde{x}'\right)$ must converge to a limit dependent on $x$ and $x'$ only through $\mathcal{S}\left(x, x'; \sigma_b^2\right)$. To finish the proof, it is thus sufficient to show the same happens for the term $\langle \nabla_{\theta_1} f_\theta(x), \nabla_{\theta_1} f_\theta(x') \rangle$. We have that

$$\langle \nabla_{\theta_1} f_\theta(x), \nabla_{\theta_1} f_\theta(x') \rangle = \langle \nabla_z f_\theta(x), \nabla_{z'} f_\theta(x') \rangle \left(\frac{1}{n_0} x^\top x' + \sigma_b^2\right)$$

$$= \langle \frac{\partial \tilde{x}}{\partial z} \nabla_{\tilde{x}} \tilde{f}_{\tilde{\theta}}(\tilde{x}), \frac{\partial \tilde{x}'}{\partial z'} \nabla_{\tilde{x}} \tilde{f}_{\tilde{\theta}}(\tilde{x}) \rangle \left(\frac{1}{n_0} x^\top x' + \sigma_b^2\right).$$

where the term $\left(\frac{1}{n_0} x^\top x' + \sigma_b^2\right)$ is due to the multipliers of the weight and bias parameters in the layer.

For the Layer Norm operator, we have:

$$\frac{\partial \tilde{\boldsymbol{x}}}{\partial \boldsymbol{z}^{(1)}} = \frac{1}{\sigma} \left( \mathcal{I} - \frac{1}{n_0} \mathbf{1}\mathbf{1}^\top \right) - \frac{1}{\sigma^3 n_0} \left( \boldsymbol{z}^{(1)} - \mathbf{1}\mu \right) \left( \boldsymbol{z}^{(1)} - \mathbf{1}\mu \right)^\top$$

$$= \frac{1}{\sigma} \left( \mathcal{I} - \frac{1}{n_0} \mathbf{1}\mathbf{1}^\top \right) - \frac{\frac{1}{n_0}\boldsymbol{z}^{(1)}\boldsymbol{z}^{(1)\top}}{\sigma^3} + \mathbf{1}\mu \frac{1}{\sigma^3 n_0} \left( 2\boldsymbol{z}^{(1)} - \mathbf{1}\mu \right)^\top.$$

Due to law of large numbers we have $\lim_{n_0 \to \infty} \frac{1}{n_0} \boldsymbol{z}^{(1)}\boldsymbol{z}^{(1)\top} \to \mathbb{E}[\boldsymbol{z}^{(1)}\boldsymbol{z}^{(1)\top}] = \text{Var}(\boldsymbol{z}^{(1)}) = \mathcal{I}\text{Var}\left( z_j^{(1)} \right)$. Applying asymptotic consistency of MLE again, we get:

$$\frac{\partial \tilde{\boldsymbol{x}}}{\partial \boldsymbol{z}^{(1)}} \to \frac{1}{\sqrt{\text{Var}\left( z_j^{(1)} \right)}} (\mathcal{I} - \frac{1}{n_0}\mathbf{1}\mathbf{1}^\top) - \mathcal{I}\frac{\text{Var}\left( z_j^{(1)} \right)}{\text{Var}\left( z_j^{(1)} \right)^{3/2}} = -\frac{1}{\sqrt{\text{Var}\left( z_j^{(1)} \right)}} \left( \frac{1}{n_0}\mathbf{1}\mathbf{1}^\top \right).$$

Plugging this expression for both $\boldsymbol{x}$ and $\boldsymbol{x}'$ in to the full gradient formula:

$$\langle \nabla_{\theta_1} f_\theta(\boldsymbol{x}), \nabla_{\theta_1} f_\theta(\boldsymbol{x}') \rangle = \left\langle \left( \frac{1}{n_0}\mathbf{1}\mathbf{1}^\top \right) \nabla_{\tilde{\boldsymbol{x}}} \tilde{f}_{\tilde{\theta}}(\tilde{\boldsymbol{x}}), (\frac{1}{n_0}\mathbf{1}\mathbf{1}^\top) \nabla_{\tilde{\boldsymbol{x}}'} \tilde{f}_{\tilde{\theta}}(\tilde{\boldsymbol{x}}') \right\rangle \frac{\left( \frac{1}{n_0}\boldsymbol{x}^\top\boldsymbol{x}' + \sigma_b^2 \right)}{\sqrt{\text{Var}\left( z_j^{(1)} \right) \text{Var}\left( z_j^{(1)'} \right)}}$$

$$= \left\langle \left( \frac{1}{n_0}\mathbf{1}\mathbf{1}^\top \right) \nabla_{\tilde{\boldsymbol{x}}} \tilde{f}_{\tilde{\theta}}(\tilde{\boldsymbol{x}}), \left( \frac{1}{n_0}\mathbf{1}\mathbf{1}^\top \right) \nabla_{\tilde{\boldsymbol{x}}'} \tilde{f}_{\tilde{\theta}}(\tilde{\boldsymbol{x}}') \right\rangle \frac{\left( \frac{1}{n_0}\boldsymbol{x}^\top\boldsymbol{x}' + \sigma_b^2 \right)}{\sqrt{\left( \frac{1}{n_0}\boldsymbol{x}^\top\boldsymbol{x} + \sigma_b^2 \right)\left( \frac{1}{n_0}\boldsymbol{x}'^\top\boldsymbol{x}' + \sigma_b^2 \right)}}$$

$$= \left\langle \left( \frac{1}{n_0}\mathbf{1}\mathbf{1}^\top \right) \nabla_{\tilde{\boldsymbol{x}}} \tilde{f}_{\tilde{\theta}}(\tilde{\boldsymbol{x}}), \left( \frac{1}{n_0}\mathbf{1}\mathbf{1}^\top \right) \nabla_{\tilde{\boldsymbol{x}}'} \tilde{f}_{\tilde{\theta}}(\tilde{\boldsymbol{x}}') \right\rangle \mathcal{S}\left( \boldsymbol{x}, \boldsymbol{x}'; \sigma_b^2 \right).$$

We see that the expression depends only on the gradient $\nabla_{\tilde{\boldsymbol{x}}} \tilde{f}_{\tilde{\theta}}(\tilde{\boldsymbol{x}})$ of the network $\tilde{f}_{\tilde{\theta}}(\tilde{\boldsymbol{x}})$ w.r.t. input $\tilde{\boldsymbol{x}}$ and the soft-cosine similarity $\mathcal{S}\left( \boldsymbol{x}, \boldsymbol{x}'; \sigma_b^2 \right)$, but as we discussed before, $\tilde{\boldsymbol{x}}$ and $\tilde{\boldsymbol{x}}'$ are independent of the inputs $\boldsymbol{x}$ and $\boldsymbol{x}'$ except for jointly through $\mathcal{S}\left( \boldsymbol{x}, \boldsymbol{x}'; \sigma_b^2 \right)$. Thus, we conclude that the NTK for any network of this form can only depend on the soft-cosine similarity between the two inputs $\boldsymbol{x}$ and $\boldsymbol{x}'$, $\mathcal{S}\left( \boldsymbol{x}, \boldsymbol{x}'; \sigma_b^2 \right)$.

$\square$

# E  Auxillary Results

**Proposition E.1.** *For a bounded-variance kernel (i.e. $\forall_{\boldsymbol{x} \in \mathbb{R}^{n_0}} k(\boldsymbol{x}, \boldsymbol{x}) \leq C$ for some $C > 0$), we have that the corresponding kernel feature map $\psi(\cdot)$ (i.e. $k(\boldsymbol{x}, \boldsymbol{x}') = \psi(\boldsymbol{x})^\top \psi(\boldsymbol{x}')$) has the property of:*

$$\forall_{\boldsymbol{x} \in \mathbb{R}^{n_0}} \|\psi(\boldsymbol{x})\| \leq \sqrt{C}$$

*Proof.* Clearly for all $\boldsymbol{x} \in \mathbb{R}^{n_0}$:

$$k(\boldsymbol{x}, \boldsymbol{x}) = \psi(\boldsymbol{x})^\top \psi(\boldsymbol{x}) = \|\psi(\boldsymbol{x})\|_2^2 \leq C.$$

Taking square root of both side of last equation completes the proof. $\square$

**Lemma E.1.** *Consider an asymptotically positive $n$-homogeneous function (with $n > 0.5$) function $\phi : \mathbb{R} \to \mathbb{R}$ satisfying Assumption 3.1. Then, the derivative of $\phi$, $\dot{\phi}(\cdot)$ will also be asymptotically positive $(n-1)$-homogeneous. Moreover, if $\phi(\cdot)$ is positive $n$-homogeneous, then $\dot{\phi}(\cdot)$ will be positive $(n-1)$-homogeneous.*

*Proof.* We can decompose $\phi$ as $\phi(\cdot) = \hat{\phi}(\cdot) + \tilde{\phi}(\cdot)$, where $\hat{\phi}(x) = \lim_{\lambda\to\infty} \frac{\phi(\lambda x)}{\lambda^n}$ is positive $n$-homogeneous, both $\hat{\phi}(\cdot)$ and $\tilde{\phi}(\cdot)$ satisfy Assumption 3.1, and $\lim_{x\to\pm\infty} \frac{\phi(x)}{\lambda^n} = 0$, ie $\tilde{\phi}(\cdot)$ has bidirectional asymptotic growth bounded by a polynomial of order $\tilde{n} < n$. Moreover, we can rewrite $\hat{\phi}(x) = \|x\|^n f(\text{sign}(x))$ where $f(\cdot) : \{-1, 0, 1\} \to \mathbb{R}$ is bounded.

By linearity of differentiation, $\dot{\phi}(\cdot) = \dot{\hat{\phi}}(\cdot) + \dot{\tilde{\phi}}(\cdot)$. $\text{sign}(\cdot)$ is constant almost-everywhere, and is multiplied by 0 where it is not, thus does not contribute to the derivative of $\hat{\phi}$. Then, $\dot{\hat{\phi}}(x) = (n-1) \|x\|^{n-1} \text{sign}(x) f(\text{sign}(x)) = \|x\|^{n-1} f_1(\text{sign}(x))$, which is clearly positive $(n-1)$-homogeneous. As $\phi(\cdot)$ is almost-everywhere differentiable by Assumption 3.1, $\dot{\tilde{\phi}}(\cdot)$ exists and is defined on $\mathbb{R} \setminus E$ where $E$ is a set of measure zero. Moreover, as $\tilde{\phi}(\cdot)$ has bidirectional asymptotic growth bounded by a polynomial of order $\tilde{n}$, $\dot{\tilde{\phi}}$ has bidirectional asymptotic growth bounded by a polynomial of order $\tilde{n} - 1 < n - 1$. Thus,

$$\lim_{\lambda\to\infty} \frac{\dot{\phi}(\lambda x)}{\lambda^{n-1}} = \lim_{\lambda\to\infty} \frac{\dot{\hat{\phi}}(\lambda x)}{\lambda^{n-1}} + \lim_{\lambda\to\infty} \frac{\dot{\tilde{\phi}}(\lambda x)}{\lambda^{n-1}}$$
$$= \dot{\hat{\phi}}(x)$$

$\square$

where the second line follows from the fact that $\dot{\hat{\phi}}(x)$ is positive $(n-1)$-homogeneous and $\tilde{n}$, $\dot{\tilde{\phi}}$ has bidirectional asymptotic growth bounded by a polynomial of order $\tilde{n} - 1 < n - 1$. Hence, $\dot{\phi}(\lambda x)$ is asymptotically positive $(n-1)$-homogeneous.

Clearly, if $\phi(\cdot)$ is positive $n$-homogeneous, then $\phi(\cdot) = \hat{\phi}(\cdot)$. Thus, $\dot{\phi}(\cdot) = \dot{\hat{\phi}}(\cdot)$, so will be positive $(n-1)$-homogeneous.

**Lemma E.2.** *Consider an asymptotically positive $n$-homogeneous (with $n > 0$), nonlinear function $\phi(\cdot) : \mathbb{R} \to \mathbb{R} \in L^2(\mathbb{R}, \gamma)$, where $\gamma$ is the standard normal measure. Then, for zero-mean, joint-distributed scalar random variables with correlation $\rho$, ie $(u, v) \sim \mathcal{N}(0, \Lambda)$ where*
$$\Lambda = \begin{bmatrix} \sigma_u^2 & \sigma_u \sigma_v \rho \\ \sigma_v \sigma_u \rho & \sigma_v^2 \end{bmatrix},$$

$$c_\phi \mathbb{E}_{(u,v)} [\phi(u)\phi(v)] = (\sigma_u \sigma_v)^n \kappa(\rho) + R(\sigma_u, \sigma_v, \rho)$$

*where $c_\phi = \left( \lim_{\sigma\to\infty} \frac{\mathbb{E}_{u\sim\mathcal{N}(0,\sigma^2)}[\phi(u)^2]}{\sigma^{2n}} \right)^{-1}$, $\kappa(\rho) : [-1, 1] \to [-1, 1]$ satisfies $\kappa(1) = 1$, and $\lim_{\sigma,\sigma'\to\infty} \frac{R(\sigma,\sigma',\rho)}{(\sigma\sigma')^n} = 0 \ \forall \sigma_v > 0, \rho \in [-1, 1]$.*

*Proof.* First, let us change variables from $(u, v)$ to $(x, y)$, where $(u, v) = (\sigma_u x, \sigma_v y)$, and thus $(x, y)$ are joint normally distributed with unit variance and covariance $\rho$. That is, $(x, y) \sim \mathcal{N}(0, \Lambda_{xy})$, where $\Lambda_{xy} = \begin{bmatrix} 1 & \rho \\ \rho & 1 \end{bmatrix}$. Thus,

$$\mathbb{E}_{(u,v)} [\phi(u)\phi(v)] = \mathbb{E}_{(x,y)} [\phi(\sigma_u x)\phi(\sigma_v y)]$$

As $\phi(\cdot)$ is asymptotically positive $n$-homogeneous, as above we can decompose $\phi$ as

$$\phi(\cdot) = \hat{\phi}(\cdot) + \tilde{\phi}(\cdot), \tag{14}$$

where $\hat{\phi}(x) = \lim_{\lambda\to\infty} \frac{\phi(\lambda x)}{\lambda^n}$ is $n$-homogeneous, and $\lim_{x\to\pm\infty} \frac{\tilde{\phi}(\lambda x)}{\lambda^n} = 0$, ie $\tilde{\phi}(\cdot)$ has bidirectional asymptotic growth bounded by a polynomial of order $\tilde{n}$. Then,

$$\phi(\sigma_u x)\phi(\sigma_v y) = \left(\hat{\phi}(\sigma_u x) + \tilde{\phi}(\sigma_u x)\right)\left(\hat{\phi}(\sigma_v y) + \tilde{\phi}(\sigma_v y)\right)$$

$$= \hat{\phi}(\sigma_u x)\hat{\phi}(\sigma_v y) + \hat{\phi}(\sigma_u x)\tilde{\phi}(\sigma_v y) + \tilde{\phi}(\sigma_u x)\hat{\phi}(\sigma_v y) + \tilde{\phi}(\sigma_u x)\tilde{\phi}(\sigma_v y)$$

$$= (\sigma_u \sigma_v)^n \hat{\phi}(x)\hat{\phi}(y) + \sigma_u^n \hat{\phi}(x)\tilde{\phi}(\sigma_v y) + \sigma_v^n \tilde{\phi}(\sigma_u x)\hat{\phi}(y) + \tilde{\phi}(\sigma_u x)\tilde{\phi}(\sigma_v y)$$

$$= (\sigma_u \sigma_v)^n \hat{\phi}(x)\hat{\phi}(y) + \tilde{R}(x, y; \sigma_u, \sigma_b)$$

where

$$\tilde{R}(x, y; \sigma_u, \sigma_b, \rho) = \sigma_u^n \hat{\phi}(x)\tilde{\phi}(\sigma_v y) + \sigma_v^n \tilde{\phi}(\sigma_u x)\hat{\phi}(y) + \tilde{\phi}(\sigma_u x)\tilde{\phi}(\sigma_v y) \tag{15}$$

clearly has asymptotic growth in $\sigma_u \sigma_v$ bounded by a polynomial of order $\tilde{n} < n$, such that

$$\lim_{\sigma, \sigma' \to \infty} \frac{\tilde{R}(x, y; \sigma, \sigma')}{(\sigma \sigma')^n} = 0 \quad \forall x, y \in \mathbb{R}, \sigma_u, \sigma_v > 0, \rho \in [-1, 1].$$

Thus,

$$\mathbb{E}_{(x,y)}\left[\phi(\sigma_u x)\phi(\sigma_v y)\right] = \mathbb{E}_{(x,y)}\left[\hat{\phi}(x)\hat{\phi}(y) + \tilde{R}(x, y; \sigma_u, \sigma_v)\right]$$

$$= (\sigma_u \sigma_v)^n \mathbb{E}_{(x,y)}\left[\hat{\phi}(\sigma_u x)\hat{\phi}(\sigma_v y)\right] + \mathbb{E}_{(x,y)}\left[\tilde{R}(x, y; \sigma_u, \sigma_v)\right]$$

$$= (\sigma_u \sigma_v)^n \hat{\kappa}(\rho) + \hat{R}(\sigma_u, \sigma_v, \rho) \tag{16}$$

where

$$\hat{\kappa}(\rho) = \mathbb{E}_{(x,y)}\left[\hat{\phi}(x)\hat{\phi}(y)\right] \tag{17}$$

trivially depends only on $\rho$ and

$$\hat{R}(\sigma_u, \sigma_v, \rho) = \mathbb{E}_{(x,y)}\left[\tilde{R}(x, y; \sigma_u, \sigma_b,)\right]. \tag{18}$$

We have that,

$$\lim_{\sigma, \sigma' \to \infty} \frac{\hat{R}(\sigma, \sigma', \rho)}{(\sigma \sigma')^n} = \lim_{\sigma, \sigma' \to \infty} \frac{\mathbb{E}_{(x,y)}\left[\tilde{R}(x, y; \sigma, \sigma')\right]}{(\sigma \sigma')^n}$$

$$= \mathbb{E}_{(x,y)}\left[\lim_{\sigma, \sigma' \to \infty} \frac{\tilde{R}(x, y; \sigma, \sigma')}{(\sigma \sigma')^n}\right]$$

$$= 0 \quad \forall \rho \in [-1, 1]$$

where the second line follows from the dominated convergence theorem. We then have that

$$\lim_{\sigma \to \infty} \frac{\mathbb{E}_{u \sim \mathcal{N}(0, \sigma^2)}\left[\phi(u)^2\right]}{\sigma^{2n}} = \lim_{\sigma \to \infty} \frac{\sigma^{2n}\hat{\kappa}(1) + \hat{R}(\sigma, \sigma, 1)}{\sigma^{2n}}$$

$$= \lim_{\sigma \to \infty} \frac{\sigma^{2n}\hat{\kappa}(1)}{\sigma^{2n}} + \lim_{\sigma \to \infty} \frac{\hat{R}(\sigma, \sigma, 1)}{\sigma^{2n}}$$

$$= \hat{\kappa}(1).$$

Thus,

$$c_\phi = \left( \lim_{\sigma \to \infty} \frac{\mathbb{E}_u \sim \mathcal{N}\left(0, \sigma^2\right)\left[\phi(u)^2\right]}{\sigma^{2n}} \right)^{-1} = \frac{1}{\hat{\kappa}(1)}. \tag{19}$$

Finally, defining

$$\kappa(\rho) = \frac{\hat{\kappa}(\rho)}{\hat{\kappa}(1)}, \tag{20}$$

$$R(\sigma_u, \sigma_v, \rho) = \frac{\hat{R}(\sigma_u, \sigma_v, \rho)}{\hat{\kappa}(1)}, \tag{21}$$

where $\kappa : [-1, 1] \to [-1, 1]$,

$$c_\phi \mathbb{E}_{(u,v)}\left[\phi(u)\phi(v)\right] = (\sigma_u \sigma_v)^n \kappa(\rho) + R(\sigma_u, \sigma_v, \rho)$$

where $c_\phi = \left( \lim_{\sigma \to \infty} \frac{\mathbb{E}_u \sim \mathcal{N}\left(0, \sigma^2\right)\left[\phi(u)^2\right]}{\sigma^{2n}} \right)^{-1}$, $\kappa(\rho) : [-1, 1] \to [-1, 1]$ satisfies $\kappa(1) = 1$, and $\lim_{\sigma, \sigma' \to \infty} \frac{R(\sigma, \sigma', \rho)}{(\sigma\sigma')^n} = 0 \; \forall \rho \in [-1, 1]$ as required, completing the proof.

$\square$

**Proposition E.2.** *The NTK of a neural network with L fully-connected layers and nonlinearities satisfying Assumption 3.1 and Assumption C.1, takes the following form:*

$$\Theta\left(\boldsymbol{x}, \boldsymbol{x}'\right) = \left( \frac{1}{n_0} \|\boldsymbol{x}\| \|\boldsymbol{x}'\| \right)^{n^L} \bar{\kappa}\left(\boldsymbol{x}, \boldsymbol{x}'\right) + \bar{R}\left(\boldsymbol{x}, \boldsymbol{x}'\right)$$

*where $\bar{\kappa}\left(\boldsymbol{x}, \boldsymbol{x}'\right) \in [-C, C]$, $\bar{\kappa}\left(\boldsymbol{x}, \boldsymbol{x}\right) = C \; \forall \boldsymbol{x}, \boldsymbol{x}' \in \mathbb{R}^{n_0}$ for some constant $C > 0$, and $\lim_{\lambda, \lambda' \to \infty} \frac{R(\lambda \boldsymbol{x}, \boldsymbol{x}')}{(\lambda\lambda')^n} = 0 \; \forall \boldsymbol{x}, \boldsymbol{x}' \in \mathbb{R}^{n_0}$, ie $R(\cdot, \boldsymbol{x}')$ has bidirectional asymptotic growth bounded by a polynomial of order $n_R < n$*

*Proof.* Recall from [22] that in the infinite-width limit, the pre-activations $f^{(h)}(\boldsymbol{x})$ at every hidden layer $h \in [L]$ have all their coordinates tending to i.i.d. centered Gaussian processes with covariance

$$\Sigma^{(h)} : \mathbb{R}^d \times \mathbb{R}^d \to \mathbb{R}$$

defined recursively as follows:

$$\Sigma^{(0)}(\boldsymbol{x}, \boldsymbol{x}') = \frac{1}{n_0}\boldsymbol{x}^\top \boldsymbol{x}' + \sigma_b^2,$$

$$\Lambda^{(h)}(\boldsymbol{x}, \boldsymbol{x}') = \begin{bmatrix} \Sigma^{(h-1)}(\boldsymbol{x}, \boldsymbol{x}) & \Sigma^{(h-1)}(\boldsymbol{x}, \boldsymbol{x}') \\ \Sigma^{(h-1)}(\boldsymbol{x}', \boldsymbol{x}) & \Sigma^{(h-1)}(\boldsymbol{x}', \boldsymbol{x}') \end{bmatrix} \in \mathbb{R}^{2 \times 2}, \tag{22}$$

$$\Sigma^{(h)}(\boldsymbol{x}, \boldsymbol{x}') = c_\phi \mathbb{E}_{(u,v) \sim \mathcal{N}(0, \Lambda^{(h)}(\boldsymbol{x}, \boldsymbol{x}'))}\left[\phi(u)\phi(v)\right] + \sigma_b^2,$$

where $c_\phi = \left( \mathbb{E}_{u \sim \mathcal{N}(0,1)}\left[\phi(u)^2\right] \right)^{-1}$. Note that while [1] use $\sigma(\cdot)$ to denote nonlinearities, we use $\phi(\cdot)$ to prevent ambiguity with standard deviations, which we denote with variants of $\sigma$. Note also that $c_\phi$ is an arbitrary normalisation constant used to control the magnitude of the variance throughout the network, but is ill-defined in this form for non-exactly positive homogeneous activation functions (ie, all but powers of ReLU, Leaky ReLU, and Parametric ReLU), hence we freely redefine this to match the definition provided in Equation 19, namely $c_\phi = \lim_{\lambda \to \infty} \left( \frac{\mathbb{E}_{u \sim \mathcal{N}(0, \lambda)}\left[\phi(u)^2\right]}{\lambda^{2n}} \right)^{-1}$ for

asymptotically positive $n$-homogeneous $\phi(\cdot)$. Note that this definition is equivalent to the previous one for positive $n$-homogeneous activation functions such as ReLU.

To give the formula of the NTK, we also define a derivative covariance:

$$\dot{\Sigma}^{(h)}(\boldsymbol{x}, \boldsymbol{x}') = c_\phi \mathbb{E}_{(u,v) \sim \mathcal{N}(0, \Lambda^{(h)}(\boldsymbol{x}, \boldsymbol{x}'))} \left[ \dot{\phi}(u) \dot{\phi}(v) \right]. \tag{23}$$

The final NTK expression for the fully-connected neural network is [1, 36]:

$$
\begin{aligned}
\Theta\left(\boldsymbol{x}, \boldsymbol{x}'\right) &= \sum_{h=1}^{L+1} \Sigma^{(h-1)}(\boldsymbol{x}, \boldsymbol{x}') \cdot \prod_{h'=h}^{L+1} \dot{\Sigma}^{(h')}(\boldsymbol{x}, \boldsymbol{x}') \\
&= \sum_{h=0}^{L} \Theta^{(L,h)},
\end{aligned}
\tag{24}
$$

where $\dot{\Sigma}^{(L+1)}(\boldsymbol{x}, \boldsymbol{x}') = 1$ for convenience and

$$\Theta^{(L,h)} = \Sigma^{(h)}\left(\boldsymbol{x}, \boldsymbol{x}'\right) \prod_{h'=h+1}^{L+1} \dot{\Sigma}^{(h')}\left(\boldsymbol{x}, \boldsymbol{x}'\right) \tag{25}$$

is the contribution of the $h$th layer to the NTK. Applying Lemma E.2 to the recursion for $\Sigma^{(h)}(\boldsymbol{x}, \boldsymbol{x}')$, Recursion 22, we get that

$$
\begin{aligned}
\Sigma^{(h+1)}(\boldsymbol{x}, \boldsymbol{x}') &= \left( \Sigma^{(h)}(\boldsymbol{x}, \boldsymbol{x}) \Sigma^{(h)}\left(\boldsymbol{x}', \boldsymbol{x}'\right) \right)^{\frac{n}{2}} \kappa\left( \rho^{(h)}\left(\boldsymbol{x}, \boldsymbol{x}'\right) \right) \\
&\quad + R\left( \sqrt{\Sigma^{(h)}(\boldsymbol{x}, \boldsymbol{x})}, \sqrt{\Sigma^{(h)}\left(\boldsymbol{x}', \boldsymbol{x}'\right)}, \rho^{(h)}\left(\boldsymbol{x}, \boldsymbol{x}'\right) \right) + \sigma_b^2,
\end{aligned}
\tag{26}
$$

where

$$\rho^{(h)}\left(\boldsymbol{x}, \boldsymbol{x}'\right) = \frac{\Sigma^{(h)}\left(\boldsymbol{x}, \boldsymbol{x}'\right)}{\sqrt{\Sigma^{(h)}(\boldsymbol{x}, \boldsymbol{x}) \Sigma^{(h)}\left(\boldsymbol{x}', \boldsymbol{x}'\right)}} \in [-1, 1]. \tag{27}$$

We therefore also get that

$$\Sigma^{(h+1)}(\boldsymbol{x}, \boldsymbol{x}) = \left( \Sigma^{(h)}(\boldsymbol{x}, \boldsymbol{x}) \right)^n + R\left( \sqrt{\Sigma^{(h)}(\boldsymbol{x}, \boldsymbol{x})}, \sqrt{\Sigma^{(h)}(\boldsymbol{x}, \boldsymbol{x})}, 1 \right) + \sigma_b^2, \tag{28}$$

which follows from the fact that $\rho^{(h)}(\boldsymbol{x}, \boldsymbol{x}) = 1 \ \forall \boldsymbol{x} \in \mathbb{R}^{n_0}, h$ by definition and $\kappa(1) = 1$. Applying this recursion to $\Sigma^{(0)}(\boldsymbol{x}, \boldsymbol{x}) = \frac{1}{n_0} \boldsymbol{x}^\top \boldsymbol{x} + \sigma_b^2$ we inductively get that

$$
\begin{aligned}
\Sigma^{(h)}(\boldsymbol{x}, \boldsymbol{x}) &= \left( \frac{1}{n_0} \boldsymbol{x}^\top \boldsymbol{x} \right)^{n^h} + R^{(h)}(\boldsymbol{x}, \boldsymbol{x}) \\
&= \frac{1}{n_0} \|\boldsymbol{x}\|^{2n^h} + R^{(h)}(\boldsymbol{x}, \boldsymbol{x})
\end{aligned}
\tag{29}
$$

where $R^{(h)}\left(\boldsymbol{x}, \boldsymbol{x}'\right)$ is defined recursively by

$$R^{(h+1)}\left(\boldsymbol{x}, \boldsymbol{x}'\right) = \left(\sum_{k=0}^{n-1} \binom{n}{k} \left(\frac{1}{n_0} \|\boldsymbol{x}\| \|\boldsymbol{x}'\|\right)^{kn^h} R^{(h)}\left(\boldsymbol{x}, \boldsymbol{x}'\right)^{(n-k)}\right)$$

$$+ R\left(\sqrt{\frac{1}{n_0} \|\boldsymbol{x}\|^{2n^h} + R^{(h)}(\boldsymbol{x}, \boldsymbol{x})}, \sqrt{\frac{1}{n_0} \|\boldsymbol{x}'\|^{2n^h} + R^{(h)}\left(\boldsymbol{x}', \boldsymbol{x}'\right)}, \rho^{(h)}\left(\boldsymbol{x}, \boldsymbol{x}'\right)\right)$$

$$+ \sigma_b^2,$$

$$R^{(0)} = \sigma_b^2,$$

$$\tag{30}$$

and satisfies $\lim_{\lambda, \lambda' \to \infty} \frac{R^{(h+1)}(\lambda \boldsymbol{x}, \lambda' \boldsymbol{x}')}{(\lambda \lambda')^{n^{(h+1)}}} = 0$. To show this, we take an inductive approach. First, note that for the case $n = 0$ it trivially holds as $R^{(0)}\left(\boldsymbol{x}, \boldsymbol{x}'\right) = \sigma_b^2 \; \forall \boldsymbol{x}, \boldsymbol{x}' \in \mathbb{R}^{n_0}$, a constant. Then assuming this holds for $R^{(h)}\left(\boldsymbol{x}, \boldsymbol{x}'\right)$, we have that

$$\lim_{\lambda, \lambda' \to \infty} \frac{R^{(h+1)}(\lambda \boldsymbol{x}, \lambda' \boldsymbol{x}')}{(\lambda \lambda')^{n^{(h+1)}}} = \lim_{\lambda, \lambda' \to \infty} \left(\sum_{k=0}^{n-1} \binom{n}{k} \frac{\left(\frac{1}{n_0} \|\lambda \boldsymbol{x}\| \|\lambda' \boldsymbol{x}'\|\right)^{kn^h} R^{(h)}(\lambda \boldsymbol{x}, \lambda' \boldsymbol{x}')^{(n-k)}}{(\lambda \lambda')^{n^{(h+1)}}}\right)$$

$$+ \frac{R\left(\sqrt{\frac{1}{n_0} \|\lambda \boldsymbol{x}\|^{2n^h} + R^{(h)}(\lambda \boldsymbol{x}, \lambda \boldsymbol{x})}, \sqrt{\frac{1}{n_0} \|\lambda' \boldsymbol{x}'\|^{2n^h} + R^{(h)}\left(\lambda' \boldsymbol{x}', \lambda' \boldsymbol{x}'\right)}, \rho^{(h)}(\lambda \boldsymbol{x}, \lambda' \boldsymbol{x}')\right)}{(\lambda \lambda')^{n^{(h+1)}}}$$

$$+ \frac{\sigma_b^2}{(\lambda \lambda')^{n^{(h+1)}}}$$

$$= \lim_{\lambda, \lambda' \to \infty} \left(\sum_{k=0}^{n-1} \binom{n}{k} \frac{(\lambda \lambda')^{kn^h} \left(\frac{1}{n_0} \|\boldsymbol{x}\| \|\boldsymbol{x}'\|\right)^{kn^h} R^{(h)}(\lambda \boldsymbol{x}, \lambda' \boldsymbol{x}')^{(n-k)}}{(\lambda \lambda')^{n^{(h+1)}}}\right)$$

$$+ \frac{R\left(\lambda^{n^h} \sqrt{\frac{1}{n_0} \|\boldsymbol{x}\|^{2n^h} + \frac{R^{(h)}(\lambda \boldsymbol{x}, \lambda \boldsymbol{x})}{\lambda^{2n^h}}}, (\lambda')^{n^h} \sqrt{\frac{1}{n_0} \|\boldsymbol{x}'\|^{2n^h} + R^{(h)}\left(\boldsymbol{x}', \boldsymbol{x}'\right)}, \rho^{(h)}(\lambda \boldsymbol{x}, \lambda' \boldsymbol{x}')\right)}{(\lambda \lambda')^{n^{(h+1)}}}$$

$$= \lim_{\lambda, \lambda' \to \infty} \left(\sum_{k=0}^{n-1} \binom{n}{k} \frac{\frac{1}{n_0} (\|\boldsymbol{x}\| \|\boldsymbol{x}'\|)^{kn^h} R^{(h)}(\lambda \boldsymbol{x}, \lambda' \boldsymbol{x}')^{(n-k)}}{(\lambda \lambda')^{n^{(h+1)} - kn^h}}\right)$$

$$+ \frac{R\left(\lambda^{n^h} \left(\frac{1}{\sqrt{n_0}} \|\boldsymbol{x}\|\right)^{n^h}, (\lambda')^{n^h} \left(\frac{1}{\sqrt{n_0}} \|\boldsymbol{x}'\|\right)^{n^h}, \rho^{(h)}(\lambda \boldsymbol{x}, \lambda' \boldsymbol{x}')\right)}{(\lambda \lambda')^{n^{(h+1)}}}$$

$$= \left(\sum_{k=0}^{n-1} \binom{n}{k} \left(\frac{1}{n_0} \|\boldsymbol{x}\| \|\boldsymbol{x}'\|\right)^{kn^h} \left(\lim_{\lambda, \lambda' \to \infty} \frac{R^{(h)}(\lambda \boldsymbol{x}, \lambda' \boldsymbol{x}')}{\lambda^{n^h}}\right)^{(n-k)}\right)$$

$$+ \lim_{\tilde{\lambda}, \tilde{\lambda}' \to \infty} \frac{R\left(\tilde{\lambda} \left(\frac{1}{\sqrt{n_0}} \|\boldsymbol{x}\|\right)^{n^h}, \tilde{\lambda}' \left(\frac{1}{\sqrt{n_0}} \|\boldsymbol{x}'\|\right)^{n^h}, \lim_{\lambda, \lambda' \to \infty} \rho^{(h)}(\lambda \boldsymbol{x}, \lambda' \boldsymbol{x}')\right)}{\left(\tilde{\lambda} \tilde{\lambda}'\right)^n}$$

$$= 0$$

where the transition from the 5th to the 7th line follows from the closely related property $\lim_{\lambda \to \infty} \frac{R^{(h)}(\lambda \boldsymbol{x}, \lambda \boldsymbol{x})}{\lambda^{2n^h}} = 0$ and the equivalent for $\lambda'$, the third-to-last line follows from direct application of the algebraic limit theorem and the fact that $n - k > 0$, and the second-to-last line follows from

an aforementioned equivalent property of $R(\boldsymbol{x}, \boldsymbol{x}', \rho)$, the fact that $\rho(\lambda\boldsymbol{x}, \lambda'\boldsymbol{x}') \in [-1, 1]$ $\lambda, \lambda' > 0$, and the fact that for $\tilde{\lambda} = \lambda^{n^h}$ $\lim_{\lambda \to \infty} \equiv \lim_{\lambda' \to \infty}$ as $n^h > 0$ and the equivalent for $\tilde{\lambda}'$.

Applying the above to Equation 26, we then get that

$$\Sigma^{(h+1)}\left(\boldsymbol{x}, \boldsymbol{x}'\right) = \left(\frac{1}{n_0} \|\boldsymbol{x}\| \|\boldsymbol{x}'\|\right)^{n^{h+1}} \kappa\left(\rho^{(h)}\left(\boldsymbol{x}, \boldsymbol{x}'\right)\right) + R^{(h+1)}\left(\boldsymbol{x}, \boldsymbol{x}'\right).$$

Due to Lemma E.1, $\dot{\phi}(\cdot)$ is asymptotically positive $(n-1)$-homogeneous. Thus, applying Lemma E.2 to Equation 23, we get that

$$\begin{aligned}
\dot{\Sigma}^{(h+1)}(\boldsymbol{x}, \boldsymbol{x}') &= \left(\Sigma^{(h)}(\boldsymbol{x}, \boldsymbol{x})\Sigma^{(h)}\left(\boldsymbol{x}', \boldsymbol{x}'\right)\right)^{\frac{n-1}{2}} \dot{\kappa}\left(\rho^{(h)}\left(\boldsymbol{x}, \boldsymbol{x}'\right)\right) \\
&\quad + \dot{R}\left(\sqrt{\Sigma^{(h)}(\boldsymbol{x}, \boldsymbol{x})}, \sqrt{\Sigma^{(h)}\left(\boldsymbol{x}', \boldsymbol{x}'\right)}, \rho^{(h)}\left(\boldsymbol{x}, \boldsymbol{x}'\right)\right),
\end{aligned} \tag{31}$$

where $\dot{\kappa}(\cdot)$ and $\dot{R}(\cdot, \cdot', \rho)$ are defined as the analogues to $\kappa(\cdot)$ and $R(\cdot, \cdot', \rho)$ for $\dot{\phi}(\cdot)$;

$$\dot{\kappa}(\rho) = \frac{\hat{\dot{\kappa}}(\rho)}{\hat{\kappa}(1)}, \tag{32}$$

$$\dot{R}\left(\cdot, \cdot', \rho\right) = \frac{\hat{\dot{R}}\left(\cdot, \cdot', \rho\right)}{\hat{\kappa}(1)} \tag{33}$$

where $\hat{\dot{\kappa}}(\rho)$ and $\hat{\dot{R}}(\cdot, \cdot', \rho)$ are defined by application of Equation 17 and Equation 18 respectively to $\dot{\phi}(\cdot)$. Note the different scaling (for example, $\dot{\kappa}(1) = \frac{\hat{\dot{\kappa}}(1)}{\hat{\kappa}(1)} \neq 1$ in general where $\hat{\dot{\kappa}}(\cdot)$ is the analogue to $\hat{\kappa}(\cdot)$). More specifically, we then have that $\dot{\kappa}(\cdot) \in \left[-\frac{\hat{\dot{\kappa}}(1)}{\hat{\kappa}(1)}, \frac{\hat{\dot{\kappa}}(1)}{\hat{\kappa}(1)}\right]$. Inserting the above expression for $\Sigma^{(h+1)}(\boldsymbol{x}, \boldsymbol{x}')$, we get that

$$\dot{\Sigma}^{(h)}\left(\boldsymbol{x}, \boldsymbol{x}'\right) = \left(\frac{1}{n_0} \|\boldsymbol{x}\| \|\boldsymbol{x}'\|\right)^{n^{h+1} - n^h} \dot{\kappa}\left(\rho^{(h)}\left(\boldsymbol{x}, \boldsymbol{x}'\right)\right) + \dot{R}^{(h+1)}\left(\boldsymbol{x}, \boldsymbol{x}'\right) \tag{34}$$

which follows from the fact that $\dot{\phi}(\cdot)$ is asymptotically positive $(n-1)$-homogeneous and $\dot{R}^{(h+1)}(\boldsymbol{x}, \boldsymbol{x}')$ satisfies equivalent properties to $R^{(h+1)}(\boldsymbol{x}, \boldsymbol{x}')$, namely $\lim_{\lambda \to \infty} \frac{\dot{R}^{(h+1)}(\lambda\boldsymbol{x}, \boldsymbol{x}')}{\lambda^{n^{h+1} - n^h}} = 0$ $\forall \boldsymbol{x}, \boldsymbol{x}' \in \mathbb{R}^{n_0}$ and $\lim_{\lambda, \lambda' \to \infty} \frac{\dot{R}^{(h+1)}(\lambda\boldsymbol{x}, \lambda'\boldsymbol{x}')}{(\lambda\lambda')^{n^{h+1} - n^h}} = 0$ $\forall \boldsymbol{x}, \boldsymbol{x}' \in \mathbb{R}^{n_0}$, which can be shown by similar derivation.

Inserting Equation 29 and Equation 34 into Equation 25, the contribution of the $h$th layer of the NTK,

$$\Theta^{(L,h)}\left(\boldsymbol{x}, \boldsymbol{x}'\right) = \Sigma^{(h)}\left(\boldsymbol{x}, \boldsymbol{x}'\right) \prod_{h'=h+1}^{L+1} \dot{\Sigma}^{(h')}\left(\boldsymbol{x}, \boldsymbol{x}'\right)$$

$$= \left(\left(\frac{1}{n_0}\|\boldsymbol{x}\|\,\|\boldsymbol{x}'\|\right)^{n^h} \kappa\left(\rho^{(h-1)}\left(\boldsymbol{x}, \boldsymbol{x}'\right)\right) + R^{(h)}\left(\boldsymbol{x}, \boldsymbol{x}'\right)\right)$$

$$\prod_{h'=h+1}^{L} \left(\left(\frac{1}{n_0}\|\boldsymbol{x}\|\,\|\boldsymbol{x}'\|\right)^{n^{h'}-n^{h'-1}} \dot{\kappa}\left(\rho^{(h'-1)}\left(\boldsymbol{x}, \boldsymbol{x}'\right)\right) + \dot{R}^{(h')}\left(\boldsymbol{x}, \boldsymbol{x}'\right)\right)$$

$$= \left(\frac{1}{n_0}\|\boldsymbol{x}\|\,\|\boldsymbol{x}'\|\right)^{n^h + \sum_{h'=h+1}^{L} n^{h'} - n^{h'-1}} \kappa\left(\rho^{(h-1)}\left(\boldsymbol{x}, \boldsymbol{x}'\right)\right) \prod_{h'=h+1}^{L} \dot{\kappa}\left(\rho^{(h'-1)}\left(\boldsymbol{x}, \boldsymbol{x}'\right)\right)$$

$$+ \bar{R}^{(h)}\left(\boldsymbol{x}, \boldsymbol{x}'\right)$$

$$= \left(\frac{1}{n_0}\|\boldsymbol{x}\|\,\|\boldsymbol{x}'\|\right)^{n^L} \bar{\kappa}^{(h)}\left(\boldsymbol{x}, \boldsymbol{x}'\right) + \bar{R}^{(h)}\left(\boldsymbol{x}, \boldsymbol{x}'\right) \tag{35}$$

implicitly accounting for the exceptions for $\dot{\Sigma}^{(L+1)}\left(\boldsymbol{x}, \boldsymbol{x}'\right) = 1$ and $\kappa\left(\rho^{(-1)}\left(\boldsymbol{x}, \boldsymbol{x}'\right)\right) = S\left(\boldsymbol{x}, \boldsymbol{x}'; \sigma_b^2\right)$ by abuse of notation, and where $\bar{R}^{(h)}\left(\boldsymbol{x}, \boldsymbol{x}'\right)$ accounts for all remaining (lower-order) terms, thus satisfying $\lim_{\lambda,\lambda'\to\infty} \frac{\bar{R}^{(h)}(\lambda\boldsymbol{x},\lambda'\boldsymbol{x}')}{(\lambda\lambda')^{n^L}} = 0 \;\forall \boldsymbol{x}, \boldsymbol{x}' \in \mathbb{R}^{n_0}, h = 0, \ldots, L$ with equivalent derivation to the above, and

$$\bar{\kappa}^{(h)}\left(\boldsymbol{x}, \boldsymbol{x}'\right) = \begin{cases} \kappa\left(\rho^{(h)}\left(\boldsymbol{x}, \boldsymbol{x}'\right)\right) \prod_{h'=h+1}^{L} \dot{\kappa}\left(\rho^{(h')}\left(\boldsymbol{x}, \boldsymbol{x}'\right)\right) & h = 0, \ldots L-1 \\ \bar{\kappa}^{(L)}\left(\boldsymbol{x}, \boldsymbol{x}'\right) = \kappa\left(\rho^{(L)}\left(\boldsymbol{x}, \boldsymbol{x}'\right)\right) & h = L \end{cases}, \tag{36}$$

noting that $\bar{\kappa}^{(h)}\left(\boldsymbol{x}, \boldsymbol{x}'\right) \in \left[-\left(\frac{\hat{\kappa}(1)}{\hat{\kappa}(1)}\right)^{L-h}, \left(\frac{\hat{\kappa}(1)}{\hat{\kappa}(1)}\right)^{L-h}\right] \;\forall h = 0, \ldots, L$. Finally, applying this to Equation 24 by summing over all layers we get

$$\Theta(\boldsymbol{x}, \boldsymbol{x}') = \sum_{h=0}^{L} \Theta^{(L,h)}\left(\boldsymbol{x}, \boldsymbol{x}'\right)$$

$$= \sum_{h=0}^{L} \left(\frac{1}{n_0}\|\boldsymbol{x}\|\,\|\boldsymbol{x}'\|\right)^{n^L} \bar{\kappa}^{(h)}\left(\boldsymbol{x}, \boldsymbol{x}'\right) + \bar{R}^{(h)}\left(\boldsymbol{x}, \boldsymbol{x}'\right)$$

$$= \left(\frac{1}{n_0}\|\boldsymbol{x}\|\,\|\boldsymbol{x}'\|\right)^{n^L} \bar{\kappa}\left(\boldsymbol{x}, \boldsymbol{x}'\right) + \bar{R}\left(\boldsymbol{x}, \boldsymbol{x}'\right) \tag{37}$$

where

$$\bar{\kappa}\left(\boldsymbol{x}, \boldsymbol{x}'\right) = \sum_{h=0}^{L} \bar{\kappa}^{(h)}\left(\boldsymbol{x}, \boldsymbol{x}'\right), \tag{38}$$

noting that $\bar{\kappa}\left(\boldsymbol{x}, \boldsymbol{x}'\right) \in [-C, C]$ where $C = \sum_{h=0}^{L} \left(\frac{\hat{\kappa}(1)}{\hat{\kappa}(1)}\right)^{h} = \frac{1 - \left(\frac{\hat{\kappa}(1)}{\hat{\kappa}(1)}\right)^{L}}{1 - \frac{\hat{\kappa}(1)}{\hat{\kappa}(1)}}$, and

$$\bar{R}\left(\boldsymbol{x}, \boldsymbol{x}'\right) = \sum_{h=0}^{L} \bar{R}^{(h)}\left(\boldsymbol{x}, \boldsymbol{x}'\right), \tag{39}$$

trivially satisfying $\lim_{\lambda,\lambda'\to\infty} \frac{\bar{R}(\lambda\boldsymbol{x},\lambda'\boldsymbol{x}')}{(\lambda\lambda')^{n^L}} = 0$ as this is satisfied by all components of the finite sum individually.

$\square$

**Definition E.1.** A function of two variables $\Theta : \mathbb{R}^d \times \mathbb{R}^d \to \mathbb{R}$ is doubly asymptotically positive $n$-homogeneous if

$$\lim_{\lambda, \lambda' \to \infty} \frac{\Theta(\lambda \boldsymbol{x}, \lambda' \boldsymbol{x}')}{(\lambda \lambda')^n} = \hat{\hat{\Theta}}(\boldsymbol{x}, \boldsymbol{x}') \quad \forall \boldsymbol{x}, \boldsymbol{x}' \in \mathbb{R}^d,$$

where $\hat{\hat{\Theta}}(\boldsymbol{x}, \boldsymbol{x}')$ is positive $n$-homogeneous in both arguments, that is, $\hat{\hat{\Theta}}(\lambda \boldsymbol{x}, \lambda' \boldsymbol{x}') = \lambda \lambda' \hat{\hat{\Theta}}(\boldsymbol{x}, \boldsymbol{x}') \ \forall \boldsymbol{x}, \boldsymbol{x}' \in \mathbb{R}^d, \lambda, \lambda' > 0$.

**Corollary E.1.** *The NTK of a neural network with fully-connected layers and nonlinearities satisfying Assumption 3.1 and Assumption C.1, is itself doubly asymptotically positive $n^L$-homogeneous. That is, $\forall \boldsymbol{x}, \boldsymbol{x}' \in \mathbb{R}^{n_0}$,*

$$\lim_{\lambda, \lambda' \to \infty} \frac{\Theta(\lambda \boldsymbol{x}, \lambda' \boldsymbol{x}')}{(\lambda \lambda')^{n^L}} = \hat{\hat{\Theta}}(\boldsymbol{x}, \boldsymbol{x}') \quad \forall \boldsymbol{x}, \boldsymbol{x}' \in \mathbb{R}^{n_0}$$

*where $\hat{\hat{\Theta}}(\boldsymbol{x}, \boldsymbol{x}')$ is positive $n^L$-homogeneous with respect to both arguments, that is,*

$$\hat{\hat{\Theta}}(\lambda \boldsymbol{x}, \boldsymbol{x}') = \hat{\hat{\Theta}}(\boldsymbol{x}, \lambda \boldsymbol{x}') = \lambda^{n^L} \hat{\hat{\Theta}}(\boldsymbol{x}, \boldsymbol{x}') \quad \forall \boldsymbol{x}, \boldsymbol{x}' \in \mathbb{R}^{n_0}, \lambda > 0, \quad and$$

$$\hat{\hat{\Theta}}(\lambda \boldsymbol{x}, \lambda' \boldsymbol{x}') = (\lambda \lambda')^{n^L} \hat{\hat{\Theta}}(\boldsymbol{x}, \boldsymbol{x}') \quad \forall \boldsymbol{x}, \boldsymbol{x}' \in \mathbb{R}^{n_0}, \lambda, \lambda' > 0.$$

*Proof.* From Proposition E.2, we have that

$$
\begin{aligned}
\lim_{\lambda, \lambda' \to \infty} \frac{\Theta(\boldsymbol{x}, \lambda' \boldsymbol{x}')}{(\lambda \lambda')^{n^L}} &= \lim_{\lambda, \lambda' \to \infty} \frac{\left(\frac{1}{n_0} \|\lambda \boldsymbol{x}\| \|\lambda' \boldsymbol{x}'\|\right)^{n^L} \bar{\kappa}(\lambda \boldsymbol{x}, \lambda' \boldsymbol{x}'; \sigma_b^2)}{(\lambda \lambda')^{n^L}} + \frac{\bar{R}(\lambda \boldsymbol{x}, \lambda' \boldsymbol{x}'; \sigma_b^2)}{(\lambda \lambda')^{n^L}} \\
&= \lim_{\lambda, \lambda' \to \infty} \frac{(\lambda \lambda')^{n^L}}{(\lambda \lambda')^{n^L}} \left(\frac{1}{n_0} \|\boldsymbol{x}\| \|\boldsymbol{x}'\|\right)^{n^L} \bar{\kappa}(\lambda \boldsymbol{x}, \lambda' \boldsymbol{x}'; \sigma_b^2) \\
&= \left(\frac{1}{n_0} \|\boldsymbol{x}\| \|\boldsymbol{x}'\|\right)^{n^L} \left(\lim_{\lambda, \lambda' \to \infty} \bar{\kappa}(\lambda \boldsymbol{x}, \lambda' \boldsymbol{x}'; \sigma_b^2)\right) \quad \forall \boldsymbol{x}, \boldsymbol{x}' \in \mathbb{R}^{n_0}.
\end{aligned}
$$

We thus define

$$\hat{\hat{\Theta}}(\boldsymbol{x}, \boldsymbol{x}') = \lim_{\lambda, \lambda' \to \infty} \frac{\Theta(\boldsymbol{x}, \lambda' \boldsymbol{x}')}{(\lambda \lambda')^{n^L}} \tag{40}$$

Now, note from Equation and 36 Equation38 that $\bar{\kappa}(\boldsymbol{x}, \boldsymbol{x}')$ is composed of a summation of products of compositions of $\kappa(\cdot)$, and $\dot{\kappa}(\cdot)$, which both only act on pre-activation correlations, which in turn are given in Equation 27 by $\rho^{(h)}(\boldsymbol{x}, \boldsymbol{x}') = \frac{\Sigma^{(h)}(\boldsymbol{x}, \boldsymbol{x}')}{\sqrt{\Sigma^{(h)}(\boldsymbol{x}, \boldsymbol{x})\Sigma^{(h)}(\boldsymbol{x}', \boldsymbol{x}')}}$. In the limiting case,

$$\lim_{\lambda,\lambda'\to\infty} \rho^{(h)}(\lambda\boldsymbol{x},\lambda'\boldsymbol{x}') = \lim_{\lambda,\lambda'\to\infty} \frac{\Sigma^{(h)}(\lambda\boldsymbol{x},\lambda'\boldsymbol{x}')}{\sqrt{\Sigma^{(h)}(\lambda\boldsymbol{x},\lambda\boldsymbol{x})\Sigma^{(h)}(\lambda'\boldsymbol{x}',\lambda'\boldsymbol{x}')}}$$

$$= \lim_{\lambda,\lambda'\to\infty} \frac{\left(\frac{1}{n_0}\|\lambda\boldsymbol{x}\|\,\|\lambda'\boldsymbol{x}'\|\right)^{n^h}\kappa\left(\rho^{(h-1)}(\lambda\boldsymbol{x},\lambda'\boldsymbol{x}')\right) + R^{(h)}(\lambda\boldsymbol{x},\lambda'\boldsymbol{x}')}{\sqrt{\left(\left(\frac{1}{n_0}\|\lambda\boldsymbol{x}\|\right)^{2n^h} + R^{(h)}(\lambda\boldsymbol{x},\lambda\boldsymbol{x})\right)\left(\left(\frac{1}{n_0}\|\lambda'\boldsymbol{x}'\|\right)^{2n^h} + R^{(h)}(\lambda'\boldsymbol{x}',\lambda'\boldsymbol{x}')\right)}}$$

$$= \lim_{\lambda,\lambda'\to\infty} \frac{(\lambda\lambda')^{n^h}\left(\frac{1}{n_0}\right)^{n^h}(\|\boldsymbol{x}\|\,\|\boldsymbol{x}'\|)^{n^h}\kappa\left(\rho^{(h-1)}(\lambda\boldsymbol{x},\lambda'\boldsymbol{x}')\right) + R^{(h)}(\lambda\boldsymbol{x},\lambda'\boldsymbol{x}')}{(\lambda\lambda')^{n^L}\left(\frac{1}{n_0}\right)^{n^h}\sqrt{\left(\|\boldsymbol{x}\|^{2n^h} + n_0^{2n^h}\frac{R^{(h)}(\lambda\boldsymbol{x},\lambda\boldsymbol{x})}{\lambda^{2n^h}}\right)\left(\|\boldsymbol{x}'\|^{2n^h} + n_0^{2n^h}\frac{R^{(h)}(\lambda'\boldsymbol{x}',\lambda'\boldsymbol{x}')}{(\lambda')^{2n^h}}\right)}}$$

$$= \lim_{\lambda,\lambda'\to\infty} \frac{(\|\boldsymbol{x}\|\,\|\boldsymbol{x}'\|)^{n^h}\kappa\left(\rho^{(h-1)}(\lambda\boldsymbol{x},\lambda'\boldsymbol{x}')\right)}{\sqrt{(\|\boldsymbol{x}\|\,\|\boldsymbol{x}'\|)^{2n^h}}} + \frac{R^{(h)}(\lambda\boldsymbol{x},\lambda'\boldsymbol{x}')}{(\lambda\lambda')^{n^h}}$$

$$= \frac{(\|\boldsymbol{x}\|\,\|\boldsymbol{x}'\|)^{n^h}}{(\|\boldsymbol{x}\|\,\|\boldsymbol{x}'\|)^{n^h}}\left(\lim_{\lambda,\lambda'\to\infty}\kappa\left(\rho^{(h-1)}(\lambda\boldsymbol{x},\lambda'\boldsymbol{x}')\right)\right)$$

$$= \kappa\left(\lim_{\lambda,\lambda'\to\infty}\rho^{(h-1)}(\lambda\boldsymbol{x},\lambda'\boldsymbol{x}')\right)$$

which follows similar reasoning to that used in the proof of Proposition E.2, with the final transition assuming $\kappa(\cdot)$ is continuous on $[-1,1]$ and the limit for $\rho^{(h-1)}$ exists. Now, the initial limit correlation is given by

$$\lim_{\lambda,\lambda'\to\infty}\rho^{(0)}(\lambda\boldsymbol{x},\lambda'\boldsymbol{x}') = \lim_{\lambda,\lambda'\to\infty}\mathcal{S}\left(\lambda\boldsymbol{x},\lambda'\boldsymbol{x}';\sigma_b^2\right)$$

$$= \lim_{\lambda,\lambda'\to\infty}\frac{\lambda\lambda'\frac{1}{n_0}\boldsymbol{x}^\top\boldsymbol{x}' + \sigma_b^2}{\sqrt{\left(\frac{1}{n_0}\|\lambda\boldsymbol{x}\|^2 + \sigma_b^2\right)\left(\frac{1}{n_0}\|\lambda'\boldsymbol{x}'\|^2 + \sigma_b^2\right)}}$$

$$= \lim_{\lambda,\lambda'\to\infty}\frac{\lambda\lambda'\boldsymbol{x}^\top\boldsymbol{x}'}{\lambda\lambda'\|\boldsymbol{x}\|\,\|\boldsymbol{x}'\|} + \frac{n_0\sigma_b^2}{\lambda\lambda'\|\boldsymbol{x}\|\,\|\boldsymbol{x}'\|}$$

$$= \hat{\boldsymbol{x}}^\top\hat{\boldsymbol{x}}' \quad \forall \boldsymbol{x},\boldsymbol{x}' \in \mathbb{R}^{n_0}\setminus\boldsymbol{0}$$

where $\hat{\boldsymbol{x}}$ and $\hat{\boldsymbol{x}}'$ are unit vectors in the directions of $\boldsymbol{x}$ and $\boldsymbol{x}'$ respectively. Note that when $\boldsymbol{x}=\boldsymbol{x}'=\boldsymbol{0}$, this expression is instead given by

$$\rho^{(0)}(\boldsymbol{0},\boldsymbol{0}) = \frac{\sigma_b^2}{\sqrt{(\sigma_b^2)(\sigma_b^2)}}$$

$$= 1.$$

In either case, the limit exists and depends only on $\hat{\boldsymbol{x}}$ and $\hat{\boldsymbol{x}}'$ (which if are equal to $\boldsymbol{0}$ indicate the adoption of the alternative case). Hence by induction, $\lim_{\lambda,\lambda'\to\infty}\rho^{(h)}(\boldsymbol{x},\boldsymbol{x}')$ exists $\forall h=0,\ldots,L$ and depends only on $\hat{\boldsymbol{x}}$ and $\hat{\boldsymbol{x}}'$. To solidify notation, we define

$$\hat{\rho}^{(h)}(\hat{\boldsymbol{x}},\hat{\boldsymbol{x}}') = \lim_{\lambda,\lambda'\to\infty}\rho^{(h)}(\lambda\boldsymbol{x},\lambda'\boldsymbol{x}') \tag{41}$$

$$\hat{\rho}^{(h+1)}(\hat{\boldsymbol{x}},\hat{\boldsymbol{x}}') = \kappa\left(\hat{\rho}^{(h)}(\hat{\boldsymbol{x}},\hat{\boldsymbol{x}}')\right)$$

$$\hat{\rho}^{(0)}(\hat{\boldsymbol{x}},\hat{\boldsymbol{x}}') = \begin{cases}\hat{\boldsymbol{x}}^\top\hat{\boldsymbol{x}}' & \hat{\boldsymbol{x}},\hat{\boldsymbol{x}}' \neq 0 \\ 1 & \hat{\boldsymbol{x}},\hat{\boldsymbol{x}}' = 0\end{cases}, \tag{42}$$

where $\hat{\boldsymbol{x}}$ and $\hat{\boldsymbol{x}}'$ are unit vectors in the direction of $\boldsymbol{x}$ and $\boldsymbol{x}'$ respectively. By identical logic, the same can be said about $\lim_{\lambda,\lambda'\to\infty}\dot{\kappa}\left(\rho^{(h-1)}(\lambda\boldsymbol{x},\lambda'\boldsymbol{x}')\right)$. As both $\kappa$ and $\dot{\kappa}$ have range $\subseteq[-1,1]$, this limiting behaviour carries over to

$$\hat{\bar{\kappa}}\left(\hat{\boldsymbol{x}},\hat{\boldsymbol{x}}'\right) = \lim_{\lambda,\lambda'\to\infty} \bar{\kappa}\left(\lambda\boldsymbol{x},\lambda'\boldsymbol{x}'\right). \tag{43}$$

Thus,

$$\hat{\Theta}\left(\lambda\boldsymbol{x},\lambda'\boldsymbol{x}';\sigma_b^2\right) = \left(\frac{1}{n_0}\|\lambda\boldsymbol{x}\|\,\|\lambda'\boldsymbol{x}'\|\right)^{n^L}\hat{\bar{\kappa}}\left(\hat{\lambda\boldsymbol{x}},\hat{\lambda'\boldsymbol{x}'}\right)$$

$$= (\lambda\lambda')^{n^L}\left(\frac{1}{n_0}\|\boldsymbol{x}\|\,\|\boldsymbol{x}'\|\right)^{n^L}\hat{\bar{\kappa}}\left(\hat{\boldsymbol{x}},\hat{\boldsymbol{x}}'\right)$$

$$= (\lambda\lambda')^{n^L}\hat{\Theta}\left(\boldsymbol{x},\boldsymbol{x}'\right) \quad \forall\boldsymbol{x},\boldsymbol{x}'\in\mathbb{R}^{n_0},\lambda,\lambda'>0,$$

where the $\hat{\lambda\boldsymbol{x}}$ and $\hat{\lambda'\boldsymbol{x}'}$ are abuses of notation referencing unit vectors in the direction of $\lambda\boldsymbol{x}$ and $\lambda'\boldsymbol{x}'$ respectively, and the penultimate transition follows from noting that for nonzero $\boldsymbol{x}$ and $\lambda$, $\hat{\lambda\boldsymbol{x}}$ and $\hat{\lambda'\boldsymbol{x}'}$ do not depend on $\lambda$ or $\lambda'$ for $\lambda,\lambda'>0$, and in the case where either are zero the change in behaviour is hidden by a multiplication by zero. Hence, $\hat{\Theta}\left(\boldsymbol{x},\boldsymbol{x}'\right)$ is positive $n^L$-homogeneous with respect to its first argument by definition, and also with respect to the second argument by symmetry. Therefore $\Theta\left(\boldsymbol{x},\boldsymbol{x}'\right)$ is doubly asymptotically positive $n^L$-homogeneous by definition.

$\square$

**Corollary E.2.** *The NTK of a neural network with fully-connected layers and positive $n$-homogeneous nonlinearities satisfying Assumption 3.1, is itself asymptotically positive $n^L$-homogeneous with respect to both arguments. That is, $\forall\boldsymbol{x},\boldsymbol{x}'\in\mathbb{R}^{n_0}$,*

$$\lim_{\lambda\to\infty}\frac{\Theta\left(\lambda\boldsymbol{x},\boldsymbol{x}'\right)}{\lambda^{n^L}} = \lim_{\lambda\to\infty}\frac{\Theta\left(\boldsymbol{x}',\lambda\boldsymbol{x}\right)}{\lambda^{n^L}} = \hat{\Theta}\left(\boldsymbol{x},\boldsymbol{x}'\right) \quad \forall\boldsymbol{x},\boldsymbol{x}'\in\mathbb{R}^{n_0}$$

*where $\hat{\Theta}\left(\boldsymbol{x},\boldsymbol{x}'\right)$ is positive $n^L$-homogeneous with respect to its first argument, that is,*

$$\hat{\Theta}\left(\lambda\boldsymbol{x},\boldsymbol{x}'\right) = \lambda^{n^L}\hat{\Theta}\left(\boldsymbol{x},\boldsymbol{x}'\right) \quad \forall\boldsymbol{x},\boldsymbol{x}'\in\mathbb{R}^{n_0},\lambda>0.$$

*Proof.* Consider Lemma E.2 for the case of positive $n$-homogeneous $\phi(\cdot)$. Clearly, with reference to Equation 14, the non-homogeneous part $\tilde{\phi}(x)=0\ \forall x\in\mathbb{R}$, and as such $\phi(x)=\hat{\phi}(x)\ \forall x\in\mathbb{R}$. It then follows from application of these conditions to Equation 15 that

$$\tilde{R}(x,y;\sigma_u,\sigma_b,\rho) = 0 \quad \forall x,y\in\mathbb{R},\sigma_u,\sigma_v>0,\rho\in[-1,1], \tag{44}$$

and thus from Equation 18, Equation 16, and Equation 21,

$$\hat{R}\left(\sigma_u,\sigma_v,\rho\right) = 0 \hspace{3cm} \forall\sigma_u,\sigma_v>0,\rho\in[-1,1] \tag{45}$$

$$\mathbb{E}_{(x,y)}\left[\phi(\sigma_u x)\phi(\sigma_v y)\right] = (\sigma_u\sigma_v)^n\hat{\kappa}(\rho) \hspace{1cm} \forall\sigma_u,\sigma_v>0,\rho\in[-1,1] \tag{46}$$

$$R\left(\sigma_u,\sigma_v,\rho\right) = 0 \hspace{3cm} \forall\sigma_u,\sigma_v>0,\rho\in[-1,1] \tag{47}$$

We now consider Proposition E.2 for the case of positive $n$-homogeneous $\phi(\cdot)$. Recall the recursion for $R^{(h)}\left(\boldsymbol{x},\boldsymbol{x}'\right)$, Recursion 30:

$$R^{(h+1)}\left(\boldsymbol{x}, \boldsymbol{x}'\right) = \left(\sum_{k=0}^{n-1} \binom{n}{k} \left(\frac{1}{n_0} \|\boldsymbol{x}\| \|\boldsymbol{x}'\|\right)^{kn^h} R^{(h)}\left(\boldsymbol{x}, \boldsymbol{x}'\right)^{(n-k)}\right)$$
$$+ R\left(\sqrt{\frac{1}{n_0}\|\boldsymbol{x}\|^{2n^h} + R^{(h)}(\boldsymbol{x}, \boldsymbol{x})}, \sqrt{\frac{1}{n_0}\|\boldsymbol{x}'\|^{2n^h} + R^{(h)}\left(\boldsymbol{x}', \boldsymbol{x}'\right)}, \rho^{(h)}\left(\boldsymbol{x}, \boldsymbol{x}'\right)\right) + \sigma_b^2$$
$$= \left(\sum_{k=0}^{n-1} \binom{n}{k}\left(\frac{1}{n_0}\|\boldsymbol{x}\|\|\boldsymbol{x}'\|\right)^{kn^h} R^{(h)}\left(\boldsymbol{x}, \boldsymbol{x}'\right)^{(n-k)}\right) + \sigma_b^2,$$
$$R^{(0)} = \sigma_b^2.$$

Under these conditions,

$$\lim_{\lambda \to \infty} \frac{R^{(h+1)}\left(\lambda\boldsymbol{x}, \boldsymbol{x}'\right)}{\lambda^{n^{h+1}}} = \lim_{\lambda \to \infty} \frac{\left(\sum_{k=0}^{n-1} \binom{n}{k}\left(\frac{1}{n_0}\|\lambda\boldsymbol{x}\|\|\boldsymbol{x}'\|\right)^{kn^h} R^{(h)}\left(\lambda\boldsymbol{x}, \boldsymbol{x}'\right)^{(n-k)}\right) + \sigma_b^2}{\lambda^{n^{h+1}}}$$
$$= \lim_{\lambda \to \infty} \sum_{k=0}^{n-1} \binom{n}{k}\left(\frac{1}{n_0}\|\boldsymbol{x}\|\|\boldsymbol{x}'\|\right)^{kn^h} \frac{R^{(h)}\left(\lambda\boldsymbol{x}, \boldsymbol{x}'\right)^{(n-k)}}{\lambda^{n^h(n-k)}}$$
$$= \sum_{k=0}^{n-1} \binom{n}{k}\left(\frac{1}{n_0}\|\boldsymbol{x}\|\|\boldsymbol{x}'\|\right)^{kn^h} \lim_{\lambda \to \infty} \left(\frac{R^{(h)}\left(\lambda\boldsymbol{x}, \boldsymbol{x}'\right)}{\lambda^{n^h}}\right)^{(n-k)}$$
$$= \sum_{k=0}^{n-1} \binom{n}{k}\left(\frac{1}{n_0}\|\boldsymbol{x}\|\|\boldsymbol{x}'\|\right)^{kn^h} \left(\lim_{\lambda \to \infty} \frac{R^{(h)}\left(\lambda\boldsymbol{x}, \boldsymbol{x}'\right)}{\lambda^{n^h}}\right)^{(n-k)},$$

where the final transition follows from direct application of the algebraic limit theorem. Considering the case of $n = 0$;

$$\lim_{\lambda \to \infty} \frac{R^{(0)}\left(\lambda\boldsymbol{x}, \boldsymbol{x}'\right)}{\lambda^{n^0}} = \lim_{\lambda \to \infty} \frac{\sigma_b^2}{\lambda}$$
$$= 0 \quad \forall \boldsymbol{x}, \boldsymbol{x}' \in \mathbb{R}^{n_0},$$

hence by induction;

$$\lim_{\lambda \to \infty} \frac{R^{(h)}\left(\lambda\boldsymbol{x}, \boldsymbol{x}'\right)}{\lambda^{n^h}} = 0 \quad \forall \boldsymbol{x}, \boldsymbol{x}' \in \mathcal{X}, h = 0, \ldots L.$$

By identical derivation,

$$\lim_{\lambda \to \infty} \frac{\dot{R}^{(h)}\left(\lambda\boldsymbol{x}, \boldsymbol{x}'\right)}{\lambda^{n^h - n^{h-1}}} = 0 \quad \forall \boldsymbol{x}, \boldsymbol{x}' \in \mathbb{R}^{n_0}, h = 0, \ldots L.$$

Again considering the contribution of the $h$th layer of the NTK and collecting lower order terms into $\bar{R}^{(h)}\left(\boldsymbol{x}, \boldsymbol{x}'\right)$, it is clear that

$$\lim_{\lambda \to \infty} \frac{\bar{R}^{(h)}\left(\lambda\boldsymbol{x}, \boldsymbol{x}'\right)}{\lambda^{n^L}} = 0 \quad \forall \boldsymbol{x}, \boldsymbol{x}' \in \mathbb{R}^{n_0}, h = 0, \ldots, L.$$

Note that this a stronger statement than the equivalent in the proof of Proposition E.2, and owes to the fact that under these stronger assumptions $R^{(h)}$ vanishes under scaling in one argument only, rather than requiring both as was previously the case. It follows that

$$\lim_{\lambda \to \infty} \frac{\bar{R}\left(\lambda \boldsymbol{x}, \boldsymbol{x}'; \sigma_b^2\right)}{\lambda^{n^L}} = \lim_{\lambda \to \infty} \frac{\sum_{h=0}^{L} R^{(h)}\left(\lambda \boldsymbol{x}, \boldsymbol{x}'\right)}{\lambda^{n^L}}$$

$$= \sum_{h=0}^{L} \lim_{\lambda \to \infty} \frac{R^{(h)}\left(\lambda \boldsymbol{x}, \boldsymbol{x}'\right)}{\lambda^{n^L}}$$

$$= 0 \quad \forall \boldsymbol{x}, \boldsymbol{x}' \in \mathbb{R}^{n_0}.$$

Following now the proof of Corollary E.1;

$$\lim_{\lambda \to \infty} \frac{\Theta\left(\lambda \boldsymbol{x}, \boldsymbol{x}'\right)}{\lambda^{n^L}} = \lim_{\lambda \to \infty} \frac{\left(\frac{1}{n_0} \|\lambda \boldsymbol{x}\| \|\boldsymbol{x}'\|\right)^{n^L} \bar{\kappa}\left(\lambda \boldsymbol{x}, \boldsymbol{x}'; \sigma_b^2\right)}{\lambda^{n^L}} + \frac{R^{(h)}\left(\lambda \boldsymbol{x}, \boldsymbol{x}'\right)}{\lambda^{n^L}}$$

$$= \lim_{\lambda \to \infty} \frac{\lambda^{n^L}}{\lambda^{n^L}} \left(\frac{1}{n_0} \|\boldsymbol{x}\| \|\boldsymbol{x}'\|\right)^{n^L} \bar{\kappa}\left(\lambda \boldsymbol{x}, \boldsymbol{x}'; \sigma_b^2\right)$$

$$= \left(\frac{1}{n_0} \|\boldsymbol{x}\| \|\boldsymbol{x}'\|\right)^{n^L} \left(\lim_{\lambda \to \infty} \bar{\kappa}\left(\lambda \boldsymbol{x}, \boldsymbol{x}'; \sigma_b^2\right)\right).$$

For convenience, we define the single-argument limit NTK

$$\hat{\Theta}\left(\boldsymbol{x}, \boldsymbol{x}'\right) = \lim_{\lambda \to \infty} \frac{\Theta\left(\lambda \boldsymbol{x}, \boldsymbol{x}'\right)}{\lambda^{n^L}}. \tag{48}$$

As before with reference to Equation 27, considering the single-argument limiting case for the layer-$h$ correlation,

$$\lim_{\lambda \to \infty} \rho^{(h)}(\lambda \boldsymbol{x}, \boldsymbol{x}') = \lim_{\lambda \to \infty} \frac{\Sigma^{(h)}(\lambda \boldsymbol{x}, \boldsymbol{x}')}{\sqrt{\Sigma^{(h)}(\lambda \boldsymbol{x}, \lambda \boldsymbol{x})\Sigma^{(h)}\left(\boldsymbol{x}', \boldsymbol{x}'\right)}}$$

$$= \lim_{\lambda \to \infty} \frac{\left(\frac{1}{n_0} \|\lambda \boldsymbol{x}\| \|\boldsymbol{x}'\|\right)^{n^h} \kappa\left(\rho^{(h-1)}(\lambda \boldsymbol{x}, \boldsymbol{x}')\right) + R^{(h)}(\lambda \boldsymbol{x}, \boldsymbol{x}')}{\sqrt{\left(\left(\frac{1}{n_0} \|\lambda \boldsymbol{x}\|\right)^{2n^h} + R^{(h)}(\lambda \boldsymbol{x}, \lambda \boldsymbol{x})\right)\Sigma^{(h)}\left(\boldsymbol{x}', \boldsymbol{x}'\right)}}$$

$$= \lim_{\lambda \to \infty} \frac{\lambda^{n^h}\left(\frac{1}{n_0}\right)^{n^h}(\|\boldsymbol{x}\| \|\boldsymbol{x}'\|)^{n^h} \kappa\left(\rho^{(h-1)}(\lambda \boldsymbol{x}, \boldsymbol{x}')\right) + R^{(h)}(\lambda \boldsymbol{x}, \boldsymbol{x}')}{\lambda^{n^L}\left(\frac{1}{n_0}\right)^{n^h}\sqrt{\left(\|\boldsymbol{x}\|^{2n^h} + n_0^{2n^h}\frac{R^{(h)}(\lambda \boldsymbol{x}, \lambda \boldsymbol{x})}{\lambda^{2n^h}}\right)\Sigma^{(h)}\left(\boldsymbol{x}', \boldsymbol{x}'\right)}}$$

$$= \lim_{\lambda \to \infty} \frac{(\|\boldsymbol{x}\| \|\boldsymbol{x}'\|)^{n^h} \kappa\left(\rho^{(h-1)}(\lambda \boldsymbol{x}, \boldsymbol{x}')\right)}{\sqrt{\|\boldsymbol{x}\|^{2n^h}\Sigma^{(h)}\left(\boldsymbol{x}', \boldsymbol{x}'\right)}} + \frac{R^{(h)}(\lambda \boldsymbol{x}, \boldsymbol{x}')}{\lambda^{n^h}}$$

$$= \frac{(\|\boldsymbol{x}\| \|\boldsymbol{x}'\|)^{n^h}}{\|\boldsymbol{x}\|^{n^h}\sqrt{\Sigma^{(h)}\left(\boldsymbol{x}', \boldsymbol{x}'\right)}}\left(\lim_{\lambda \to \infty} \kappa\left(\rho^{(h-1)}(\lambda \boldsymbol{x}, \boldsymbol{x}')\right)\right)$$

$$= \frac{(\|\boldsymbol{x}'\|)^{n^h}}{\sqrt{\Sigma^{(h)}\left(\boldsymbol{x}', \boldsymbol{x}'\right)}}\kappa\left(\lim_{\lambda \to \infty} \rho^{(h-1)}(\lambda \boldsymbol{x}, \boldsymbol{x}')\right)$$

which follows similar reasoning to that used in the proof of Proposition E.2, with the final transition assuming $\kappa(\cdot)$ is continuous on $[-1, 1]$ and the limit for $\rho^{(h-1)}$ exists as before. Now, the initial limit correlation is given by

$$\lim_{\lambda \to \infty} \rho^{(0)}(\lambda \boldsymbol{x}, \boldsymbol{x}') = \lim_{\lambda \to \infty} \mathcal{S}\left(\lambda \boldsymbol{x}, \boldsymbol{x}'; \sigma_b^2\right)$$

$$= \lim_{\lambda \to \infty} \frac{\lambda \frac{1}{n_0} \boldsymbol{x}^\top \boldsymbol{x}' + \sigma_b^2}{\sqrt{\left(\frac{1}{n_0} \|\lambda \boldsymbol{x}\|^2 + \sigma_b^2\right)\left(\frac{1}{n_0} \|\boldsymbol{x}'\|^2 + \sigma_b^2\right)}}$$

$$= \lim_{\lambda \to \infty} \frac{\lambda \boldsymbol{x}^\top \boldsymbol{x}'}{\lambda \|\boldsymbol{x}\| \sqrt{\|\boldsymbol{x}'\|^2 + n_0 \sigma_b^2}} + \frac{n_0 \sigma_b^2}{\lambda \|\boldsymbol{x}\| \sqrt{\|\boldsymbol{x}'\|^2 + n_0 \sigma_b^2}}$$

$$= \frac{\hat{\boldsymbol{x}}^\top \boldsymbol{x}'}{\sqrt{\|\boldsymbol{x}'\|^2 + n_0 \sigma_b^2}} \quad \forall \boldsymbol{x} \in \mathbb{R}^{n_0} \setminus \boldsymbol{0}, \boldsymbol{x}' \in \mathbb{R}^{n_0}$$

where $\hat{\boldsymbol{x}}$ is a unit vector in the direction of $\boldsymbol{x}$. Note that when $\boldsymbol{x} = \boldsymbol{0}$, this expression is instead given by

$$\rho^{(0)}(\boldsymbol{0}, \boldsymbol{x}') = \frac{n_0 \sigma_b^2}{\sqrt{(n_0 \sigma_b^2)\left(\|\boldsymbol{x}'\|^2 + n_0 \sigma_b^2\right)}}$$

$$= \frac{\sigma_b}{\sqrt{\frac{1}{n_0} \|\boldsymbol{x}'\|^2 + \sigma_b^2}} \quad \forall \boldsymbol{x}' \in \mathbb{R}^{n_0}.$$

In either case, the limit exists and depends only on $\hat{\boldsymbol{x}}$ and $\boldsymbol{x}'$ (the former of which if equal to $\boldsymbol{0}$ indicate the adoption of the alternative case). Hence by induction, $\lim_{\lambda \to \infty} \rho^{(h)}(\lambda \boldsymbol{x}, \boldsymbol{x}')$ exists $\forall h = 0, \ldots, L$ and depends only on $\hat{\boldsymbol{x}}$ and $\boldsymbol{x}'$. By identical logic, the same can be said about $\lim_{\lambda \to \infty} \dot{\kappa}\left(\rho^{(h-1)}(\lambda \boldsymbol{x}, \boldsymbol{x}')\right)$. To solidify notation, we then define

$$\hat{\rho}^{(h)}(\hat{\boldsymbol{x}}, \boldsymbol{x}') = \lim_{\lambda \to \infty} \rho^{(h)}(\lambda \boldsymbol{x}, \boldsymbol{x}'), \tag{49}$$

$$\hat{\rho}^{(h+1)}(\hat{\boldsymbol{x}}, \boldsymbol{x}') = \frac{(\|\boldsymbol{x}'\|)^{n^h}}{\sqrt{\Sigma^{(h)}(\boldsymbol{x}', \boldsymbol{x}')}} \kappa\left(\hat{\rho}^{(h)}(\hat{\boldsymbol{x}}, \boldsymbol{x}')\right)$$

$$\hat{\rho}^{(0)}(\hat{\boldsymbol{x}}, \boldsymbol{x}') = \begin{cases} \frac{\hat{\boldsymbol{x}}^\top \boldsymbol{x}'}{\sqrt{\|\boldsymbol{x}'\|^2 + n_0 \sigma_b^2}} & \boldsymbol{x} \in \mathbb{R}^{n_0} \setminus \boldsymbol{0} \\ \frac{\sigma_b}{\sqrt{\frac{1}{n_0} \|\boldsymbol{x}'\|^2 + \sigma_b^2}} & \boldsymbol{x} = \boldsymbol{0} \end{cases} \tag{50}$$

As before, since both $\kappa$ and $\dot{\kappa}$ have range $\subseteq [-1, 1]$, this limiting behaviour carries over to

$$\hat{\bar{\kappa}}(\hat{\boldsymbol{x}}, \boldsymbol{x}') = \lim_{\lambda \to \infty} \bar{\kappa}(\lambda \boldsymbol{x}, \boldsymbol{x}'). \tag{51}$$

Thus,

$$\hat{\Theta}(\lambda \boldsymbol{x}, \boldsymbol{x}') = \left(\frac{1}{n_0} \lambda \|\boldsymbol{x}\| \|\boldsymbol{x}'\|\right)^{n^L} \hat{\bar{\kappa}}\left(\hat{\lambda \boldsymbol{x}}, \boldsymbol{x}'; \sigma_b^2\right)$$

$$= \lambda^{n^L} \left(\frac{1}{n_0} \|\boldsymbol{x}\| \|\boldsymbol{x}'\|\right)^{n^L} \hat{\bar{\kappa}}\left(\hat{\boldsymbol{x}}, \boldsymbol{x}'; \sigma_b^2\right)$$

$$= \lambda^{n^L} \hat{\Theta}(\boldsymbol{x}, \boldsymbol{x}') \quad \forall \boldsymbol{x}, \boldsymbol{x}' \in \mathbb{R}^{n_0}, \lambda > 0,$$

where $\hat{\lambda \boldsymbol{x}}$ is an abuse of notation representing a unit vector in the direction of $\lambda \boldsymbol{x}$, and the penultimate transition follows from noting that $\hat{\lambda \boldsymbol{x}}$ does not depend on $\lambda$ for all $\boldsymbol{x} \in \mathbb{R}^{n_0}, \lambda > 0$, and in the

case where either are zero the change in behaviour is hidden by a multiplication by zero. Hence, $\hat{\Theta}\left(\boldsymbol{x}, \boldsymbol{x}'\right)$ is positive $n^L$-homogeneous with respect to its first argument by definition (and also with respect to the second argument by symmetry). Therefore, $\Theta\left(\boldsymbol{x}, \boldsymbol{x}'; \sigma_b^2\right)$ is asymptotically positive $n^L$-homogeneous in its first argument (and indeed the second) by definition.

$\square$

**Lemma E.3.** *The probabilists' Hermite polynomials satisfy*

$$\mathbb{E}_{(x,y)}\left[H_n(x)H_m(y)\right] = n!\rho^n\delta_{nm}, \quad where$$
$$(x, y) \sim \mathcal{N}\left(\mathbf{0}, \begin{bmatrix} 1 & \rho \\ \rho & 1 \end{bmatrix}\right).$$

*Proof.* We begin by noting that the differential joint density function of the given $x$ and $y$ is

$$
\begin{aligned}
df_{X,Y}(x, y) &= \frac{1}{2\pi\sqrt{1 - \rho^2}} \exp\left(-\frac{x^2 + y^2 - 2\rho xy}{2\left(1 - \rho^2\right)}\right) \\
&= \sum_{n=0}^{\infty} \frac{\rho^n}{n!} H_n(x) H_n(y) d\gamma(x) d\gamma(y),
\end{aligned}
$$

where $\gamma$ is the standard normal measure and the second line is a direct statement of Mehler's formula. Hence,

$$
\begin{aligned}
\mathbb{E}_{(x,y)}\left[H_n(x)H_m(y)\right] &= \int_{\mathbb{R}^2} H_n(x) H_m(y) df_{X,Y}(x, y) \\
&= \int_{\mathbb{R}^2} H_n(x) H_m(y) \sum_{l=0}^{\infty} \frac{\rho^l}{l!} H_l(x) H_l(y) d\gamma(x) d\gamma(y) \\
&= \sum_{l=0}^{\infty} \frac{\rho^l}{l!} \int_{\mathbb{R}^2} H_n(x) H_m(y) H_l(x) H_l(y) d\gamma(x) d\gamma(y) \\
&= \sum_{l=0}^{\infty} \frac{\rho^l}{l!} \int_{\mathbb{R}} H_n(x) H_l(x) d\gamma(x) \int_{\mathbb{R}} H_m(y) H_l(y) d\gamma(y) \\
&= \sum_{l=0}^{\infty} \frac{\rho^l}{l!} \left(n!\delta_{nl}\right)\left(m!\delta_{ml}\right) \\
&= \rho^n n!\delta_{nm},
\end{aligned}
$$

where the second transition follows from the dominated convergence theorem, the penultimate transition follows from the orthogonality of the probabilist's Hermite polynomials with respect to the standard normal measure $\gamma$.

$\square$

**Lemma E.4.** *For all nonlinearities $\phi : \mathbb{R} \to \mathbb{R}$ satisfying Assumption 3.1 and Assumption C.1, the associated $\kappa(\rho)$ and $\dot{\kappa}(\rho)$ defined as in Lemma E.2 exist and satisfy*

$$
\begin{aligned}
\kappa(\rho) &> 0 \quad \forall \rho > 0, \quad and \\
\dot{\kappa}(\rho) &> 0 \quad \forall \rho > 0.
\end{aligned}
$$

*Proof.* First, we recall the definition of $\hat{\kappa}(\rho)$:

$$\hat{\kappa}(\rho) = \mathbb{E}_{(x,y)} \left[ \hat{\phi}(x)\hat{\phi}(y) \right], \text{where}$$

$$(x,y) \sim \mathcal{N} \left( \mathbf{0}, \begin{bmatrix} 1 & \rho \\ \rho & 1 \end{bmatrix} \right),$$

where $\hat{\phi}(\cdot)$ is the positive $n$-homogeneous part of $\phi(\cdot)$, i.e. $\hat{\phi}(x) = \lim_{\lambda \to \infty} \frac{\phi(\lambda x)}{\lambda^n} \ \forall x \in \mathbb{R}$.

Observe that $\hat{\kappa}(\cdot) \in L^2(\mathbb{R}, \gamma)$, where $\gamma$ is the standard normal measure, which follows trivially from the fact that $\hat{\phi}(\cdot)$ has polynomial bidirectional asymptotic growth (specifically monomial growth of order $n$). Indeed, this is a requirement for the mere existence of $\kappa(\cdot)$.

The probabilists' Hermite polynomials $H_n(\cdot)_{n=0}^{\infty}$ are well-known to form an orthogonal basis in $L^2(\mathbb{R}, \gamma)$, with

$$\int_{\mathbb{R}} H_n(x) H_m(x) d\gamma(x) = n! \delta_{nm}$$

where $\delta_{nm}$ is the Kronecker delta, nonzero only for $n = m$. Thus, we can write

$$\hat{\phi}(\cdot) = \sum_{n=0}^{\infty} a_n H_n(\cdot), \text{where}$$

$$a_n = \frac{1}{n!} \mathbb{E}_{x \sim \mathcal{N}(0,1)} \left[ \hat{\phi}(x) H_n(x) \right].$$

Hence,

$$\hat{\kappa}(\rho) = \mathbb{E}_{(x,y)} \left[ \hat{\phi}(x)\hat{\phi}(y) \right]$$

$$= \mathbb{E}_{(x,y)} \left[ \left( \sum_{n=0}^{\infty} a_n H_n(x) \right) \left( \sum_{m=0}^{\infty} a_m H_m(y) \right) \right]$$

$$= \sum_{n,m=0}^{\infty} a_n a_m \mathbb{E}_{(x,y)} \left[ H_n(x) H_m(y) \right]$$

$$= \sum_{n,m=0}^{\infty} a_n a_m n! \rho^m \delta_{nm}$$

$$= \sum_{n=0}^{\infty} a_n^2 n! \rho^n, \quad \text{where}$$

$$(x,y) \sim \mathcal{N} \left( \mathbf{0}, \begin{bmatrix} 1 & \rho \\ \rho & 1 \end{bmatrix} \right),$$

where the second transition follows from the dominated convergence theorem and the third transition follows from direct application of Lemma E.3.

By assumption of our definition of asymptotic positive $n$-homogeneity, $\hat{\kappa}(\cdot)$ is nontrivial; hence there must exist at least one $a_n \neq 0$, so the corresponding term $a_n^2 n! \rho^n$ is strictly positive for all $\rho > 0$. Thus noting that $1 > 0$,

$$\hat{\kappa}(\rho) > 0 \quad \forall \rho > 0.$$

Noting that under Assumption C.1, $\dot{\phi}(\cdot) \in L^2(\mathbb{R}, \gamma)$, so the above also holds for $\hat{\dot{\kappa}}(\rho)$.

Finally, noting that as $1 > 0$, $\hat{\kappa}(1) > 0$, and recalling $\kappa(\rho) = \frac{\hat{\kappa}(\rho)}{\hat{\kappa}(1)}$ and $\dot{\kappa}(\rho) = \frac{\dot{\hat{\kappa}}(\rho)}{\hat{\kappa}(1)}$,

$$\kappa(\rho) > 0 \quad \forall \rho > 0, \quad \text{and}$$
$$\dot{\kappa}(\rho) > 0 \quad \forall \rho > 0.$$

$\square$

**Corollary E.3.** *For all neural network with fully-connected layers and nonlinearities satisfying Assumption 3.1 and Assumption 3.2,*

$$\hat{\bar{\kappa}}(\hat{\boldsymbol{x}}, \boldsymbol{x}') = \lim_{\lambda \to \infty} \bar{\kappa}(\lambda \boldsymbol{x}, \boldsymbol{x}') > 0 \quad \forall \boldsymbol{x}, \boldsymbol{x}' \in \mathbb{R}^{n_0} : \boldsymbol{x}^\top \boldsymbol{x} > 0,$$

*where $\bar{\kappa}(\boldsymbol{x}, \boldsymbol{x}')$ is defined as in Equation 38 of Proposition E.2.*

*Proof.* We begin by showing that the limit correlation $\hat{\rho}(\hat{\boldsymbol{x}}, \boldsymbol{x}') = \lim_{\lambda \to \infty} \rho^{(h)}(\lambda \boldsymbol{x}, \boldsymbol{x}') > 0 \ \forall \boldsymbol{x}, \boldsymbol{x}' \in \mathbb{R}^{n_0} : \boldsymbol{x}^\top \boldsymbol{x}' > 0, h = 0, \ldots, L$. First, recall the recursion for the single-argument limit correlations derived in Corollary E.1, Recursion 50:

$$\hat{\rho}^{(h+1)}(\hat{\boldsymbol{x}}, \boldsymbol{x}') = \frac{(\|\boldsymbol{x}'\|)^{n^h}}{\sqrt{\Sigma^{(h)}(\boldsymbol{x}', \boldsymbol{x}')}} \kappa\left(\hat{\rho}^{(h)}(\hat{\boldsymbol{x}}, \boldsymbol{x}')\right)$$

$$\hat{\rho}^{(0)}(\hat{\boldsymbol{x}}, \boldsymbol{x}') = \begin{cases} \frac{\hat{\boldsymbol{x}}^\top \boldsymbol{x}'}{\sqrt{\|\boldsymbol{x}'\|^2 + n_0 \sigma_b^2}} & \boldsymbol{x} \in \mathbb{R}^{n_0} \setminus \boldsymbol{0} \\ \frac{\sigma_b}{\sqrt{\frac{1}{n_0}\|\boldsymbol{x}'\|^2 + \sigma_b^2}} & \boldsymbol{x} = \boldsymbol{0} \end{cases}.$$

Due to Lemma E.4, and the strict positivity of $\Sigma^{(h)}(\boldsymbol{x}', \boldsymbol{x}')$ as it is a valid kernel and $\boldsymbol{x}' > 0$ by assumption of the theorem, if $\hat{\rho}^{(h)}(\hat{\boldsymbol{x}}, \boldsymbol{x}') > 0$ then it follows that $\hat{\rho}^{(h+1)}(\hat{\boldsymbol{x}}, \boldsymbol{x}') > 0$ as the nonlinearity $\phi(\cdot)$ satisfies the conditions for this Lemma by assumption of the theorem. Note also that by assumption of the theorem $\hat{\rho}^{(h)}(\hat{\boldsymbol{x}}, \boldsymbol{x}') > 0$, hence by induction

$$\hat{\rho}^{(h)}(\hat{\boldsymbol{x}}, \boldsymbol{x}') > 0 \qquad \forall \boldsymbol{x}, \boldsymbol{x}' \in \mathbb{R}^{n_0} : \hat{\boldsymbol{x}}^\top \hat{\boldsymbol{x}} > 0, h = 0, \ldots, L. \qquad (52)$$

Again due to Lemma E.4,

$$\kappa\left(\hat{\rho}^{(h)}(\hat{\boldsymbol{x}}, \boldsymbol{x}')\right) > 0 \qquad \forall \boldsymbol{x}, \boldsymbol{x}' \in \mathbb{R}^{n_0} : \hat{\boldsymbol{x}}^\top \boldsymbol{x}' > 0, h = 0, \ldots, L, \qquad (53)$$

$$\dot{\kappa}\left(\hat{\rho}^{(h)}(\boldsymbol{x}, \boldsymbol{x}')\right) > 0 \qquad \forall \boldsymbol{x}, \boldsymbol{x}' \in \mathbb{R}^{n_0} : \boldsymbol{x}^\top \boldsymbol{x}' > 0, h = 0, \ldots, L. \qquad (54)$$

To conclude the proof, recall from Equation 38 $\hat{\bar{\kappa}}(\hat{\boldsymbol{x}}, \boldsymbol{x}') = \lim_{\lambda \to \infty} \bar{\kappa}(\lambda \boldsymbol{x}, \boldsymbol{x}')$ is composed of a summation of products of $\hat{\kappa}^{(h)}(\hat{\boldsymbol{x}}, \boldsymbol{x}')$ and $\hat{\dot{\kappa}}^{(h)}(\hat{\boldsymbol{x}}, \boldsymbol{x}')$ for various $h = 0, \ldots, L$, all of which are existing limits and are strictly positive. Thus concluding the proof,

$$\hat{\bar{\kappa}}(\hat{\boldsymbol{x}}, \boldsymbol{x}') = \lim_{\lambda \to \infty} \bar{\kappa}(\lambda \boldsymbol{x}, \boldsymbol{x}') > 0 \qquad \forall \boldsymbol{x}, \boldsymbol{x}' \in \mathbb{R}^{n_0} : \boldsymbol{x}^\top \boldsymbol{x}' > 0 \qquad (55)$$

$\square$

# F  Proof of Theorem C.2 and Theorem 3.2

**Theorem C.2.** *The NTK of a network $f_\theta(\boldsymbol{x})$ with nonlinearities satisfying Assumption 3.1 and Assumption C.1, fully-connected layers, and at least one layer-norm anywhere in the network, enjoys the property that:*

$$\forall_{\boldsymbol{x} \in \mathbb{R}^{n_0}} |\Theta(\boldsymbol{x}, \boldsymbol{x})| \leq C$$

*for some constant $C > 0$.*

*Proof.* We first consider the effect of placing a first layer-norm operation between the linear component and nonlinearity of arbitrary layer $h_{LN}$, with $0 \leq h_{LN} \leq L$ of such a network. The first observation is that this destroys the variance for both inputs $\boldsymbol{x}$ and $\boldsymbol{x}'$ subsequent to this layer, while maintaining correlation. Thus, the activations and therefore $\Sigma^{(\hat{h})}(\boldsymbol{x}, \boldsymbol{x}')$ and $\dot{\Sigma}^{(\hat{h})}(\boldsymbol{x}, \boldsymbol{x}')$ can have no dependence on $\|\boldsymbol{x}\|$ and $\|\boldsymbol{x}'\|$ for all $h_{LN} \leq \hat{h} \leq L$. Not accounting for the effect of the layer-norm on the backwards pass, following the reasoning of the proof of Corollary E.1, the contribution of layer $h$, where $0 \leq h < h_{LN}$ to the NTK would be asymptotically positive $n^{h_{LN}}$-homogeneous with respect to both $\boldsymbol{x}$ and $\boldsymbol{x}'$. That is, at leading order they are polynomials of order $n^{h_{LN}}$ in both $\boldsymbol{x}$ and $\boldsymbol{x}'$. Similarly, the contributions of layers $\hat{h}$ will depend only on the correlation of the activations at layer $h_{LN}$.

We must now consider the effect of the layer-norm operation on the backwards pass. From the proof of Theorem C.2, we see that the layer-norm operation essentially divides the backwards pass by the square root of the variance directly preceding the operation, which in this case is given by the square roots of $\Sigma^{(h_{LN})}(\boldsymbol{x}, \boldsymbol{x})$ and $\Sigma^{(h_{LN})}(\boldsymbol{x}', \boldsymbol{x}')$ for inputs $\boldsymbol{x}$ and $\boldsymbol{x}'$ respectively, both of which are asymptotically positive $n^{h_{LN}}$-homogeneous, i.e. at leading order a polynomial of order $n^{h_{LN}}$.

Accounting for this effect on backpropagation, we see that for all layers $h < h_{LN}$, the contribution to the NTK is now given by the ratio of two asymptotically positive $n^{h_{LN}}$-homogeneous terms in both $\boldsymbol{x}$ and $\boldsymbol{x}'$. Thus, in the limit as $\|\boldsymbol{x}\|, \|\boldsymbol{x}'\| \to \infty$, the contribution to the NTK must approach a finite limit dependent only on the layer correlations between the two inputs.

For the case $\boldsymbol{x} = \boldsymbol{x}'$, all layer correlations are 1, thus in the limit $\|\boldsymbol{x}\| \to \infty$, $\Theta(\boldsymbol{x}, \boldsymbol{x})$ approaches a positive constant independent of the direction of $\boldsymbol{x}$. By the assumptions of the Theorem, specifically that the nonlinearities are locally-bounded, almost-everywhere differentiable, and of locally-bounded derivative (which apply to all commonly activation functions), $\Theta(\boldsymbol{x}, \boldsymbol{x})$ is also bounded by a finite positive constant for all $\|\boldsymbol{x}\|$ not inducing the homogeneous regime. Thus, the NTK of such a network satisfies $\forall_{\boldsymbol{x} \in \mathbb{R}^{n_0}} \|\Theta(\boldsymbol{x}, \boldsymbol{x})\| \leq C$ for some finite constant $C > 0$. Hence, the proof is concluded. $\square$

**Theorem 3.2.** *The NTK of a network $f_\theta(\boldsymbol{x})$ with nonlinearities satisfying Assumption 3.1 and Assumption 3.2, fully-connected layers, and at least one Layer Norm anywhere in the network, enjoys the property that:*

$$\forall_{\boldsymbol{x} \in \mathbb{R}^{n_0}} |\Theta(\boldsymbol{x}, \boldsymbol{x})| \leq C$$

*for some constant $C > 0$.*

*Proof.* By Corollary C.1, positive $n$-homogeneous nonlinearities satisfying Assumption 3.1 also satisfy the conditions for Theorem C.2, hence its result also applies here. $\square$

# G  Proof of Theorem 3.3

**Theorem 3.3.** *Given an infinitely-wide network $f_\theta(\boldsymbol{x})$ satisfying Assumption 3.1 and Assumption 3.2, fully-connected layers, and at least one layer-norm anywhere in the network, trained until convergence on any training data $\mathcal{D}_{train} = (\mathcal{X}_{train}, \mathcal{Y}_{train})$, we have that for any $\boldsymbol{x} \in \mathbb{R}^{n_0}$:*

$$|\mathbb{E}[f_\theta(\boldsymbol{x})]| \leq B(\mathcal{D}_{train}) = \sqrt{\frac{C \max_{y \in \mathcal{Y}_{train}} |y|}{\lambda_{min}}} |D_{train}|,$$

*where the expectation is taken over initialisation and $\lambda_{min}$ is the smallest eigenvalue of $\Theta_{train}$ and $C$ is the kernel-dependent constant from Theorem C.2 or Theorem 3.2.*

*Proof.* For an inifinitely wide NN after convergence we have:

$$f_\theta(\boldsymbol{x}) = \Theta(\boldsymbol{x}, \mathcal{X}_{\text{train}})\Theta_{\text{train}}^{-1}(y - N_0(\boldsymbol{x})) + N_0(\boldsymbol{x}),$$

where $(\Theta(\boldsymbol{x}, \mathcal{X}_{\text{train}}))_i = \Theta(\boldsymbol{x}, \boldsymbol{x}_i)$ and $(\Theta_{\text{train}})_{i,j} = \Theta(\boldsymbol{x}_i, \boldsymbol{x}_j)$ and $(y)_i = y_i$ for $i, j = 1, \ldots, |\mathcal{D}_{\text{train}}|$ and $N_0(\boldsymbol{x})$ is the output of freshly initialised network with property $\mathbb{E}[N_0(\boldsymbol{x})] = 0$. Let us denote by $\psi(\cdot)$ the feature map of network's NTK, such that $\Theta(\boldsymbol{x}, \boldsymbol{x}') = \psi(\boldsymbol{x})^\top \psi(\boldsymbol{x})$ and let $\lambda_{\min}$ be the

smallest eigenvalue of the $\Theta_{\text{train}}$ matrix. In expectation we thus have:

$$
\begin{aligned}
|\mathbb{E}[f_\theta(\boldsymbol{x})]| &= \left|\Theta(\boldsymbol{x}, \mathcal{X}_{\text{train}})\Theta_{\text{train}}^{-1} y\right| \\
&\leq \left\|\Theta(\boldsymbol{x}, \mathcal{X}_{\text{train}})\Theta_{\text{train}}^{-1}\right\|_2 \|y\|_2 \\
&\leq \frac{1}{\sqrt{\lambda_{\min}}} \|\Theta(\boldsymbol{x}, \mathcal{X}_{\text{train}})\|_2 \sqrt{|D_{\text{train}}| \max_{y \in \mathcal{Y}_{\text{train}}} |y|} \\
&= \frac{1}{\sqrt{\lambda_{\min}}} \sqrt{\sum_{\boldsymbol{x}' \in \mathbb{R}_{\text{train}}^{n_0}} \psi(\boldsymbol{x})^\top \psi(\boldsymbol{x}')} \sqrt{|D_{\text{train}}| \max_{y \in \mathcal{Y}_{\text{train}}} |y|} \\
&\leq \frac{1}{\sqrt{\lambda_{\min}}} \sqrt{|D_{\text{train}}| C} \sqrt{|D_{\text{train}}| \max_{y \in \mathcal{Y}_{\text{train}}} |y|} \\
&= \sqrt{\frac{C \max_{y \in \mathcal{Y}_{\text{train}}} |y|}{\lambda_{\min}}} |D_{\text{train}}|,
\end{aligned}
$$

where the penultimate transition is due to Proposition E.1. $\qquad\square$

## H  Proof of Theorem 3.1

**Theorem 3.1.** *Consider an infinitely-wide network $f_\theta(\boldsymbol{x})$ with nonlinearities satisfying Assumption 3.1 and Assumption 3.2, and fully-connected layers. Then, there exists a finite dataset $\mathcal{D}_{train} = (\mathcal{X}_{train}, \mathcal{Y}_{train})$ such that any such network trained upon $\mathcal{D}$ until convergence has* [4]

$$
\sup_{\boldsymbol{x} \in \mathbb{R}^{n_0}} |\mathbb{E}[f_\theta(\boldsymbol{x})]| = \infty,
$$

*where the expectation is taken over initialisation.*

*Proof.* For an infinitely-wide NN after convergence we have:

$$
f_\theta(\boldsymbol{x}) = \Theta(\boldsymbol{x}, \mathcal{X}_{\text{train}})\Theta_{\text{train}}^{-1}(y - N_0(\boldsymbol{x})) + N_0(\boldsymbol{x}),
$$

where $(\Theta(\boldsymbol{x}, \mathcal{X}_{\text{train}}))_i = \Theta(\boldsymbol{x}, \boldsymbol{x}_i)$ and $(\Theta_{\text{train}})_{i,j} = \Theta(\boldsymbol{x}_i, \boldsymbol{x}_j)$ and $(y)_i = y_i$ for $i, j = 1, \dots, |\mathcal{D}_{\text{train}}|$ and $N_0(\boldsymbol{x})$ is the output of freshly initialised network with property $\mathbb{E}[N_0(\boldsymbol{x})] = 0$. As such, we have:

$$
\mathbb{E}[f_\theta(\boldsymbol{x})] = \Theta(\boldsymbol{x}, \mathcal{X}_{\text{train}})\Theta_{\text{train}}^{-1} \mathcal{Y}_{\text{train}},
$$

which is just the mean of a noiseless Gaussian Process conditioned on $\mathcal{D}_{\text{train}}$ with kernel $\Theta(\boldsymbol{x}, \boldsymbol{x}')$. Due to the Representer Theorem, we can always rewrite this mean as:

$$
\mathbb{E}[f_\theta(\boldsymbol{x})] = \sum_{\boldsymbol{x}' \in \mathbb{R}_{\text{train}}^{n_0}} \alpha_{\boldsymbol{x}'} \Theta(\boldsymbol{x}, \boldsymbol{x}'),
$$

for some $\alpha_{\boldsymbol{x}'} \in \mathbb{R}$. Due to the assumption of Theorem 3.1, we must have at least one $\tilde{y} \in \mathcal{Y}_{\text{train}}$ such that $\tilde{y} \neq 0$, and hence $\boldsymbol{\alpha} \in \mathbb{R}^{\|\mathcal{X}_{\text{train}}\|} \neq \boldsymbol{0}$, where $\boldsymbol{\alpha}_i = \alpha_{\boldsymbol{x}_i}$ for $i = 1, \dots, \|\mathcal{D}_{\text{train}}\|$.

Let us study the predictions for the set $\lambda \mathcal{X}_{\text{train}}$, where $(\lambda \mathcal{X}_{\text{train}})_i \in \mathbb{R}^{n_0} = \lambda (\mathcal{X}_{\text{train}})_i$ for some $\lambda > 0$, given by

$$
\begin{aligned}
\mathbb{E}[N(\lambda \mathcal{X}_{\text{train}})]_i &= \mathbb{E}[N(\lambda \boldsymbol{x}_i)] \\
&= \sum_{\boldsymbol{x}' \in \mathcal{X}_{\text{train}}} \alpha_{\boldsymbol{x}'} \Theta(\lambda \boldsymbol{x}_i, \boldsymbol{x}').
\end{aligned}
$$

We can rewrite this as $\mathbb{E}[N(\lambda \mathcal{X}_{\text{train}})] = \Theta(\lambda \mathcal{X}_{\text{train}}, \mathcal{X}_{\text{train}}) \boldsymbol{\alpha}$, where $\Theta(\lambda \mathcal{X}_{\text{train}}, \mathcal{X}_{\text{train}})_{i,j} = \Theta(\lambda \boldsymbol{x}_i, \boldsymbol{x}_j)$ for $i, j = 1, \dots, \|\mathcal{D}_{\text{train}}\|$. As $\mathcal{X}_{\text{train}}$ is non-degenerate and $\Theta(\cdot, \cdot')$ is a valid kernel, $\Theta(\lambda \mathcal{X}_{\text{train}}, \mathcal{X}_{\text{train}})$ is full-rank $\forall \lambda > 0$, and thus has trivial nullspace. Therefore, as $\boldsymbol{\alpha} \neq$

---

[4]When proving statements about infinite networks $f_\theta(\boldsymbol{x})$, we refer to the limiting network. Eg. in this case the statement is equivalent to $\sup_{\boldsymbol{x} \in \mathbb{R}^{n_0}} |\mathbb{E}[\lim_{n \to \infty} f_\theta(\boldsymbol{x})]|$, where $n$ is the width of smallest layer.

$0$, $\mathbb{E}\left[N(\lambda \mathcal{X}_{\text{train}})\right] = \Theta\left(\lambda \mathcal{X}_{\text{train}}, \mathcal{X}_{\text{train}}\right)\boldsymbol{\alpha} \neq \mathbf{0}$. Hence, $\forall \lambda > 0, \exists \tilde{\boldsymbol{x}} \in \mathcal{X}_{\text{train}}$ such that $\mathbb{E}\left[N(\tilde{\boldsymbol{x}})\right] \neq 0$. We can arbitrarily multiply this expression by $\frac{\lambda^{n^L}}{\lambda^{n^L}} = 1$, giving

$$\mathbb{E}\left[N(\lambda\tilde{\boldsymbol{x}})\right] = \lambda^{n^L} \sum_{\boldsymbol{x}' \in \mathcal{X}_{\text{train}}} \alpha_{\boldsymbol{x}'} \frac{\Theta\left(\lambda\tilde{\boldsymbol{x}}, \boldsymbol{x}'\right)}{\lambda^{n^L}}.$$

Consider now the case where $\mathcal{X}_{\text{train}}$ is constructed such that $\boldsymbol{x}^\top \boldsymbol{x}' > 0 \ \forall \boldsymbol{x}, \boldsymbol{x}' \in \mathcal{X}_{\text{train}}$. Then, due to Corollary E.2 and Corollary E.3,

$$\begin{aligned}
\lim_{\lambda \to \infty} \frac{\Theta\left(\lambda\tilde{\boldsymbol{x}}, \boldsymbol{x}'\right)}{\lambda^{n^L}} &= \hat{\Theta}\left(\tilde{\boldsymbol{x}}, \boldsymbol{x}'\right) \\
&= \|\tilde{\boldsymbol{x}}\| \, \|\boldsymbol{x}'\| \, \hat{\hat{\kappa}}\left(\hat{\tilde{\boldsymbol{x}}}, \boldsymbol{x}'\right) \\
&> 0 \quad \forall \boldsymbol{x}' \in \mathcal{X}_{\text{train}},
\end{aligned}$$

where $\hat{\tilde{\boldsymbol{x}}}$ is a unit vector in the direction of $\tilde{\boldsymbol{x}}$ and the final transition holds as the condition $\boldsymbol{x}^\top \boldsymbol{x}' > 0 \ \forall \boldsymbol{x}, \boldsymbol{x}' \in \mathcal{X}_{\text{train}}$ guarantees $\|\boldsymbol{x}\| > 0 \ \forall \boldsymbol{x} \in \mathbb{R}^{n_0}$.

Invoking Corollary E.1, in the limit we then have

$$\begin{aligned}
\lim_{\lambda \to \infty} \left|\mathbb{E}\left[N(\lambda\tilde{\boldsymbol{x}})\right]\right| &= \lim_{\lambda \to \infty} \left|\lambda^{n^L} \sum_{\boldsymbol{x}' \in \mathcal{X}_{\text{train}}} \alpha_{\boldsymbol{x}'} \frac{\Theta\left(\lambda\tilde{\boldsymbol{x}}, \boldsymbol{x}'\right)}{\lambda^{n^L}}\right| \\
&= \lim_{\lambda \to \infty} \left|\lambda^{n^L}\right| \lim_{\lambda \to \infty} \left|\sum_{\boldsymbol{x}' \in \mathcal{X}_{\text{train}}} \alpha_{\boldsymbol{x}'} \frac{\Theta\left(\lambda\tilde{\boldsymbol{x}}, \boldsymbol{x}'\right)}{\lambda^{n^L}}\right| \\
&= \lim_{\lambda \to \infty} \lambda^{n^L} \left|\sum_{\boldsymbol{x}' \in \mathcal{X}_{\text{train}}} \alpha_{\boldsymbol{x}'} \hat{\Theta}\left(\tilde{\boldsymbol{x}}, \boldsymbol{x}'\right)\right|
\end{aligned}$$

where the last transition follows from the assumption that $n > 0$ and the first transition is valid as we can choose $\tilde{\boldsymbol{x}} \in \mathbb{R}^{n_0}$ such that $\lim_{\lambda \to \infty} \left|\sum_{\boldsymbol{x}' \in \mathcal{X}_{\text{train}}} \alpha_{\boldsymbol{x}'} \frac{\Theta\left(\lambda\tilde{\boldsymbol{x}}, \boldsymbol{x}'\right)}{\lambda^{n^L}}\right| = \left|\sum_{\boldsymbol{x}' \in \mathcal{X}_{\text{train}}} \alpha_{\boldsymbol{x}'} \hat{\Theta}\left(\tilde{\boldsymbol{x}}, \boldsymbol{x}'\right)\right| \neq 0$. Concluding the proof, it follows that

$$\sup_{\boldsymbol{x} \in \mathbb{R}^{n_0}} \left|\mathbb{E}\left[f_\theta(\boldsymbol{x})\right]\right| = \infty.$$

$\square$

# I  Proof of Proposition 3.4

**Proposition 3.4.** *There exists a trained network $\tilde{f}_{\tilde{\theta}}(\boldsymbol{x})$ including Layer Norm layers such that*

$$\sup_{\boldsymbol{x} \in \mathbb{R}^{n_0}, \tilde{\theta} \in \tilde{\Omega}^\star} \left|\tilde{f}_{\tilde{\theta}}(\boldsymbol{x})\right| = \infty,$$

*where $\tilde{\Omega}^\star = \arg\min_{\tilde{\theta} \in \tilde{\Theta}} L\left(\tilde{\theta}, \mathcal{D}_{train}\right)$ is the set of minimisers of the training loss.*

Consider some network $f_\theta(\boldsymbol{x})$ with fully-connected layers, operating in an over-parametrised regime such that additional linear layer parameters may not further decrease the training loss over some finite training set $\mathcal{D}_{\text{train}}$. Suppose further that

$$\sup_{\boldsymbol{x} \in \mathbb{R}^{n_0}, \theta \in \Omega^\star} |f_\theta(\boldsymbol{x})| = \infty, \qquad\qquad\qquad \text{where}$$

$$\Omega^\star = \arg\min_{\theta \in \Omega} L\left(\theta, \mathcal{D}_{\text{train}}\right) = \left\{ \theta \in \Omega : L\left(\theta, \mathcal{D}_{\text{train}}\right) = \min_{\theta \in \Omega} L\left(\theta, \mathcal{D}_{\text{train}}\right) \right\}$$

Such a network trivially exists; for any finite training dataset, any network interpolating the training targets (or, in the case of a degenerate training set with duplicated inputs, interpolating the minimal predictions with respect to the loss) will achieve the minimum possible loss of all networks. Moreover, there always exists a finite fully connected network achieving this [8].

Consider now the modified network $\tilde{f}_{\tilde{\theta}}(\boldsymbol{x})$ formed by the insertion of a linear layer followed by an LN operation and ReLU nonlinearity anywhere in the network immediately followed by another linear layer. Specifically, we choose this additional layer to have $2n + 2$ output nodes, where $n$ is the number of nodes in the preceding layer.

Suppose we are free to choose the parameters of this linear layer. In block diagonal form, we choose the following:

$$\boldsymbol{W} = \begin{bmatrix} \mathbf{0}_n & \mathcal{I}_n & -\mathcal{I}_n & \mathbf{0}_n \end{bmatrix}^\top,$$

$$\boldsymbol{b} = \begin{bmatrix} \frac{\sqrt{n}}{\varepsilon} & \mathbf{0}_n^\top & \mathbf{0}_n^\top & -\frac{\sqrt{n}}{\varepsilon} \end{bmatrix}^\top,$$

where $\mathbf{0}_n \in \mathbb{R}^n$ is a vector of zeros, $\mathcal{I}_n$ is the $n \times n$ identity matrix, and $\varepsilon > 0$ is some positive constant.

Denoting the input to the linear layer as $\boldsymbol{z} \in \mathbb{R}^n$, the first and second moments of the output of the linear layer are given by

$$\mu = \frac{1}{2n+2} \left( \frac{\sqrt{n}}{\varepsilon} + \boldsymbol{z} - \boldsymbol{z} - \frac{\sqrt{n}}{\varepsilon} \right) = 0,$$

$$\sigma^2 = \frac{1}{2n+2} \left( \frac{n}{\varepsilon^2} + \sum_{i=1}^{n} z_i^2 + \sum_{i=1}^{n} z_i^2 + \frac{n}{\varepsilon^2} \right) = \frac{n}{n+1} \left( \frac{1}{\varepsilon^2} + \bar{z}_{\text{rms}}^2 \right),$$

where $\bar{z}_{\text{rms}} = \sqrt{\frac{1}{n} \sum_{i=1}^{n} z_i^2}$ is the root mean square value of $\boldsymbol{z}$. As all $\boldsymbol{x} \in \mathcal{X}_{\text{train}}$ are finite, so are all $\bar{z}_{\text{rms}}$, hence we can upper bound $\bar{z}_{\text{rms}}$ by the largest value of any element observed during training, $\bar{z}_{\text{max}}$. for all forward passes during training. Thus, setting $\varepsilon \ll \frac{1}{\bar{z}_{\text{max}}}$,

$$\sigma^2 \approx \frac{n}{n+1} \frac{1}{\varepsilon^2}$$

for all forward passes during training. Consider now the outputs from the Layer Norm during training, given by

$$\tilde{\boldsymbol{z}} = \frac{\boldsymbol{W}\boldsymbol{z} + \boldsymbol{b} - \mu}{\sigma}$$

$$\approx \sqrt{\frac{n+1}{n}} \varepsilon \left( \boldsymbol{W}\boldsymbol{z} + \boldsymbol{b} \right)$$

$$= \sqrt{\frac{n+1}{n}} \begin{bmatrix} \sqrt{n} & \varepsilon\boldsymbol{z} & -\varepsilon\boldsymbol{z} & -\sqrt{n} \end{bmatrix}^\top.$$

Consider now the next linear layer, with weight matrix $\boldsymbol{W}' \in \mathbb{R}^{m \times (2n+2)}$. Suppose in the absence of our additions (and noting the differing dimensions), this layer originally had optimal weight $\boldsymbol{W}^\star \in \mathbb{R}^{m \times n}$ and bias $\boldsymbol{b}^\star \in \mathbb{R}^m$. We can freely decompose $\boldsymbol{W}'$ as

$$\boldsymbol{W}' = \boldsymbol{W}^\star \begin{bmatrix} \boldsymbol{0}_n & \frac{1}{\varepsilon}\mathcal{I}_n & -\frac{1}{\varepsilon}\mathcal{I}_n & \boldsymbol{0}_n \end{bmatrix},$$

such that the output of this linear layer is

$$\begin{aligned}
\boldsymbol{z}' &= \boldsymbol{W}'\phi_{\mathrm{ReLU}}\left(\tilde{\boldsymbol{z}}\right) + \boldsymbol{b}^\star \\
&= \boldsymbol{W}^\star \begin{bmatrix} \boldsymbol{0}_n & \frac{1}{\varepsilon}\mathcal{I}_n & -\frac{1}{\varepsilon}\mathcal{I}_n & \boldsymbol{0}_n \end{bmatrix} \phi_{\mathrm{ReLU}}\left(\begin{bmatrix} \sqrt{n} & \varepsilon\boldsymbol{z} & -\varepsilon\boldsymbol{z} & -\sqrt{n} \end{bmatrix}^\top\right) + \boldsymbol{b}^\star \\
&= \boldsymbol{W}^\star\frac{1}{\varepsilon}\left(\phi_{\mathrm{ReLU}}\left(\varepsilon\boldsymbol{z}\right) - \phi_{\mathrm{ReLU}}\left(-\varepsilon\boldsymbol{z}\right)\right) + \boldsymbol{b}^\star \\
&= \boldsymbol{W}^\star\frac{1}{\varepsilon}\left(\max\left(\boldsymbol{0}_n, \varepsilon\boldsymbol{z}\right) - \max\left(\boldsymbol{0}_n, -\varepsilon\boldsymbol{z}\right)\right) + \boldsymbol{b}^\star \\
&= \boldsymbol{W}^\star\frac{1}{\varepsilon}\left(\varepsilon\boldsymbol{z}\right) + \boldsymbol{b}^\star \\
&= \boldsymbol{W}^\star\boldsymbol{z} + \boldsymbol{b}^\star,
\end{aligned}$$

i.e. the unmodified output of the network, where the second transition follows from the element-wise distribution of ReLU. By assumption of the Theorem, this minimises the empirical risk over the training set and the additional parameters and layer-norm do not further decrease the value of this minimum. Hence, denoting a complete set of unaugmented network parameters drawn from $\Omega^\star$, the set of minimisers of $L\left(\boldsymbol{x}, \mathcal{D}_{\mathrm{train}}\right)$, augmented with our additional and modified parameters, as $\theta\left(\boldsymbol{W}, \boldsymbol{b}, \boldsymbol{W}'\right)$ it follows that

$$\begin{aligned}
L\left(\theta\left(\boldsymbol{W}, \boldsymbol{b}, \boldsymbol{W}'\right), \mathcal{D}_{\mathrm{train}}\right) &= \min_{\theta \in \Omega} L\left(\theta, \mathcal{D}_{\mathrm{train}}\right) \\
&= \min_{\tilde{\theta} \in \tilde{\Theta}} L\left(\tilde{\theta}(\boldsymbol{x}), \mathcal{D}_{\mathrm{train}}\right) \quad \text{and so} \\
\theta\left(\boldsymbol{W}, \boldsymbol{b}, \boldsymbol{W}'\right) &\in \Omega^\star \quad \forall \varepsilon \ll \boldsymbol{z}_{\max},
\end{aligned}$$

where the first transition follows from the assumption of the theorem that the network is sufficiently overparametrised such that these augmentations cannot further reduce the empirical risk over the training set. Thus, when considering supremums over $\tilde{\Omega}^\star$, we can consider the limiting case $\varepsilon \to 0$. Considering the dual-limiting case where $\varepsilon \to 0$ faster than $\|\boldsymbol{x}\| \to \infty$ for some $\boldsymbol{x} \in \mathbb{R}^{n_0}$, it is clear that the assumption on $\varepsilon$ holds for all such $\boldsymbol{x}$, thus the Layer Norm is effectively bypassed and we approach a network exactly equivalent to the unmodified network. More precisely,

$$\forall \boldsymbol{x} \in \mathbb{R}^{n_0}, \quad \exists \tilde{\theta} \in \tilde{\Omega}^\star \text{ such that } \tilde{f}_{\tilde{\theta}}(\boldsymbol{x}) = f_\theta(\boldsymbol{x}).$$

Noting that by assumption that

$$\sup_{\boldsymbol{x} \in \mathbb{R}^{n_0}, \theta \in \Omega^\star} |f_\theta(\boldsymbol{x})| = \infty,$$

it is clear that

$$\sup_{\boldsymbol{x} \in \mathbb{R}^{n_0}, \tilde{\theta} \in \tilde{\Omega}^\star} \left|\tilde{f}_{\tilde{\theta}}(\boldsymbol{x})\right| = \infty.$$

## J Experimental Details

To run all experiments we used NVIDIA GeForce RTX 3090 with 24GB of memory. For all experiments we used the same architecture consisting of fully-connected layers with ReLU non-linearities. All hidden layers have a size of 128. To be consistent with the theory, we initialised

weights of fully-connected layers with Kaiming initialisation [16] and biases were sampled from $\mathcal{N}(0, \sigma_b^2)$ with $\sigma_b^2 = 0.01$. For the UTK experiments in Section 4.3, a frozen ResNet-18 [17] is used before the fully-connected layers. We utilised Adam optimiser [21] and MSE Loss for all experiments. See Table 3 below for exact hyperparameters settings. For XGBoost we use the XGBoost Python package [5] with hyperparameter values set to default values and number of boosting rounds set to 100. Training took less than 10 seconds per seed for experiments in Section 4.1 and 4.2, and around 2 minutes per seed for experiments in Section 4.3.

Table 3: Hyperparameter values used throughout the experiments.

| Experiment | Method | Batch size | Epochs | Learning rate |
|---|---|---|---|---|
| 4.1 | All | 100 (entire dataset) | 3000 | 0.001 |
| 4.2 | All | 40132 (entire dataset) | 2500 | 0.001 |
| 4.3 | Standard NN | 128 | 10 | 0.001 |
| | LN after 1st | 128 | 10 | 0.001 |
| | LN after 2nd | 128 | 10 | 0.003 |
| | LN after every | 128 | 10 | 0.003 |

# K    Additional Experiments: Different Activations

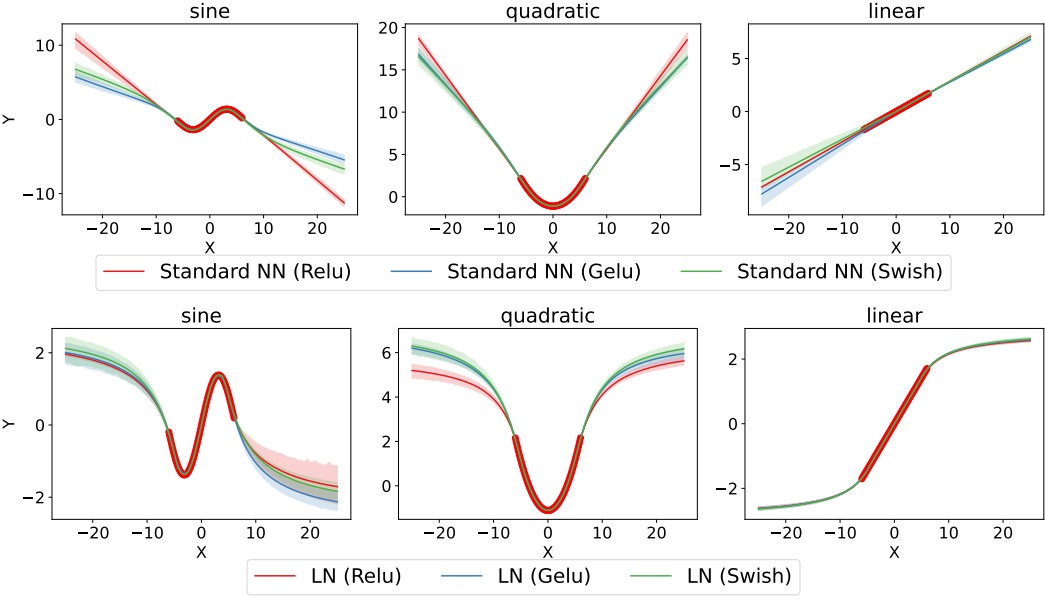

Figure 5: Predictions made by networks with various activations when trained on synthetic datasets. Plots above consider the case of standard NN without LayerNorm, whereas plots below show the case of varying activations while keeping the LayerNorm in the architecture. Red dots show the train set datapoints. The solid lines indicate average values over 5 seeds and shaded areas are 95% confidence intervals of the mean estimator.

---

[5] https://xgboost.readthedocs.io/en/stable/index.html

## L  Additional Experiments: Batch Norm vs Layer Norm

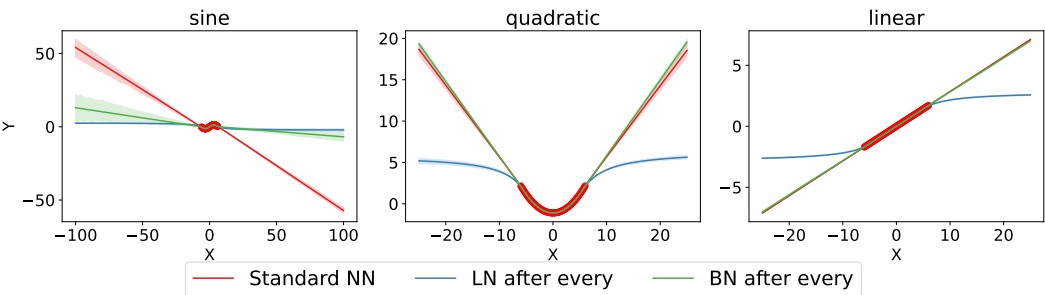

Figure 6: Predictions made by networks with various activations when trained on synthetic datasets. Red dots show the train set datapoints. The solid lines indicate average values over 5 seeds and shaded areas are 95% confidence intervals of the mean estimator.

## M  Additional Experiments: Transformer

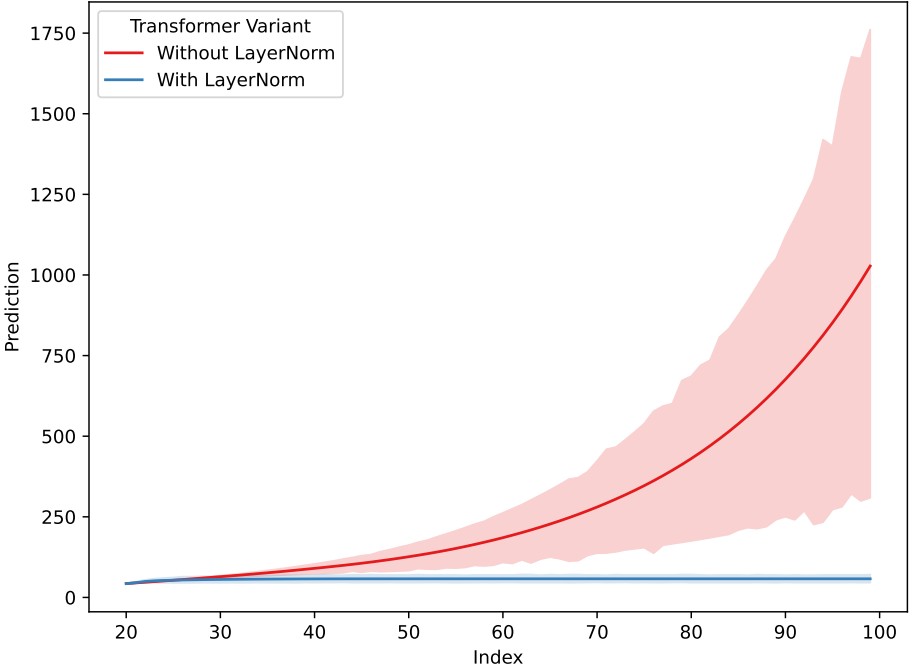

Figure 7: Results on a transformer toy problem. We train a 2-layer decoder-only transformer with 4 heads and embedding size of 64 with two versions: with and without layer normalisation after input embedding layer. The model is trained to on sequences of form $f(i) = a + b * i$, where $a, b \sim \mathcal{N}(0, 1)$ and $i \in \{1, 2, \dots, 30\}$ to predict the value with index $i + 1$ given value with indices from 1 to $i$. On the plot above we show the average prediction made by each model, when tested for index $i \in \{20, 21, \dots, 100\}$ and as such beyond its training domain. In red we show model without LayerNorm, whereas blue show the model with LayerNorm. Solid lines are means over 5 seeds and shaded areas are 95% confidence intervals.

