# OpenReview forum: "Just One Layer Norm Guarantees Stable Extrapolation"
_NeurIPS.cc/2025/Conference — NeurIPS 2025 poster_

### Official Review · Reviewer_WTJo · 2025-06-12

**Clarity:** 3
**Significance:** 2
**Originality:** 2
**Rating:** 4
**Confidence:** 3

**Summary:**

This paper shows that a single LayerNorm layer induces boundedness of the learned function in the infinite-width limit under neural tangent parameterization and MSE loss. It is argued that this boundedness causes extrapolatory stability outside the training domain in practice.

**Questions:**

- **Layernorm position.** It is interesting that one normalization layer — irrespective of its position — suffices. Can you provide an informed intuition why the position does not seem to matter and under which circumstances? Concretely analysing the resulting NTKs could enable to quantify this statement. It could also enable to quantify whether the differences in Table 2 between layernorm positions stem from the data or from the convolutional architecture.
- **Generalizability beyond the kernel regime.** The potential statement ‘Normalizing the last-layer activations induces norm-bounded outputs, as long as the output layer has bounded operator norm throughout training.’ almost sounds tautological. Do you think it would be feasible to show similar boundedness results in much more generality beyond the kernel regime?
- **Other stabilizing components.** Your results hold for LayerNorm. Would other stabilizing components such as BatchNorm layers or the cross-entropy loss (as would be relevant for practice) also ìnduce boundedness OOD?
- **RKHS statement.** In lines 127 to 129 you write an equivalence statement that the RKHS only contains functions which grow without bound, but is it not enough, that there exist functions in the RKHS that grow without bound and that the coefficients learned by the training procedure on these unbounded functions are non-negligible?

**Ethical Concerns:**

["NO or VERY MINOR ethics concerns only"]

**Final Justification:**

The paper is technically solid, but the results are not very surprising: Layernorm induces boundedness which often improves OOD performance. The theory is limited to the kernel regime, which limits practicality. The promised changes will improve the paper but do not affect my overall slightly positive evaluation.

**Limitations:**

The limitation that NTP is used for the theoretical results and that the consequence for practical settings remains unclear is not sufficiently clarified or empirically evaluated.

**Paper Formatting Concerns:**

The instructions of the NeurIPS Paper Checklist should be deleted.

**Quality:**

3

**Strengths And Weaknesses:**

**Strengths:**

The paper is well written and the proof sketches are informative. It is interesting that one Layernorm suffices for boundedness of the NTK, irrespective of its position. The boundedness result holds under weak distributional and architectural assumptions. Albeit a simple and unsurprising intervention, the improved OOD generalization is an impactful consequence, even though all popular architectures already employ normalization layers so that the contribution is limited to a principled understanding of the benefits, which is still interesting. The kernel theory results look correct.


**Weaknesses:**

The following main weaknesses should be addressed:

- **Limited correspondence to practical neural networks due to NTP.** The paper uses neural tangent parameterization, as explicitly specified only in Appendix A. This results in a kernel regime in the infinite-width limit, which yields its theoretical tractability. This could have been clarified. Even Jacot et al. (2018) noted that NTP is not standard parameterization (SP) (as is ambiguously suggested in l.80), hence the implications to finite networks in SP remains unclear, and has been shown to not hold at realistic widths in various papers. Hence a closer empirical evaluation in practical settings would be crucial to quantify how well boundedness results hold in finite networks at the end of training and which impact non-vanishing feature learning has on the main conclusions in this paper. Is the soft cosine similarity still useful for describing Layernorm networks at the end of training? Do predictions of networks without Layernorm indeed diverge on realistic test and OOD inputs, or is this not a practical issue? The experimental results on real data are only tangentially related to the theory, and it remains unclear whether they are cause by the fact that the main theoretical conclusions still hold in practical finite networks after training on vision, protein or language data. Using frozen ResNet features is arguably an artificial setting. Full training would be more interesting. It would be interesting to see whether the main claims also hold in feature learning networks, such as parameterized in muP.
- **Missing acknowledgement of related work.** After a 5-minute literature search, it looks like normalization for improved OOD performance has been previously considered, see Regmi et al. (2024) and references therein such as LogitNorm (Wei et al., 2024). It seems that this line of work should at least be mentioned in the Related Work section.

**Minor point:**

- MSE loss is nice for theoretical tractability, but CE loss is important for capturing practical calibration properties and could result in very different conclusions.

**Typos:** l. 78: then, 95: the the, 146: NTK variance?, 179: …it is possible…, 207: respectively.

**References:**

Regmi, Sudarshan et al. “T2FNorm: Train-time Feature Normalization for OOD Detection
in Image Classification.” *CVPR workshop paper, 2024.*

Wei, Hongxin, et al. "Mitigating neural network overconfidence with logit normalization." *ICML*, 2022.

---

> ### Author Rebuttal · Authors · 2025-07-28
>
> We would like to thank the reviewer for reading our submission and providing valuable feedback. We are glad to know the reviewer finds our results interesting. We address all questions/concerns below:
>
> **Layernorm position**
>
> Intuitively, if after the LayerNorm operation, the distribution of activation is ensured to have a mean of 0 and variance of 1, regardless of the input. As such no matter what the input, the activations after LayerNorm cannot get too “extreme”.  As the final output of the network is just the output of the remaining layers of the network, and activations serving as an input to the rest of the network cannot be too “extreme”, the output also cannot get too “extreme” and, as such remains bounded.
>
> **Missing related work**
>
> We thank the reviewer for pointing out related work. We will include these papers in the related work section. We would like to highlight that while these papers conduct a form of normalisation to improve OOD performance, they appear to be quite different from layer normalisation, and explicit stability bounds are not derived.
>
> **MSE vs CE Loss**
>
> Our results regarding the resulting NTK of networks with LayerNorm hold regardless of the loss function being used, but the question of how the NTK affects the trained network’s prediction when a loss different than MSE is used is generally an open problem in the literature. However, if an analytical expression was derived connecting the NTK to the trained network’s prediction for a loss other than MSE, we believe one could easily plug our NTK bound (which is a property only of the kernel and is independent of the choice of loss function) into it and analyse the resulting expression.
>
> **Generalizability beyond the kernel regime**
>
> While this is an interesting and desirable extension to the theorems presented in this work, the machinery for proving such statements beyond the kernel regime is simply not sufficiently developed yet. As the field evolves, this is certainly a question worth addressing, but for now we point the reviewer towards the empirical results demonstrating the practical applicability of our theory: finitely-wide networks also exhibit characteristics of a feature-learning regime, so the supportive results here also lend credibility to such an extension.
>
> **Other stabilizing components**
>
> For the case of batch norm these results would not hold, as batch norm does not fundamentally alter the NTK and only manifests as an affine transformation. Other stabilising components such as the softmax associated with cross-entropy loss, or even manually clipping network outputs, also of course achieve the stability property, but these are explicit design choices; we focus on LayerNorm, as it is a popular component already present in many deep learning architectures, which interestingly causes this bounded output behaviour organically.
>
> To empirical validate the claim about BatchNorm, we ran more experiments on the sine function with a fully-connected neural network utilising batchnorm instead of LayerNorm, we present new results below (due to new policy we cannot share figures via links, so have to do it via a table):
> | X         | Standard NN        | LayerNorm              | BatchNorm              |
> |-----------|----------------------------|-----------------------------|-----------------------------|
> | -979.80   | 558.49 ± 77.99             | 2.63 ± 0.14                 | 134.86 ± 103.28             |
> | -595.96   | 338.44 ± 46.98             | 2.62 ± 0.14                 | 81.68 ± 62.56               |
> | -191.92   | 106.79 ± 14.34             | 2.56 ± 0.14                 | 25.70 ± 19.69               |
> | 10.10     | -2.19 ± 0.20               | -0.78 ± 0.26                | -0.29 ± 0.20                |
> | 212.12    | -125.70 ± 4.36             | -2.12 ± 0.61                | -15.03 ± 7.94               |
> | 616.16    | -372.86 ± 12.51            | -2.17 ± 0.62                | -44.40 ± 23.51              |
> | 1000.00   | -607.64 ± 20.24            | -2.18 ± 0.63                | -72.31 ± 38.30              |
>
> Where values after ± are 95% CIs over 5 seeds. We can thus see that while LayerNorm quickly saturates for extreme inputs with both negative and positive signs, the same is not true for BatchNorm which exhibits the explosion, similarly as the Standard NN.
>
> **RKHS statement**
>
> The reviewer is correct in this assertion. However, in the particular case of a network with ReLU activation functions it is indeed true that the RKHS *only* contains functions exhibiting this property, from which the theorem intuitively follows, rather than an assumption being made on the RKHS as a result of the theorem. This can be seen explicitly in Corollary D.2.
>
> **The limitation that NTP is used for the theoretical results**
>
> This is again a limitation of the entire field making use of NTK theory. We would like to point out that for finite-width networks, the difference between standard (LeCun) parametrization (SP) and NTP is of no importance when considering proper scaling of the learning rate (instead manifesting only as a difference in the empirical NTK, which is never calculated), and as such all empirical evidence we provide also support applicability of the theory in practical applications using SP. Theoretical applicability to more modern parametrizations such as MuP would require extensions to the theory as mentioned above, although we note that, at initialisation, our assumptions on distribution of parameters remain true here, and thus it would be highly unintuitive that the learned function would sufficiently deviate from our theory.

---

> > ### Comment · Reviewer_WTJo · 2025-08-05
> >
> > I thank the authors for the detailed response and the additional experiments. I think the promised changes in all responses jointly will improve the paper. I believe the Tensor Program machinery might enable an analysis beyond the kernel regime, but I understand that this would entail a significant amount of work, and result in different main results. Overall, I will keep my slightly positive evaluation of the work.

---

### Official Review · Reviewer_uiPW · 2025-06-20

**Clarity:** 3
**Significance:** 3
**Originality:** 3
**Rating:** 5
**Confidence:** 1

**Summary:**

The paper proves—using Neural Tangent Kernel (NTK) theory—that inserting just one Layer Normalization (LN) anywhere in an (infinitely‑wide) fully‑connected network fundamentally changes the kernel: it becomes a bounded‑variance kernel. Consequently, the posterior mean of the corresponding Gaussian Process remains bounded for all inputs, no matter how far they lie from the training domain, while standard architectures without LN can diverge arbitrarily.
Key theoretical results are
1) Theorem 3.1: networks with positive n-homogeneous activations but no LN can yield unbounded outputs on some dataset/input pair.
2) Theorem 3.2 & B.2: adding a single LN forces the variance term Θ(x,x) of the NTK to be uniformly bounded.
3) Theorem 3.3: this bounded kernel implies a data‑dependent uniform bound on |E f(x)| for the trained network.
The authors confirm the theory on (i) 1‑D toy regressions, (ii) the UCI‑Protein RMSD task—testing proteins larger than seen in training, and (iii) UTKFace age‑prediction for ethnicities absent from training. In every case, architectures with ≥1 LN avoid “exploding” predictions and achieve markedly higher OOD R² than baselines.

**Questions:**

1) Finite‑width behaviour: Have you explored how large a width is needed for the theoretical bound to manifest? A sensitivity plot (width vs. max|f(x)| on extrapolation grid) would strengthen the practical message.

2) Beyond fully‑connected layers: Can Theorem B.2’s extension via Tensor Programs be instantiated for a modern ResNet/Transformer? A small experiment or discussion would clarify applicability.

3) Bound tightness: The worst‑case bound in Theorem 3.3 scales with √|D|/λ_min. In practice this may be very loose. Could the authors report empirical maxima and compare to the theoretical bound?

4) Activation assumptions: Assumption 3.2 excludes GELU/Swish (except via asymptotic argument). Are the bounds still valid for popular Transformer activations numerically?

**Ethical Concerns:**

["NO or VERY MINOR ethics concerns only"]

**Final Justification:**

I appreciate the authors' efforts to address my concerns. I raised my score to 5.

**Limitations:**

yes

**Paper Formatting Concerns:**

N.A.

**Quality:**

3

**Strengths And Weaknesses:**

Strengths
1) Strong Theoretical Guarantee: Provides a general and rigorous NTK-based proof that a single LayerNorm anywhere in an (infinitely-wide) fully connected network suffices to guarantee bounded network outputs for all inputs, regardless of distance from the training set.

2) Novelty: This is the first work to formally prove that LayerNorm fundamentally changes the extrapolatory behavior of deep networks, rather than merely showing this effect empirically.

3) Empirical Validation: The paper supports its theory with systematic experiments on toy regression tasks and two real-world benchmarks (protein RMSD and UTKFace age prediction), using multiple seeds and reporting confidence intervals.

4) Practical Safety Implications: Highlights the practical significance of output boundedness for safety-critical ML deployments, such as robot control.

5) Clarity of Empirical Results: Visualizations (e.g., Figures 1–3) and tables provide clear evidence of the “exploding” vs. “stable” behaviors, and OOD gains are easy to interpret.

Weaknesses
1) Limitation to Infinite-Width Theory: All theoretical results hold strictly in the infinite-width, full-batch regime; no non-asymptotic or finite-width generalization bounds are provided.

2) Limited Architectural Breadth: Experiments are restricted to shallow MLPs; the only use of a convolutional backbone (ResNet-16) is as a frozen feature extractor in the UTKFace task, so end-to-end results for CNNs or Transformers are not established.

3) Boundedness ≠ Correctness: The guarantee is about boundedness of output magnitude, not correctness, calibration, or improved prediction accuracy. The practical relevance for tasks not threatened by output explosions may thus be limited.

4) Dense Theoretical Presentation: Some notation and proof sketches (especially kernel recursions) are dense, and several technical steps are deferred to appendices, making parts of the analysis less accessible to non-experts.

5) LN Sufficient but Not Proven Necessary: The necessity of LayerNorm is empirically indicated (i.e., no alternative mechanism is shown to work), but not formally established; other mechanisms could potentially provide similar guarantees.

---

> ### Author Rebuttal · Authors · 2025-07-28
>
> We would like to thank the reviewer for spending time to read through our submission and write a comprehensive review. We are glad to hear the reviewer finds our theoretical results strong. We address all questions/concerns below. As per reviewer request, we ran a number of additional experiments and include the results below in the relevant sections.
>
> **Limitation to Infinite-Width Theory**
>
> While it is true our theoretical guarantees hold for infinitely wide networks, this is true for most of the theoretical results in the field of theoretical analysis of neural networks. As such, although it is a limitation, it is a limitation of the entire field at the moment rather than our paper specifically.
>
> **Beyond fully‑connected layers: Can Theorem B.2’s extension via Tensor Programs be instantiated for a modern ResNet/Transformer?**
>
> Yes, Theorem B.2 can be instantiated with any architecture that can be expressed via Tensor Programs, which includes Transformer and ResNet (and convolutional networks in general).
>
> We conduct a simple experiment with an **end-to-end training using a transformer architecture**, where during training the model is given sequences of form $y_i = a*i + b$, where i is the index that is restricted to [0,20] during training and $a$ and $b$ are sampled from standard normal. As such the model is trained to infer the pattern in the sequence and continue it. During testing, we use $a=1$, $b=1$ and use $i$ from range [0, 100]. We use a decoder-only transformer architecture with embeddings being linear projections from input space to latent space of size 64, we use 4 attention heads, and we use a total of 2 attention layers. We try two versions of this architecture, one with a single LayerNorm right after the input embeddings and the other one with no LayerNorm anywhere in the entire architecture. We provide the table of results over 5 seeds below:
>
> | Index ($i$) | Transformer w/ LN (95% CI)    | Transformer w/o LN (95% CI)     |
> |----------------|--------------------------------------|---------------------------------|
> | 50              | 55.4613 ± 11.591                 | 62.0827 ± 3.068                 |
> | 70              | 57.7766 ± 13.492                 | 146.1958 ± 59.857               |
> | 100            | 57.7941 ± 13.508                 | 412.1160 ± 276.456              |
>
> As such, it would appear the presence of a single LayerNorm makes the output quickly saturate, whereas lack of any LayerNorm causes an explosion to infinity. We plan on adding this experiment to the final version of the manuscript.
>
> **Activation assumptions:**
>
> We point the reviewer towards Theorem B.2 and Theorem B.3, which extend our boundedness results to a much more general class of activation functions, including all popular modern activation functions such as GeLU and Swish. The weaker statement was included in the main body for consistency with Theorem 3.1, which relies on analytical computation of the NTK, which is not tractable for such activation functions even if empirically true.
>
> To verify this claim experimentally, we rerun the sine experiment with different activation functions.  We present new results below (due to new policy we cannot share figure via links, so have to do it via a table):
>
> | X        | ReLU (No Norm)        | Swish (No Norm)      | GeLU (No Norm)       | ReLU (LN)            | Swish (LN)           | GeLU (LN)            |
> |----------|------------------------|-----------------------|-----------------------|-----------------------|-----------------------|-----------------------|
> | -1000.00 | 570.08 ± 79.62         | 276.34 ± 44.08        | 271.30 ± 32.08        | 2.63 ± 0.14           | 2.65 ± 0.47           | 2.49 ± 0.51           |
> | -595.96  | 338.44 ± 46.98         | 164.36 ± 26.30        | 161.29 ± 19.21        | 2.62 ± 0.14           | 2.64 ± 0.46           | 2.48 ± 0.51           |
> | -191.92  | 106.79 ± 14.34         | 52.44 ± 8.61          | 51.28 ± 6.36          | 2.56 ± 0.14           | 2.59 ± 0.46           | 2.43 ± 0.50           |
> | 10.10    | -2.19 ± 0.20           | -2.18 ± 0.25          | -1.74 ± 0.13          | -0.78 ± 0.26          | -0.83 ± 0.16          | -1.05 ± 0.23          |
> | 212.12   | -125.70 ± 4.36         | -57.29 ± 6.52         | -53.20 ± 7.23         | -2.12 ± 0.61          | -2.36 ± 0.39          | -2.67 ± 0.30          |
> | 616.16   | -372.86 ± 12.51        | -167.79 ± 18.82       | -156.46 ± 21.32       | -2.17 ± 0.62          | -2.40 ± 0.40          | -2.71 ± 0.30          |
> | 1000.00  | -607.64 ± 20.24        | -272.77 ± 30.50       | -254.56 ± 34.72       | -2.18 ± 0.63          | -2.41 ± 0.40          | -2.72 ± 0.30          |
>
>
> Where values after ± are 95% CIs over 5 seeds. We can thus see that regardless of the activation function, network with LayerNorm quickly saturates, whereas network without LN explodes.
>
> **Boundedness ≠ Correctness:**
>
> We agree with the reviewer on this point, as we wrote in the conclusion: “we do not claim that LN ‘solves’ the problem of extrapolation. Instead, we aimed to highlight the differences in extrapolatory behaviour of networks with and without LN, so that practitioners can make more informed architectural choices”. Indeed, making guarantees about OOD predictions would be likely impossible without further assumptions.
>
> **Dense Theoretical Presentation**
>
> We will use the additional page in camera-ready version to improve presentation and provide a more intuitive explanation of the derived results and proofs.
>
> **LN Sufficient but Not Proven Necessary**
>
> The reviewer is correct that one can guarantee bounded input in various other ways, simplest one being just manually clipping the output at some user-specified threshold. We focus on LayerNorm, as it is a popular component already present in many deep learning architectures, which interestingly causes this bounded output behaviour organically.
>
> **Finite‑width behaviour**
>
> As per request of the reviewer, we trained neural networks with different sizes on the sine synthetic dataset and recorded max value in the evaluation domain. We present the results below:
> | **Model Size** | $\max \|f(x)\|$ | **CI**       | **% Error to GP** |
> |----------------|------------------|--------------|-------------------|
> | 16             | 1.076239         | 0.049857     | 21.93%            |
> | 32             | 1.068731         | 0.019022     | 22.44%            |
> | 64             | 1.094828         | 0.025697     | 20.54%            |
> | 128            | 1.152339         | 0.022382     | 16.38%            |
> | 256            | 1.203813         | 0.036533     | 12.63%            |
> | 512            | 1.234375         | 0.019526     | 10.43%            |
> | 1024           | 1.282959         | 0.016616     | 6.88%             |
> | 2048           | 1.317124         | 0.007575     | 4.42%             |
> | NTK-GP      | 1.378103         | 0.000000     | 0.00%             |
>
>
> NTK-GP denotes the theoretically calculated max value of the infinite neural network (Gaussian Process with Neural Tangent Kernel). It would thus appear that for widths larger than 512, one can already get the max absolute prediction to converge within 10% of the max value of the theoretical infinite network.
>
> **Bound tightness:**
>
> As the bound grows with dataset size, we expect it to be relatively tight on small dataset, but it might become loose as the number of datapoints grows. As per reviewer request, below we plot the theoretical bound together with the observed empirical maximum for a network with LN after the first layer and a standard NN on the sine synthetic dataset, where we measure the maximum over the domain [-10K, 10K].
>
> | **Dataset Size** | **Theoretical Bound** | **LN** $\max \|f(x)\|$ | **Standard** $\max \|f(x)\|$ |
> |------------------|------------------------|--------------------------|-------------------------------|
> | 2                | 2.35                   | 0.73                     | 3304                          |
> | 3                | 5.11                   | 1.01                     | 3390                          |
> | 10               | 104.24                 | 0.61                     | 4031                          |
>
> The bound is of the same order of magnitude for small datasets, but starts to become loose for larger ones. However, the output of standard NN can easily exceed even that loose bound, so the bound is by no means vacuous.

---

> > ### Comment · Area_Chair_LxKd · 2025-08-07
> > **IMPORTANT REMINDER: PLEASE PARTICIPATE IN THE REBUTTAL**
> >
> > Dear Reviewer,
> >
> > I would like to remind you that reviewers MUST participate in discussions with authors.
> > If you don not engage, your review will be flagged as insufficient, and you have to live
> > with the consequences.
> >
> > best
> > your ac

---

> > ### Comment · Reviewer_uiPW · 2025-08-09
> >
> > I appreciate the authors' efforts to address my concerns. I will raise my score.

---

### Official Review · Reviewer_oWLy · 2025-06-29

**Clarity:** 3
**Significance:** 2
**Originality:** 2
**Rating:** 4
**Confidence:** 3

**Summary:**

This paper studies the impact of layer normalization on the output of neural network on out-of-distribution (OOD) data. The analysis is carried out using NTK (neural tangent kernel) of the neural network with layer
normalization. The main result is that, as long as there is one layer that performs layer normalization, the output of the trained NTK model will be bounded for out-of-distribution data, regardless of how large the
input is. The intuition is that, with such layer normalization, the normalized input will be bounded in expectation. Hence, the output of the network cannot be too large.

**Questions:**

1. The reviewer is a bit skeptical why keeping the output bounded is necessarily a desirable goal for out-of-distribution data. For example, consider Figure 1, the rightmost sub-figure. The desirable regressor appears to be a line, not a bounded function. If the line is indeed the ground truth, then having a bounded output just means that the difference from the ground truth will be unbounded! Then, why is the boundedness desirable?

2. The output bound in Theorem 3.3 depends on $\lambda_\min$. However, the value of $\lambda_\min$ itself may depend on the training data and network architecture. This dependency may make the advantage of the layer normalization less clear. For example, is it possible that, with layer normalization, the $\lambda_\min$ value become significantly smaller than that without layer normalization?

**Ethical Concerns:**

["NO or VERY MINOR ethics concerns only"]

**Final Justification:**

The response partially addresses my earlier doubts. Thus, I am willing to increase my score to be slightly positive.

**Limitations:**

yes

**Paper Formatting Concerns:**

None.

**Quality:**

3

**Strengths And Weaknesses:**

Strength:

1. The main result that one layer of normalization will keep the output bounded on out-of-distribution data, while not surprising, is interesting.

Weakness:

1. It is unclear why keeping the output bounded is the desirable goal for dealing with out-of-distribution data.

---

> ### Author Rebuttal · Authors · 2025-07-28
>
> We would like to thank the reviewer for taking time to read our submission and ask insightful questions. We are happy to know the reviewer finds our results interesting. We address the questions/concerns below:
>
> **Why keeping the output bounded is the desirable goal for dealing with out-of-distribution data**
>
> As we wrote in the Conclusion section:
> “As we go outside of data distribution, what exactly constitutes a “good” extrapolation is unknown to us unless further assumptions can be made. A stable (i.e. bounded) extrapolation, while desirable in many cases outlined in this paper, does not need to be always correct.”.
> Within the paper we merely wanted to point out that the introduction of LayerNorm makes the network output’s bounded, but whether that property is desirable on the problem at hand is a different question. We highlighted real-world examples, where this is the case, but ultimately, as we also wrote in Conclusions:
> “we do not claim that LN ‘solves’ the problem of extrapolation. Instead, we aimed to highlight the differences in extrapolatory behaviour of networks with and without LN, so that practitioners can make more informed architectural choices”.
>
> **Is it possible layer norm will shrink the eigenvalue**
>
> In certain situations, such as the case of very close training points very far from the origin, this is indeed possible. However, standard normalisation of the training set and appropriate architecture design limits the practical impact, and regularisation such as that suggested by “Observation Noise and Initialization in Wide Neural Networks” Calvo-Ordoñez et al. 2025 or even early stopping can arbitrarily lower-bound the eigenvalues altogether.
>
> We take the opportunity to stress here that it is not the value of the bound itself, which is derived as an absolute worst-case from Gaussian Process theory, but its mere existence which is the remarkable finding of this work, along with the implications for the RKHS of such networks. As an illustration of this, consider the case of a one-dimensional Gaussian Process with 0 prior mean, a kernel defined as $k(x,x’)=1$, and with infinitesimal observation noise. Such a model will clearly have a posterior mean function equal to a constant at the mean of the training outputs, despite having infinitesimal kernel matrix eigenvalues and being subject to the same bound as the one provided in the paper.

---

> > ### Comment · Reviewer_oWLy · 2025-08-03
> >
> > I wish to thank the authors for providing the response. However, the response does not exactly address my earlier doubts (on why bounded output is the desirable goal and whether the shrinking of eigenvalues in the proposed model can be controlled).. Thus, I will keep my score.

---

> > > ### Author Response · Authors · 2025-08-03
> > > **Expanded clarification - why boundedness is beneficial**
> > >
> > > We apologise that the reviewer did not find our earlier rebuttal sufficient to exactly address all concerns, and provide additional comments here to remedy this:
> > >
> > > **Why boundness is beneficial**
> > >
> > > It is typical for statistical modelling techniques to assume we can make predictions slightly beyond the range of data, but if we go too far off, our model might become arbitrarily erroneous (e.g. in Gaussian Process regression, this is controlled by the kernel lengthscale). Standard ReLU neural networks extrapolate linearly, which causes the explosion to infinity as we show in Theorem 3.1. While continuing a local trend might be accurate close to the training data, as we go further away, our trust in model correctness should decrease and thus we should be cautious of making extreme predictions.
> > >
> > > In the protein experiment we present in the paper, inspecting the histogram in Figure 2, one can see that a standard neural network is much more likely to predict extreme values of RMSD, which are highly implausible. In fact, we can see that the standard network made a substantial number of negative predictions, which are physically impossible, as RMSD is strictly positive. On this particular task, the LayerNorm network, which continues the local linear trend, before saturating at an upper bound is more accurate, precisely because it avoids these extreme, highly unlikely predictions.
> > >
> > > Ultimately, there might be some tasks where continuing a linear trend is the “correct answer”, however, we would argue that exploding to infinity is unlikely to be an accurate extrapolation in most real-world problems. For example, a network predicting energy or temperature of a physical system, should not be outputting arbitrarily large values. When modelling value functions in reinforcement learning, predicting an extremely large value for an unseen state, different from anyone we visited so far, is likely going to bias the algorithm and hinder learning. In any control task, outputting arbitrarily large values with no bound creates a serious risk hazard.
> > >
> > > As such, we believe in real physical systems, control and learning, boundness is rather a desired property. In our experiments we highlighted examples of modelling a physicochemical system (protein experiment) and learning in computer vision (UTK), where this is the case.
> > >
> > > We also want to highlight that the primary aim of our paper was understanding the behaviour and properties of networks equipped with LayerNorm. We are not proposing that they should necessarily be used under every circumstance, but we believe (and this belief is supported by the extensive use of layer norms in modern deep learning) there will be numerous problems where these properties are beneficial.
> > >
> > > As a final point, note that whenever one chooses a network architecture they are inadvertently expressing a prior belief over the solution to the problem at hand. It is thus essential to fully understand the implications of any architectural decision, such that the practitioner can ensure that their true prior belief is indeed translated correctly to the model. This is analogous, for example, to choosing a periodic vs non-periodic kernel in a Gaussian process regression. Given the pervasiveness of layer norms in modern deep learning, we believe fully understanding their implications is absolutely essential, both for confirming correct translation of a prior belief that we should not deviate too far in prediction from the training data as will be the case for most real-world tasks, and indeed preventing this boundedness in the tasks where it is undesirable.

---

> ### Author Response · Authors · 2025-08-03
> **Expanded clarification - controlling the shrinking of the eigenvalues of the proposed model**
>
> **Controlling the shrinking of the eigenvalues of the proposed model**
>
> To address this concern, we now provide mathematical analysis on the effect of including the layer norm. For the sake of brevity, we restrict this to the case of an $L$-layer network equipped with ReLU activations, biases initialised with variance $1$, and a single layer norm positioned at some layer $h_{LN}$, although this analysis can easily be extended to any situation considered in the paper. From the proof of Theorem 3.2, we see that the NTK of this network, $\Theta_{LN}(x,x’)$ differs from that of the equivalent network without a layer norm operation, $ \Theta(x,x’)$ by a factor of 1 over the geometric mean of the variances of the pre-activations at layer $h_{LN}$, or $\frac{1}{\sqrt{(h_{LN}+\lVert x\rVert)(h_{LN}+\lVert x’\rVert)}}$. Thus, the Gram matrix of the NTK for the LN network, $K_{LN}$, relates to the Gram matrix of the no-LN network, $K$, by $K=DK_{LN}D$, where $D_{ij}=\frac{1}{\sqrt{h_{LN}+\lVert X_i\rVert}}\delta_{ij}$. Hence in the absolute worst case, the minimum eigenvalue of the LN network is scaled from the minimum eigenvalue of the no-LN network by a factor of $\frac{1}{h_{LN}+\max_{i}\lVert X_i\rVert}$.
> Under standard data normalisation practices, this is not likely to be an issue. That said, to alleviate any further concerns the reviewer may have we provide analyses of the two proposed regularisation methods to mitigate this issue. For the case of observation noise regularisation (an L2 regulariser from initialisation in weight space) of strength $\sigma_n^2$, the effective eigenvalues used by the theorem are given by $\lambda_{eff}=\lambda+\sigma_n^2$, trivially and arbitrarily lower-bounding the minimum eigenvalue in question. For the case of early stopping at time $T=n_{\text{iter}}\eta$ for some learning rate $\eta\approx0$, the effective eigenvalues are given by $\lambda_{eff}=\frac{\lambda}{1-\exp(-\lambda T)}$ (or an equivalent expression for discrete gradient descent (large $\eta$)), which lower bounds the minimum eigenvalue in question by $\frac{1}{T}$.
> Finally, we again direct the reviewer’s attention to the empirical results presented in the paper, which not only directly show that in practice this boundedness effect is always observed, but that the bound itself is very rarely reached due to its worst-case nature. We have thus demonstrated that the shrinking of $\lambda_{\min}$ can occur, but in practice this does not have implications for the extrapolatory properties of a layer norm network, even aside from the RKHS arguments, and can easily be mitigated (in a sense this is always the case as training is always for a finite time). We trust this expanded analysis is to the reviewer’s satisfaction.

---

> ### Comment · Reviewer_oWLy · 2025-08-06
>
> I thank the authors for the additional explanations. However, if the minimum eigenvalue decreases inversely proportional to ||x||, won't it invalidate the whole boundedness result? Specifically, your finiteness bound is supposed to hold for arbitrary large ||x||. If after LN, the minimum eigenvalue becomes so small, I suppose that your bound will no longer be finite for arbitrary large ||x||?

---

> > ### Author Response · Authors · 2025-08-06
> >
> > We thank the reviewer for the nuanced question. The subtlety here is that the minimum eigenvalue scales inversely proportionally to the the norms of the *training* inputs, not the query or test input post-training, which is the input of relevance $x$ for all our theorems. Regardless of normalisation, it is reasonable to assume that the training inputs will always be of finite magnitude, and thus the minimum eigenvalue will be nonzero (if very small), meaning that the bound on the output for arbitrarily large *query* input is indeed finite, as the training dataset on which this bound solely depends is fixed at this point.
> >
> > In other words, for finite training data the bound on the network output for an arbitrarily large test input is **finite**, as the minimum eigenvalue is fixed after training and has no dependency on the test input.
> >
> > As another aside, the assumption on the strict positivity of the minimum eigenvalue is required for all results derived from NTK theory. Finally, again note that the effect of the layer norm on this minimum eigenvalue is controlled entirely by the scale of the training data, which can also be arbitrarily rescaled (as is standard practice anyway) to explicitly control this eigenvalue.

---

> > > ### Comment · Reviewer_oWLy · 2025-08-08
> > >
> > > Thank you for the additional explanation! I think I agree with you that the minimum eigenvalue will unlikely cause a major problem for your final result. I still think that the objective of keeping output bounded is questionable, but I think it is okay to have different opinions. Thus, I am willing to slightly increase my score.

---

### Official Review · Reviewer_L8DJ · 2025-07-02

**Clarity:** 3
**Significance:** 1
**Originality:** 2
**Rating:** 2
**Confidence:** 4

**Summary:**

This paper studies the neural tangent kernel (NTK) with layer normalization, showing that the corresponding kernel predictor remains bounded for any input. In contrast, without layer normalization, neural network outputs can become pathologically large. The theoretical results are validated through experiments on synthetic and UCI datasets.

**Questions:**

In the proof of Proposition 3.4, how do you establish the existence of such a network? The statement "There exists a network with fully-connected layers, operating in an overparameterized regime such that..." requires clarification.

**Ethical Concerns:**

["NO or VERY MINOR ethics concerns only"]

**Final Justification:**

This paper is positioned as a theory paper, but I did not find any new insights or techniques in the analysis. Therefore, I tend to keep my score unchanged.

**Limitations:**

The theoretical contribution is incremental.

**Paper Formatting Concerns:**

I did not notice any major formatting issues in the paper.

**Quality:**

2

**Strengths And Weaknesses:**

Strengths:

The paper is well-written with clear logic, and experiments effectively support the theoretical results.

Weaknesses:

1. The contribution is incremental—the result is somewhat expected given the assumption of homogeneous activation functions.
2. The motivation for studying this problem is unclear. In practical settings, input data are typically not large due to standard normalization practices. The term "stable" may be inappropriate here, as large inputs can be easily controlled and are not a primary source of instability.
Studying stability within bounded domains would be more meaningful.
3. The definition of layer normalization should be introduced in the preliminaries.

---

> ### Author Rebuttal · Authors · 2025-07-28
>
> We would like to thank the reviewer for reading our submission and providing feedback. We are glad to know the reviewer thinks our experiments effectively support the theoretical results. We address all questions/ concerns below.
>
> **Results expected given the homogeneous activation functions**
>
> As we show in Theorem 3.1, a network with homogenous activation functions, but without layer norm **does not** enjoy the stability guarantee. As such the homogeneity assumption by itself does not suffice to guarantee bounded outputs. As we then present in Theorem 3.3, it is the presence of at least one layer norm in the architecture that grants the stability property.
>
> It is thus the presence of Layer Norm that guarantees the stability, not the homogeneity. We would like to thus politely disagree with the statement that the homogeneity makes the results expected. In fact, when it comes to our theoretical contributions, Reviewer QsAy described our results as “surprising”, Reviewer uiPW listed novelty as one of the strengths of our submission and Reviewer WTJo described our results as "interesting", while judging our assumptions to be “weak”.
>
> **The motivation for studying this problem is unclear**
>
> On any finite domain $D$, one can always bound the network outputs by $\max_{x \in D}|f(x)|$ and as such, basically any network is stable in this sense. What matters is how large is the bound $\max_{x \in D}|f(x)|$.
>
> The result of Theorem 3.1 states that the supremum over an infinite domain is infinity, meaning that we can always find an input that causes an arbitrary large output. This means that if the inputs are restricted to some domain $D$, then as the domain size grows, the maximum output of a network with ReLU activation functions, $\max_{x \in D}|f(x)|$, also grows, by the same argument as used in the proof of Theorem 3.1. However, for a network with a LayerNorm, there is an upper bound that is unaffected by the domain size (and thus by the distance from training data).
>
> A more important consideration here, however, is the prevalence of significantly out of distribution inputs at test-time etc. Standard normalisation practices assume the training distribution is representative of the true distribution encountered throughout the model’s use, however in practice this is well known to be untrue. This is precisely the notion of extrapolation, which is a problem of universally acknowledged importance. Furthermore, in high-dimensional problems it is well-known that almost all real inputs will be extrapolatory (for example, see “Learning in High Dimension Always Amounts to Extrapolation”, Balestriero et al. 2021), thus understanding the behaviour of models in this setting is of considerable importance. We direct the reviewer towards our experiments section for empirical demonstration of this.
>
> **Layer Norm not defined**
>
> The reviewer is absolutely right, that for completeness we should explicitly define the layer norm operation in the main body. We will add this definition in the camera ready revision.
>
> **Establishing the existence of the network in Proposition 3.4**
>
> This is a straightforward application of the well-known universal approximation theorem, which guarantees the existence of a network capable of learning any function. For a non-degenerate training set, it is clear that any function perfectly interpolating the training points will achieve the minimum MSE loss, and for a degenerate training set, i.e. one containing repeated training inputs with varying training outputs, there will be an optimal “virtual” point to be interpolated corresponding to the maximum likelihood estimator for the true function function value there under a homoscedastic Gaussian noise model. This establishes the existence of a function achieving minimum training error. By the universal approximation theorem, there exists a network capable of learning such a function, and thus achieving the minimum possible training error. It then follows that the addition of further parameters cannot lower this training loss, the rest of the proof follows.

---

> > ### Comment · Reviewer_L8DJ · 2025-08-03
> >
> > I thank the authors for the explanations.
> >
> > Just to clarify, I have never claimed that homogeneity guarantees stability. My point is that homogeneous activation functions allow the network output to grow with the magnitude of the input. In that case, results such as Theorem 3.1 are expected. If one instead uses activation functions with bounded outputs, such as the sigmoid, then the network output is naturally bounded, and I believe a stability result can be established without layer norm.
> >
> > Overall, I find the intuition to be straightforward, and the paper does not appear to offer new insights or techniques in the analysis.

---

> ### Author Response · Authors · 2025-08-04
>
> We thank the reviewer for the clarification of their intuition on homogeneous activation functions. Without specific criticism to address beyond a disinterest in the paper, which we again note is an opinion held in isolation from the other reviewers, we instead offer the following further details:
>
> **Boundlessness of (n>0)-homogeneous networks**
>
> While it is indeed intuitive that a network equipped only with homogeneous activation functions will grow without bound, this is by no means guaranteed when disregarding training dynamics; indeed, it is trivial to construct a network by hand which achieves minimum training loss on any given finite dataset while extrapolating at a constant. The power of NTK theory, which we again stress is one of a very few respected theories explaining the behaviour of neural networks, is in allowing us to apply rigorous mathematical treatments to these networks, leading to results such as the ones presented in this paper which provide theoretical support to existing intuitions, and in many cases can provide new and useful ones.
>
> As a final comment, focusing excessively on Theorem 3.1 somewhat misses the point of the paper, as while the theorem is indeed a novel extension of existing analyses of standard neural networks, its main purpose in this work is providing context for the later theorems and experiments.
>
> **Boundedness of 0-homogeneous activations**
>
> We agree with the reviewer that trained networks equipped with only (asymptotically) 0-homogeneous activation functions are trivially bounded for all but noiseless, exactly linear datasets. One can take this a step further and pass the network’s output through a bounded-range mapping, or even clip the network output directly. However, these measures are very rarely applied in modern deep learning (with the obvious exception of classification tasks), which overwhelmingly favours asymptotically positive 1-homogeneous activation functions.
>
> What is, however, exceptionally pervasive in modern deep learning is the layer norm operation which features by default in every pytorch-implemented transformer, for example. It is hard to believe that any practitioner of such a network made the conscious decision to bound the extrapolation of their network; much more likely is the case that they wished to take advantage of the well-studied benefits to training stability (noting the very different notion of stability here). It should then be clear why understanding the implication of such an unassuming architecture choice is absolutely essential for informed application of such networks -- for instance, unstable extrapolation may accurately reflect the practitioner’s prior for the problem at hand? In that case, layer norm operations should be avoided, and yet without the results in this paper, there is no formal reason why this should be the case. Finally, we note that unlike the trivially bounding cases mentioned above, without the NTK-based analysis no such guarantee can be derived for the case of layer norm -- a notion we formalise in Proposition 3.4.
>
> We trust this additional discussion will be of some aid in exposing the nuances of the paper and the significance of our theoretical results.

---

### Official Review · Reviewer_QsAy · 2025-07-02

**Clarity:** 3
**Significance:** 3
**Originality:** 3
**Rating:** 5
**Confidence:** 3

**Summary:**

The paper studies the boundedness of neural networks with layer norm in the infinite width NTK regime. The first result is that there is a “bad” dataset such that training the network on it will result in an unbounded function. Next, it is shown that on any training set, if the network contains layer norm, then it converges to a bounded function. Finally, it is shown that if we consider only ERM classifiers (rather than considering the training dynamics), then layer norm is not enough to bound the converged network. Several experiments are given that verify the results on toy and realistic datasets.

**Questions:**

Would it be tractable to make the analysis in the finite width NTK? Will similar results still hold, assuming a large but finite width?

**Ethical Concerns:**

["NO or VERY MINOR ethics concerns only"]

**Final Justification:**

I think this is a nice theory paper. It provides some insights on the boundness of layernorm, although I agree with the other reviewers, it doesn't provide new techniques, and it is limited to the infinite width NTK regime. I retain my score, but slightly lowered my confidence in light of this.

**Limitations:**

Yes

**Quality:**

3

**Strengths And Weaknesses:**

Strengths:
- The results in this paper are strong and somewhat surprising. First, they imply that one of the major advantages of using LN, which is very popular in modern architectures, specifically transformers, is that it makes the resulting function bounded. Second, that the boundedness indeed depends on the learning procedure, and performing ERM (not through gradient methods) may result in a non-bounded function.
- The experiments are quite thorough. They also show that not only in the NTK regime, the boundedness property holds. This is important since, in practice, networks are not trained in the kernel regime.

Weaknesses:
- I think the main weakness in the paper is that the authors analyze LN in the infinite width NTK regime. Now, I understand that using the NTK regime makes the analysis tractable, but why is the infinite width necessary? Looking at the proofs, I think they can be generalized to the finite width NTK regime, with slightly more work.
- This is a paper about layer normalization, where layer normalization is not defined. I highly suggest devoting some space in the preliminaries to properly define LN, since even if it is “common knowledge” for the community, in theoretical papers, it would make the results easier to understand.

Minor comments:
- Typo - Line 66 should be 1 until n
- Proposition 3.4 seems to be true for any dataset. It would be better to phrase it this way; it also makes the proposition stronger.

To conclude, I think this is a strong paper with interesting results and a thorough experimental study.

---

> ### Author Rebuttal · Authors · 2025-07-28
>
> We would like to thank the reviewer for spending the time to read our submission. We are happy to know the reviewer finds our results strong and the experiments thorough. We address all questions/ concerns below.
>
> **Extending results to finite width NTK regime**
>
> As is standard within all NTK-based research, our main arguments make use of the central limit theorem, law of large numbers, and constancy of the empirical neural tangent kernel as a means of simplifying analysis to the point of tractability, which require the infinite-width setting for deterministic application. While some methods exist for studying the finite-width regime, these would considerably complicate an already substantial analysis, and thus a full theoretical treatment of this setting is beyond the scope of this work. We point the reviewer instead towards the empirical evidence for this behaviour in finite networks, again as is standard in all NTK-based research.
>
> **Layer Norm not defined**
>
> The reviewer is absolutely right, that for completeness we should explicitly define the layer norm operation in the main body. We will add this definition in the camera ready revision.

---

> > ### Comment · Reviewer_QsAy · 2025-08-03
> >
> > I thank the authors for the response. After reading the other reviews and the rebuttal I still feel this is an interesting paper and I will keep my score.

---

### Note · Authors · 2025-08-14

We believe the rebuttal sufficiently addressed the reviewers' concerns and resulted in two of the reviewers increasing their score. At present, four out of five reviewers are positive about our paper. The only reviewer expressing a negative opinion, does not raise a specific criticism and their only concern is alleged lack of significance. However, we would like to emphasise that this opinion is held in complete isolation, as all of the remaining reviewers described our results as interesting and/or surprising.

---

### Decision · Program_Chairs · 2025-09-17

**Decision:**

Accept (poster)

**Comment:**

The paper shows that in an infinite-width NTK regime
one layer norm guarantees that the outputs of a network
trained until convergence are bounded (and not stable as
suggested by the title), while in the absence of a
layer norm this is not the case for certain, malign data sets.

During the discussion, two reviewers raised their score
so that now 4 out of 5 are voting for acceptance. The fifth
reviewer remained sceptical staying at his rating 2.
Here are his major concerns:

- infinite-width NTK regime and parametrization
- results are not surprising in view of the assumed homogeneity of the activation function
- the paper is not about `extrapolation stability' as it only considers extrapolation boundedness, which could also
  be ensured by other means

All of these concerns have certainly some validity and are shared to some extend
by the other reviewers (though one reviewer finds the results surprising).
On the other hand, the paper also has its merits:

- rigorous analysis and novelty of results
- good supporting experiments

In summary, making a decision for this paper is really about weighting
the pros and cons. Personally, I think, the two reviewers rating it
by 5 may be a bit too positive, while the negative reviewer is certainly
way too negative.

I tend to vote to accept the paper, although some comparison to
other papers is probably a good idea.